# Self-Boost via Optimal Retraining: An Analysis via Approximate Message Passing

**Adel Javanmard**
University of Southern California
Google Research
ajavanma@usc.edu

**Rudrajit Das**
Google Research
dasrudrajit@google.com

**Alessandro Epasto**
Google Research
aepasto@google.com

**Vahab Mirrokni**
Google Research
mirrokni@google.com

## Abstract

Retraining a model using its own predictions together with the original, potentially noisy labels is a well-known strategy for improving the model's performance. While prior works have demonstrated the benefits of specific heuristic retraining schemes, the question of how to optimally combine the model's predictions and the provided labels remains largely open. This paper addresses this fundamental question for binary classification tasks. We develop a principled framework based on approximate message passing (AMP) to analyze iterative retraining procedures for two ground truth settings: Gaussian mixture model (GMM) and generalized linear model (GLM). Our main contribution is the derivation of the Bayes optimal aggregator function to combine the current model's predictions and the given labels, which when used to retrain the same model, minimizes its prediction error. We also quantify the performance of this optimal retraining strategy over multiple rounds. We complement our theoretical results by proposing a practically usable version of the theoretically-optimal aggregator function and demonstrate its superiority over baseline methods under different label noise models.

## 1 Introduction

Learning effectively from data with noisy labels remains a significant challenge in supervised machine learning (ML). A simple approach to mitigate the impact of label noise involves leveraging the trained model's own (soft or hard) predictions to ameliorate the learning process. More specifically, prior works [25, 11, 10, 32, 9] have shown that **retraining** an already trained model using its own predictions and the given labels (with which the model is initially trained) can lead to higher model accuracy, especially when the given labels are noisy. [11, 10, 32, 9] theoretically quantify the gains of simple retraining schemes and also provide principled explanations of why retraining is beneficial, such as the predicted labels leading to variance-reduction, leveraging large separability of the classes, etc. However, a key question that remains unanswered is the following:

*"What is the optimal way to utilize the model's predicted labels and the given labels?"*

In this work, we address this question for binary classification problems where the underlying ground truth model is: (*i*) a Gaussian mixture model (GMM), or (*ii*) a generalized linear model (GLM). Further, we also analyze the effect of multiple rounds of retraining. More specifically, for a sample $\boldsymbol{x}$, suppose $\widehat{y}$ is the given noisy label and $y^t$ is the model's predicted label at the $t^{\text{th}}$ round of retraining,

39th Conference on Neural Information Processing Systems (NeurIPS 2025).

then we derive the Bayes optimal aggregator function $g_t(y^t, \widehat{y})$ which should be used to retrain the model in the $(t+1)^{\text{th}}$ round in order to minimize its prediction error. Our analysis technique is very different from the aforementioned prior works in this space and is based on Approximate Message Passing (AMP) [13, 35, 18]. We believe ours is the *first work* that addresses this question of optimally using the predicted and given labels. Building on our theoretical framework, we also develop an aggregator function that can be used in practice for training with the binary cross-entropy loss.

Before stating our contributions, we need to briefly introduce the problem setting. We consider a binary classification setting (with labels $\in \{\pm 1\}$), where for each sample $x$, the given label $\widehat{y}$ is equal to the true label $y$ with probability $1 - p > \frac{1}{2}$ and the flipped label $-y$ otherwise, independently for each sample. In the Gaussian mixture model (GMM) case, we have $x = y\mu + z$, where $\mu \in \mathbb{R}^d$ and $z \sim \mathsf{N}(0, I)$. In the generalized linear model (GLM) case, we have $\mathbb{P}(y = 1 | x) = h(x^\mathsf{T}\beta)$ for some $\beta \in \mathbb{R}^d$ and a known function $h(\cdot)$. We are now ready to state our **main contributions**.

**(a)** We introduce and analyze an iterative retraining scheme based on *Approximate Message Passing (AMP)* [13, 35, 18]; see (4)-(5) for GMM and (17)-(18) for GLM. We provide a precise characterization of the effect of retraining in an asymptotic regime, where the ratio of the number of examples to the feature dimension converges to a constant. In particular, we derive the *state evolution* recursion for our setting – a concise deterministic recursion that captures the asymptotic behavior of the AMP-based estimator; see (8), Theorem 3.1 for GMM and (22), Theorem 4.2 for GLM.

**(b)** We derive the *Bayes optimal aggregator function* $g_t(y^t, \widehat{y})$ to combine the model's prediction $y^t$ at the $t^{\text{th}}$ round of retraining and the given label $\widehat{y}$ to be used in the $(t+1)^{\text{th}}$ round of retraining for *minimizing prediction error* in Theorems 3.2 and 4.3 for GMM and GLM, respectively. To our knowledge, ours is the *first work* to analyze the optimal way to use the predicted and given labels for any retraining-like idea in any setting.

**(c)** Based on our theoretical analysis, we develop a strategy that can be used in practice to combine the given labels and predictions for training any model with the binary cross-entropy loss (Section 5). We show that our method outperforms existing retraining baselines under different label noise models.

## 2 Related Work

(*i*) **Retraining (fully supervised setting).** [9] theoretically analyze the idea of retraining a model with only its predicted hard labels and not reusing the given labels when they are noisy; they call this "full retraining". [9] also propose "consensus-based retraining" which is the process of retraining only using the samples for which the predicted label matches the given label. Another related idea is "self-distillation" (SD) [15, 26], where a teacher model's predicted *soft* (and not hard) labels are used to train a student model having the same architecture as the teacher. [25] empirically demonstrate that SD can improve performance when the given labels are noisy. [11, 10] theoretically analyze the benefits of one round of SD in the presence of noisy labels, while [32] analyze the effect of multiple rounds of SD in a setting similar to [10]. [39] do a statistical analysis of SD in the asymptotic limit, compare the effect of soft labels & hard labels, and also investigate the effect of multiple rounds of SD. But unlike us, [39] do not analyze the optimal way to combine given labels and model predictions.

(*ii*) **Self-training (ST).** In the semi-supervised setting, ST [38, 43, 23] is the process of iteratively training a model wherein it is initially trained using the labeled samples, then labeling the unlabeled samples, followed by retraining the model on the labeled samples as well as the unlabeled samples on which the model is confident. This process is often repeated a few times. While this sounds similar to retraining, our work is in the *fully supervised* setting and retraining does not particularly rely on the model's confidence. See [2] for a survey on ST and some related approaches. There is also a significant amount of theoretical work on characterizing the efficacy of ST and related approaches [7, 34, 21, 8, 31, 41, 44]; but these results are not in the presence of noisy labels. Furthermore, there are empirical ideas related to ST in the context of label noise [37, 40, 17, 30, 24, 16].

(*iii*) **Approximate Message Passing.** Due to lack of space, we defer this to Appendix A.

**Notation:** We use the boldface symbols to denote vectors and matrices. The $\ell_2$ norm of a vector $v$ is denoted by $\|v\|_{\ell_2}$. For any $n \in \mathbb{N}$, the set $\{1, \ldots, n\}$ is denoted by $[n]$. With a slight abuse of notation, we use $\mathbb{P}(X)$ to indicate the probability mass function (for discrete random variable $X$) as well as the probability density function (for continuous variable $X$). We write $\xrightarrow{p}$ to denote

convergence 'in probability'. $\Phi(z) = \frac{1}{\sqrt{2\pi}} \int_{-\infty}^{z} e^{-(u^2/2)} \mathrm{d}u$ denotes the standard gaussian CDF. A function $\psi : \mathbb{R}^p \mapsto \mathbb{R}$ is said to be pseudo-Lipschitz (of order 2) if for all $\boldsymbol{u}, \boldsymbol{v} \in \mathbb{R}^p$, we have $|\psi(\boldsymbol{u}) - \psi(\boldsymbol{v})| \le L(1 + \|\boldsymbol{u}\| + \|\boldsymbol{v}\|)\|\boldsymbol{u} - \boldsymbol{v}\|$ for some constant $L$.

# 3   Gaussian Mixture Model (GMM)

**Data model.** We assume the training data $\{(\boldsymbol{x}_i, y_i)\}_{i=1}^n$ are generated i.i.d according to a Gaussian mixture model. In this model, each data point belongs to one of two classes $\{\pm 1\}$ with corresponding probabilities $\pi_+, \pi_-$, such that $\pi_+ + \pi_- = 1$. Denoting by $y_i \in \{-1, +1\}$ the label for data point $i$, the features vectors $\boldsymbol{x}_i \in \mathbb{R}^d$, for $i \in [n]$, are generated independently as

$$\boldsymbol{x} = y\boldsymbol{\mu} + \boldsymbol{z}, \tag{1}$$

where $\boldsymbol{z} \sim \mathsf{N}(\boldsymbol{0}, \boldsymbol{I})$. In other words the mean of features vectors are $\pm\boldsymbol{\mu}$ depending on its class. Throughout, we denote the features matrix and the labels vector by

$$\boldsymbol{X} = [\boldsymbol{x}_1 | \ldots | \boldsymbol{x}_n]^\mathsf{T} \in \mathbb{R}^{n \times d}, \quad \boldsymbol{y} = [y_1, \ldots, y_n]^\mathsf{T} \in \mathbb{R}^n .$$

The learner does not observe the labels $y_i$, rather she has access to features vectors $\boldsymbol{x}_i$ and noisy labels $\widehat{y}_i$ which are generated according to the following model:

$$\widehat{y}_i = \begin{cases} y_i & \text{with probability } 1 - p, \\ -y_i & \text{with probability } p, \end{cases} \tag{2}$$

where $p < \frac{1}{2}$ is the mislabeling or label flipping probability. Let $\widehat{\boldsymbol{y}} = [\widehat{y}_1, \ldots, \widehat{y}_n]^\mathsf{T} \in \mathbb{R}^n$ denote the vector of the observed noisy labels.

**Standard linear classifier model.** Let us first consider the following simple classifier model:

$$\widehat{\boldsymbol{\theta}} = \frac{1}{n} \boldsymbol{X}^\mathsf{T} \widehat{\boldsymbol{y}}. \tag{3}$$

The above model has been used before in [9, 7].[1] For a given point $\boldsymbol{x}$, the model's soft prediction is $\boldsymbol{x}^\mathsf{T} \widehat{\boldsymbol{\theta}}$ and its predicted label is $\mathrm{sign}(\boldsymbol{x}^\mathsf{T} \widehat{\boldsymbol{\theta}})$.

In this work, we consider a slightly different model inspired by Approximate Message Passing (AMP). Since our focus is on retraining over multiple rounds, we will delineate this as an iterative procedure.

## 3.1   Retraining Framework Inspired by Approximate Message Passing

For ease of exposition, we begin by giving a high-level outline of the retraining process. At iteration $t$ of the process, let $\boldsymbol{\theta}^t \in \mathbb{R}^d$ denote the current model and $\boldsymbol{y}^t \in \mathbb{R}^n$ denote the vector of the soft predictions of the model on the training data. In the next iteration, the algorithm combines $\boldsymbol{y}^t$ and the observed noisy labels $\widehat{\boldsymbol{y}}$ using the function $g_t$ to obtain $g_t(\boldsymbol{y}^t, \widehat{\boldsymbol{y}})$ which are the target labels for retraining in the $(t+1)^{\text{th}}$ iteration. We refer to $g_t$ as *aggregator* function and keep it general for now. We will later discuss different options for $g_t$, including the Bayes-optimal choice.

Let us now delve into the details. With an aggregator function $g_t$, we have the following update rule for $t \ge 0$:

$$\boldsymbol{\theta}^{t+1} = \frac{1}{\sqrt{n}} \boldsymbol{X}^\mathsf{T} g_t(\boldsymbol{y}^t, \widehat{\boldsymbol{y}}) - C_t \boldsymbol{\theta}^t \qquad \text{(model-update step)}, \tag{4}$$

$$\boldsymbol{y}^{t+1} = \frac{1}{\sqrt{n}} \boldsymbol{X} \boldsymbol{\theta}^{t+1} - g_t(\boldsymbol{y}^t, \widehat{\boldsymbol{y}}) \frac{d}{n} \qquad \text{(soft-prediction step)}. \tag{5}$$

The function $g_t$ is applied entry wise, i.e., $g_t(\boldsymbol{y}^t, \widehat{\boldsymbol{y}})$ is the vector whose $i^{\text{th}}$ ($i \in [n]$) entry is given by $g_t(y_i^t, \widehat{y}_i)$. The coefficient $C_t \in \mathbb{R}$ is given by

$$C_t = \frac{1}{n} \sum_{i=1}^{n} \frac{\partial g_t}{\partial y}(y, \widehat{y}_i)\Big|_{y = y_i^t} \tag{6}$$

---

[1]As mentioned in [9], this is a reasonable simplification of the least squares' solution for analysis purposes.

We initialize this process with $g_0(\cdot, \widehat{\boldsymbol{y}}) = \widehat{\boldsymbol{y}}$; notice that $C_0 = 0$.

There are two crucial differences from the standard model in (3): (*i*) instead of having a normalization factor of $1/n$ in the model-update step, we split it between the model-update and soft-prediction steps in our process by incorporating a factor of $1/\sqrt{n}$ at each step. Note that the scaling of the estimator does not matter since the predicted labels only depend on the direction of the estimator. (*ii*) we have 'memory correction' terms $(-C_t \boldsymbol{\theta}^t$ and $-g_t(\boldsymbol{y}^t, \widehat{\boldsymbol{y}}) \frac{d}{n})$ in both steps.

The updates (4) and (5) are in the form of Approximate Message Passing (AMP), which was introduced by adapting ideas from graphical models (belief propagation) and statistical physics to estimation problems [13, 35, 18]. The memory correction terms $-C_t \boldsymbol{\theta}^t$ and $-g_t(\boldsymbol{y}^t, \widehat{y}) \frac{d}{n}$ (also called 'Onsager' correction in statistical physics and the AMP literature) can be thought of as a momentum term, and plays a key role in ensuring that the asymptotic distributions of $(\boldsymbol{\theta}^t, \boldsymbol{y}^t)$ are Gaussian. To build some insight on the role of these memory terms, note that the data matrix $\boldsymbol{X}$ is fixed across iterations, and so $\boldsymbol{\theta}^t$, $\boldsymbol{y}^t$ and $\boldsymbol{X}$ are correlated, which induces some bias in the estimates. The memory terms are designed specifically to act as debiasing terms to compensate for this dependence. Specifically, the effect of these corrections is the same as an iterative procedure without these terms, wherein the data matrix is resampled at every iteration, making it independent from the current estimates (as pointed out by [4, 12]). Of course the latter is not a practical algorithm, since the data matrix is fixed, but it is shown that both will have the same limiting behavior.

## 3.2 Analysis of the Retraining Process

**Assumption 1** *We assume the following:*

- *As $n, d \to \infty$, the ratio $d/n \to \alpha \in (0, \infty)$.*

- *The empirical distributions of the entries of $(\sqrt{d}\boldsymbol{\mu})$ (recall $\pm\boldsymbol{\mu}$ are the class means; see (1)) converges weakly to a probability distribution $\nu_M$ on $\mathbb{R}$ with bounded second moment. Let $\gamma^2 = \mathbb{E}_{\nu_M}[M^2]$.*

- *The function $g_t : \mathbb{R} \times \mathbb{R} \mapsto \mathbb{R}$ is Lipschitz continuous.*

We first characterize the test error of a model $\boldsymbol{\theta}$ under our data model.

**Test classification error.** For a model $\boldsymbol{\theta}$, its predicted label for a test point $\boldsymbol{x}$ is given by $\text{sign}(\boldsymbol{x}^\top \boldsymbol{\theta})$. Therefore, the classification error amounts to (recall that $\Phi(u) = \frac{1}{\sqrt{2\pi}} \int_{-\infty}^u e^{-t^2/2} \mathrm{d}t$):

$$P_e(\boldsymbol{\theta}) = \mathbb{P}(y\boldsymbol{x}^\top \boldsymbol{\theta} < 0) = \mathbb{P}(y(y\boldsymbol{\mu}^\top + \boldsymbol{z}^\top)\boldsymbol{\theta} < 0) = \mathbb{P}(y\boldsymbol{z}^\top \boldsymbol{\theta} < -\boldsymbol{\mu}^\top \boldsymbol{\theta}) = \Phi\left(-\frac{\boldsymbol{\mu}^\top \boldsymbol{\theta}}{\|\boldsymbol{\theta}\|_{\ell_2}}\right). \quad (7)$$

**State evolution.** A remarkable property of AMP algorithms is that their high-dimensional behavior admits an *exact description*. In essence, the vectors $\boldsymbol{\theta}^t, \boldsymbol{y}^t$ have asymptotically i.i.d. Gaussian entries in the asymptotic regime (Assumption 2), at fixed $t$. The mean and variance of $\boldsymbol{\theta}^t, \boldsymbol{y}^t$ can be computed through a deterministic recursion called *state evolution*. The validity of state evolution has been proved for a broad class of random matrices. Before describing it for our current setting, we establish some notation.

Let $Y$ be a random variable distributed as original labels, namely $\mathbb{P}(Y = 1) = \pi_+$ and $\mathbb{P}(Y = -1) = \pi_-$. Also, let $\widehat{Y}$ be a random variable distributed as noisy labels, namely $\mathbb{P}(\widehat{Y} = Y) = (1 - p)$ and $\mathbb{P}(\widehat{Y} = -Y) = p$. The state evolution involves sequence of deterministic quantities $(m_t, \sigma_t)_{t \geq 0}$ defined by the following recursions:

$$\bar{m}_t = \gamma\sqrt{\alpha}m_t, \quad \bar{\sigma}_t^2 = \alpha(m_t^2 + \sigma_t^2)$$

$$m_{t+1} = \frac{\gamma}{\sqrt{\alpha}} \mathbb{E}\left\{g_t(\bar{m}_t Y + \bar{\sigma}_t G, \widehat{Y})Y\right\}, \quad \sigma_{t+1}^2 = \mathbb{E}\left\{g_t^2(\bar{m}_t Y + \bar{\sigma}_t G, \widehat{Y})\right\}, \quad (8)$$

where $G \sim \mathsf{N}(0, 1)$ is independent of other random variables. The next theorem (proved in Appendix B) implies that the the empirical distribution of $\{(\sqrt{d}\mu_i, \theta_i^t)\}_{i=1}^d$ converges weakly to the probability distribution of $(M, \frac{m_t}{\gamma}M + \sigma_t G)$, where $(m_t, \sigma_t)$ are given by the state evolution sequence. As a consequence, it can be used to characterize the limiting behavior of $P_e(\boldsymbol{\theta})$ (test error of the model) in terms of state evolution sequence.

**Theorem 3.1** *Let $(\boldsymbol{\theta}^t, \boldsymbol{y}^t)_{t\geq 0}$ be the AMP iterates given by (4)-(5). Also let $(m_t, \sigma_t)_{t\geq 0}$ be the state evolution recursions given by (8). Then under Assumption 1, for any pseudo-Lipschitz function $\psi : \mathbb{R}^2 \to \mathbb{R}$ the following holds almost surely for $t \geq 0$:*

$$\lim_{n\to\infty} \left| \frac{1}{d} \sum_{i=1}^{d} \psi(\theta_i^t, \sqrt{d}\mu_i) - \mathbb{E}\left[ \psi\left( \frac{m_t}{\gamma} M + \sigma_t G, M \right) \right] \right| = 0 \,, \tag{9}$$

*where $G \sim \mathsf{N}(0,1)$, $M \sim \nu_M$ (see second bullet point of Assumption 1) are independent. Recall that $g_0(\cdot, \widehat{y}) = \widehat{y}$, which corresponds to the initialization of state evolution recursion with $m_1 = \gamma(1-2p)/\sqrt{\alpha}$ and $\sigma_1 = 1$. In addition, we have almost surely:*

$$\lim_{n\to\infty} P_e(\boldsymbol{\theta}^t) = \Phi\left( -\frac{m_t \gamma}{\sqrt{m_t^2 + \sigma_t^2}} \right) \,. \tag{10}$$

### 3.3 Optimal Choice of Aggregator Functions

In the AMP iterations (4) and (5), $\boldsymbol{\theta}^t$ is the model estimate and $\boldsymbol{y}^t$ is the soft predictions vector at iteration $t$. We will now discuss some examples of the *aggregator* function $g_t$; in particular, these examples describe the retraining methods proposed in [9]:[2]

- **Full retraining [9]:** In this case, $g_t(\boldsymbol{y}^t, \widehat{\boldsymbol{y}}) = \text{sign}(\boldsymbol{y}^t)$ for $t \geq 1$. So in this type of retraining, the noisy labels are not used and only the current model's predicted labels are used.
- **Consensus-based retraining [9]:** Here, $g_t(\boldsymbol{y}^t, \widehat{\boldsymbol{y}}) = \widehat{\boldsymbol{y}} \mathbb{1}(\boldsymbol{y}^t \widehat{\boldsymbol{y}} > 0)$ for $t \geq 1$. Thus, retraining is done only using the samples for which the predicted label matches the noisy label.

Note that the result of Theorem 3.1 does not apply to these examples directly, because the $g_t$'s here are not Lipschitz. However, we can approximate them by Lipschitz functions, see (37) and Appendix F for further details.

In this section, we aim to derive the *optimal choice of aggregator functions*. Note that from Equation (10), the test error of $\boldsymbol{\theta}^t$ is an increasing function of $m_t/\sigma_t$. Suppose that the retraining is run for $t$ iterations, so $\bar{m}_t, \bar{\sigma}_t$ are determined. The optimal aggregator is the one that maximizes $m_t/\sigma_t$. Recalling the state evolution (8), we have

$$\frac{m_{t+1}^2}{\sigma_{t+1}^2} = \frac{\gamma^2}{\alpha} \frac{\mathbb{E}\{g_t(\bar{m}_t Y + \bar{\sigma}_t G, \widehat{Y})Y\}^2}{\mathbb{E}\{g_t^2(\bar{m}_t Y + \bar{\sigma}_t G, \widehat{Y})\}} \leq \frac{\gamma^2}{\alpha} \mathbb{E}\{(2q_t - 1)^2\} \,, \tag{11}$$

with $q_t := \mathbb{P}(Y = 1 | \bar{m}_t Y + \bar{\sigma}_t G, \widehat{Y})$. The above inequality holds because by the law of iterated expectations, we have

$$\mathbb{E}\{g_t(\bar{m}_t Y + \bar{\sigma}_t G, \widehat{Y})Y\} = \mathbb{E}\{g_t(\bar{m}_t Y + \bar{\sigma}_t G, \widehat{Y}) \mathbb{E}\{Y | \bar{m}_t Y + \bar{\sigma}_t G, \widehat{Y}\}\}$$
$$= \mathbb{E}\{g_t(\bar{m}_t Y + \bar{\sigma}_t G, \widehat{Y})(2q_t - 1)\} \,,$$

and so (11) follows from Cauchy-Schwarz inequality. Also, the upper bound is achieved when $g_t$ is (any scaling of) $2q_t - 1$. Therefore, for this optimal choice of $g_t^*$ we have

$$m_{t+1} = \frac{\gamma}{\sqrt{\alpha}} \mathbb{E}\left\{ \left( g_t^*(\bar{m}_t Y + \bar{\sigma}_t G, \widehat{Y}) \right)^2 \right\}, \quad \sigma_{t+1}^2 = \mathbb{E}\left\{ \left( g_t^*(\bar{m}_t Y + \bar{\sigma}_t G, \widehat{Y}) \right)^2 \right\}, \tag{12}$$

and so $m_{t+1} = \frac{\gamma}{\sqrt{\alpha}} \sigma_{t+1}^2$. We let $\eta_t := \frac{m_t}{\sigma_t}$ and so

$$m_t = \frac{\sqrt{\alpha}}{\gamma} \eta_t^2, \quad \sigma_t = \frac{\sqrt{\alpha}}{\gamma} \eta_t, \quad \bar{m}_t = \alpha \eta_t^2, \quad \bar{\sigma}_t^2 = \frac{\alpha^2}{\gamma^2} \eta_t^2 (\eta_t^2 + 1) \,. \tag{13}$$

Therefore, we can write the state evolution and the optimal aggregator, only in terms of $\eta_t$. We formally state it in the next theorem (proved in Appendix C).

---

[2] It is worth mentioning that [9] analyze the simple model in (3) and not our AMP-based model.

**Theorem 3.2** *Recall that $\pi_+, \pi_-$ are the class probabilities and $p$ is the label flipping probability. The optimal aggregator functions for the AMP-based procedure in (4)-(5) is given by:*

$$g_t^*(y, \widehat{y}) = \frac{2}{1 + \left(\frac{p}{1-p}\right)^{\widehat{y}} \exp\left(-\frac{2\gamma^2 y}{\alpha(\eta_t^2+1)}\right)\frac{\pi_-}{\pi_+}} - 1, \tag{14}$$

*for $t \geq 1$ (recall that $g_0^*(\cdot, \widehat{y}) = \widehat{y}$). In addition, we have:*

$$P_e(\boldsymbol{\theta}^t) \xrightarrow{p} \Phi\left(-\frac{\gamma\eta_t}{\sqrt{\eta_t^2+1}}\right),$$

*where $(\eta_t)_{t\geq 1}$ is given by the following state evolution recursion:*

$$\eta_{t+1}^2 = \frac{\gamma^2}{\alpha} \mathbb{E}\left\{\left(g_t^*\left(\alpha\eta_t^2 Y + \frac{\alpha}{\gamma}\eta_t\sqrt{\eta_t^2+1}G, \widehat{Y}\right)\right)^2\right\}, \tag{15}$$

*with initialization $\eta_1 = \gamma(1-2p)/\sqrt{\alpha}$.*

We perform some simulations in Appendix E to verify our theory, and compare the performance of optimal aggregator with the full retraining and the consensus-bases retraining schemes. Note that in the noiseless regime ($p = 0$), the optimal aggregator becomes $g_t^*(y, 1) = 1$ and $g_y^*(y, -1) = -1$ which is expected because in this regime, $\widehat{y}$ is the ground truth label.

**Discussion.** The state evolution sequence $(\eta_t)_{t\geq 0}$ in (15) can be rewritten as $\eta_{t+1}^2 = F(\eta_t^2)$ with

$$F(u) = \frac{\gamma^2}{\alpha} \mathbb{E}\left\{\left(\tilde{g}\left(\gamma^2 \frac{u}{1+u}Y + \gamma\sqrt{\frac{u}{1+u}}G, \widehat{Y}\right)\right)^2\right\}, \quad \tilde{g}(y, \widehat{y}) = \frac{2}{1 + \left(\frac{p}{1-p}\right)^{\widehat{y}}e^{-2y}\frac{\pi_-}{\pi_+}} - 1. \tag{16}$$

In Figure 1, we illustrate an example of this mapping along with the sequence $(\eta_t)_{t\geq 1}$ (Cobweb diagram). As can be observed from the figure (and also formalized in Proposition 3.3), the mapping $F$ is non-decreasing. When $\eta_1$ is small, the sequence $\eta_t$ increases, while if $\eta_1$ is large, the sequence decreases. Since the test error decreases as $\eta$ increases, this leads to an interesting observation: *if the initial model is poor, retraining helps improve its performance, but if the initial model is already good, retraining can actually hurt its performance.* We formalize this observation in the next proposition (proved in Appendix D).

**Proposition 3.3** *The following statements hold:*

*(i) The function $F$ defined in (16) is non-decreasing on $[0, \infty)$. Also, it has at least one fixed point, i.e., there exists a solution to $\eta^2 = F(\eta^2)$. Let $\eta_*^2$ be the smallest fixed point. If $\eta_1 \leq \eta_*$ then the state evolution sequence $(\eta_t)_{t\geq 1}$ is non-decreasing, and hence retraining reduces the test error.*

*(ii) Suppose that $\gamma^2 \geq \sqrt{\frac{\pi\alpha}{2}}$. A sufficient condition for the sequence $(\eta_t)_{t\geq 1}$ to be non-decreasing is that $p \in [p_*, \frac{1}{2})$, with $p_* < \frac{1}{2}$ being the unique solution of the equation: $\Phi\left(\frac{-\gamma^2(1-2p)}{\sqrt{\gamma^2(1-2p)^2+\alpha}}\right) = p$.*

## 4 Generalized Linear Model (GLM)

Consider a data matrix $\boldsymbol{X} \in \mathbb{R}^{n\times d}$, with rows $\boldsymbol{x}_1, \ldots, \boldsymbol{x}_n \sim \mathsf{N}(0, \boldsymbol{I}_d/n)$,[3] and corresponding labels $\boldsymbol{y} = [y_1, \ldots, y_n]^\top \in \mathbb{R}^n$ generated with the following probabilistic rule:

$$\mathbb{P}(y_i = 1|\boldsymbol{x}_i) = h(\boldsymbol{x}_i^\top\boldsymbol{\beta}),$$

for a known link function $h$. However, the learner observes noisy labels $\widehat{\boldsymbol{y}} = [\widehat{y}_1, \ldots, \widehat{y}_n]^\top \in \mathbb{R}^n$, where the $\widehat{y}_i$'s follow the same noise model as (2) with $p$ being the label flipping probability. We also define $\hat{h}_p(z) := (1-p)h(z) + p(1 - h(z))$. Note that $\mathbb{P}(\widehat{y}_i = 1|\boldsymbol{x}_i) = \hat{h}_p(\boldsymbol{x}_i^\top\boldsymbol{\beta})$.

---

[3]The scaling of $1/n$ in the covariance matrix is for ease of exposition. Our analysis can be extended even if the covariance matrix is $\boldsymbol{I}_d$ at the cost of more tedious exposition.

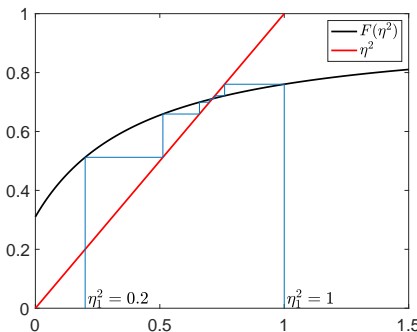

Figure 1: Cobweb plot for the state evolution in Theorem 3.1, with two initializations: (small) $\eta_1 = 0.2$ and (large) $\eta = 1$. Here, $\gamma = 1.5$, $p = 0.3$, $\alpha = 2$, $\pi_+ = 0.3$, $\pi_- = 0.7$.

**AMP-based update rule:** Similar to the AMP-based iterative update rule for GMM in (4)-(5), we progressively update our classifier $\boldsymbol{\beta}^t$ at iteration $t$ with an aggregator function $g_t$ as follows:

$$
\begin{aligned}
\boldsymbol{\beta}^{t+1} &= \boldsymbol{X}^{\mathsf{T}} g_t(\boldsymbol{y}^t, \widehat{\boldsymbol{y}}) - C_t \boldsymbol{\beta}^t, \qquad && (17) \\
\boldsymbol{y}^{t+1} &= \boldsymbol{X}\boldsymbol{\beta}^{t+1} - g_t(\boldsymbol{y}^t, \widehat{\boldsymbol{y}})\frac{d}{n}. \qquad && (18)
\end{aligned}
$$

Similar to the GMM case, the aggregator function $g_t$ is applied entry wise, the Onsager coefficient $C_t \in \mathbb{R}$ is given by

$$
C_t = \frac{1}{n}\sum_{i=1}^{n} \frac{\partial g_t}{\partial y}(y, \widehat{y}_i)\Big|_{y=y_i^t},
$$

and the process is initialized with $g_0(\cdot, \widehat{\boldsymbol{y}}) = \widehat{\boldsymbol{y}}$.

We track the performance of the classifier $\boldsymbol{\beta}^t$ via its test error. For a model $\boldsymbol{\theta}$, its predicted label for a test point $\boldsymbol{x}$ is given by $\mathrm{sign}(\boldsymbol{x}^{\mathsf{T}}\boldsymbol{\theta})$. We first state the assumptions needed for our subsequent results.

**Assumption 2** *We assume the following:*

- *As $n, d \to \infty$, the ratio $d/n \to \alpha \in (0, \infty)$.*

- *The empirical distributions of the entries of $\boldsymbol{\beta}$ converges weakly to a probability distribution $\pi_{\bar{\beta}}$ on $\mathbb{R}$ with bounded second moment. Let $\gamma^2 = \mathbb{E}_{\pi_{\bar{\beta}}}[\bar{\beta}^2]$.*

- *The function $g_t : \mathbb{R} \times \mathbb{R} \mapsto \mathbb{R}$ is Lipschitz continuous.*

Our next lemma (proved in Appendix G) characterizes the test error of a model $\boldsymbol{\theta}$.

**Lemma 4.1** *For a model $\boldsymbol{\theta}$, define its test error $P_e(\boldsymbol{\theta}) := \mathbb{P}(y\boldsymbol{x}^{\mathsf{T}}\boldsymbol{\theta} < 0)$, where $(\boldsymbol{x}, y)$ is generated from the GLM. Define $\rho := \boldsymbol{\beta}^{\mathsf{T}}\boldsymbol{\theta}/(\|\boldsymbol{\beta}\|_{\ell_2}\|\boldsymbol{\theta}\|_{\ell_2})$. Then,*

$$
P_e(\boldsymbol{\theta}) = F(\rho) := \mathbb{E}\left[\Phi\left(\frac{\rho Z}{\sqrt{1-\rho^2}}\right)\Big(1 - h\big(\sqrt{\alpha}\gamma Z\big)\Big) + \Phi\left(\frac{-\rho Z}{\sqrt{1-\rho^2}}\right)h\big(\sqrt{\alpha}\gamma Z\big)\right], \qquad (19)
$$

*with $Z \sim \mathsf{N}(0, 1)$. In addition, $F$ is a decreasing function, if $h(u) > h(-u)$, for all $u > 0$.*

**State evolution.** We will describe state evolution just as we did for the GMM case. The state evolution parameters $(\mu_t, \sigma_t)_{t \geq 1}$ are recursively defined by:

$$
\mu_1 = \frac{2}{\alpha\gamma^2}\,\mathbb{E}[Z\hat{h}_p(Z)], \quad \sigma_1 = \sqrt{\alpha}
$$

$$
\mu_{t+1} = \mathbb{E}\left[\frac{\mathbb{E}(Z|Z_t, \widehat{Y}) - \mathbb{E}(Z|Z_t)}{\mathrm{Var}(Z|Z_t)}g_t(Z_t, \widehat{Y})\right], \quad \sigma_{t+1}^2 = \alpha\,\mathbb{E}\big[g_t^2(Z_t, \widehat{Y})\big], \qquad (20)
$$

where $Z \sim \mathsf{N}(0, \alpha\gamma^2)$ and $Z_t = \mu_t Z + \sigma_t G$, where $G \sim \mathsf{N}(0,1)$ is independent of other random variables. In addition, $\widehat{Y} \in \{-1, +1\}$ with $\mathbb{P}(\widehat{Y} = 1) = \hat{h}_p(Z)$. Using the joint Gaussianity of $(Z_t, Z)$, we can calculate $\mathbb{E}(Z|Z_t)$ and $\mathrm{Var}(Z|Z_t)$ more explicitly, as follows:

$$\mathbb{E}(Z|Z_t) = \frac{\alpha\mu_t\gamma^2}{\sigma_t^2 + \alpha\mu_t^2\gamma^2} Z_t, \qquad \mathrm{Var}(Z|Z_t) = \frac{\alpha\sigma_t^2\gamma^2}{\sigma_t^2 + \alpha\mu_t^2\gamma^2}. \tag{21}$$

Using these identities, we can rewrite the state evolution recursion as:

$$\mu_1 = \frac{2}{\alpha\gamma^2}\,\mathbb{E}\big[Z\hat{h}_p(Z)\big], \quad \sigma_1 = \sqrt{\alpha}, \tag{22}$$

$$\mu_{t+1} = \left(\frac{1}{\alpha\gamma^2} + \frac{\mu_t^2}{\sigma_t^2}\right)\mathbb{E}\big[\mathbb{E}(Z|Z_t, \widehat{Y})g_t(Z_t, \widehat{Y})\big] - \frac{\mu_t}{\sigma_t^2}\,\mathbb{E}\big[Z_t g_t(Z_t, \widehat{Y})\big], \quad \sigma_{t+1}^2 = \alpha\,\mathbb{E}(g_t^2(Z_t, \widehat{Y})).$$

The next theorem establishes that for every fixed $t$, the asymptotic behavior of $\boldsymbol{\beta}^t$ is precisely characterized by the state evolution recursion, which in turn precisely provides the limiting behavior of the test error of $\boldsymbol{\beta}^t$, for every fixed iteration $t$. The proof largely follows from the analysis of generalized AMP (GAMPs) algorithm proposed by [35] along with Stein's lemma to further simplify the state evolution recursion (similar to [27, Proposition 3.1]). Please refer to Appendix H for further details and initialization of the state evolution.

**Theorem 4.2** *Let* $(\boldsymbol{\beta}^t, \boldsymbol{y}^t)_{t\geq0}$ *be the AMP iterates given by* (17)-(18)*. Also let* $(\mu_t, \sigma_t)_{t\geq0}$ *be the state evolution recursions given by* (22)*. Then under Assumption 2, for any pseudo-Lipschitz function* $\psi : \mathbb{R}^2 \to \mathbb{R}$ *the following holds almost surely for* $t \geq 0$*:*

$$\lim_{n\to\infty}\left|\frac{1}{d}\sum_{i=1}^{d}\psi(\beta_i^t, \beta_i) - \mathbb{E}\left[\psi\Big(\mu_t\bar{\beta} + \frac{\sigma_t}{\sqrt{\alpha}}G, \bar{\beta}\Big)\right]\right| = 0, \tag{23}$$

*where* $G \sim \mathsf{N}(0,1)$ *and* $\bar{\beta} \sim \pi_{\bar{\beta}}$ *(see second bullet point of Assumption 2) are independent. Recall that* $g_0(\cdot, \widehat{y}) = \widehat{y}$*, which corresponds to the initialization of state evolution recursion with* $\mu_1 = \frac{2}{\alpha\gamma^2}\,\mathbb{E}[Z\hat{h}_p(Z)]$ *where* $Z \sim \mathsf{N}(0, \alpha\gamma^2)$*, and* $\sigma_1 = \sqrt{\alpha}$*. In addition, we have almost surely,*

$$\lim_{n\to\infty} P_e(\boldsymbol{\beta}^t) = F\left(\frac{\eta_t\gamma}{\sqrt{\eta_t^2\gamma^2 + \frac{1}{\alpha}}}\right), \tag{24}$$

*where* $\eta_t = \mu_t/\sigma_t$ *and* $F(\rho)$ *is given by* (19)*.*

**Optimal choice of aggregator functions** $g_t$**.** Using Equation 24, and since $F$ is a decreasing function, the optimal $g_t$ is the one that, fixing the history of the algorithm, maximizes $\eta_{t+1}$. Invoking the state evolution (20), we have

$$\eta_{t+1}^2 := \frac{\mu_{t+1}^2}{\sigma_{t+1}^2} = \frac{\mathbb{E}\left(\frac{\mathbb{E}(Z|Z_t, \widehat{Y}) - \mathbb{E}(Z|Z_t)}{\mathrm{Var}(Z|Z_t)}g_t(Z_t, \widehat{Y})\right)^2}{\alpha\,\mathbb{E}(g_t^2(Z_t, \widehat{Y}))} \leq \frac{1}{\alpha}\,\mathbb{E}\left(\left[\frac{\mathbb{E}(Z|Z_t, \widehat{Y}) - \mathbb{E}(Z|Z_t)}{\mathrm{Var}(Z|Z_t)}\right]^2\right),$$

using the Cauchy-Schwarz inequality. The optimal aggregator for which the equality happens in the above equation, is when the aggregator is (any deterministic scalar) of

$$g_t^*(Z_t, \widehat{Y}) = \frac{\mathbb{E}(Z|Z_t, \widehat{Y}) - \mathbb{E}(Z|Z_t)}{\mathrm{Var}(Z|Z_t)} = \frac{\mathbb{E}(Z|Z_t, \widehat{Y})}{\frac{\alpha\sigma_t^2\gamma^2}{\sigma_t^2 + \alpha\mu_t^2\gamma^2}} - \frac{\mu_t}{\sigma_t^2}Z_t = \left(\frac{1}{\alpha\gamma^2} + \frac{\mu_t^2}{\sigma_t^2}\right)\mathbb{E}(Z|Z_t, \widehat{Y}) - \frac{\mu_t}{\sigma_t^2}Z_t. \tag{25}$$

We give a more explicit characterization of the optimal aggregator in the next proposition, by writing $\mathbb{E}(Z|Z, \widehat{Y})$ explicitly using the Bayes rule. We also show that the state evolution for the optimal aggregator can be written directly in terms of $\eta_t = \mu_t/\sigma_t$.

**Theorem 4.3** *The optimal aggregator functions in the AMP-based procedure is given by:*

$$g_t^*(y, \widehat{y}) = \left(\frac{1}{\alpha\gamma^2} + \eta_t^2\right)\frac{\int_{-\infty}^{\infty} z e^{-\frac{\eta_t^2 z^2}{2} + \frac{yz}{\alpha}}\left(\hat{h}_p(z)\right)^{\frac{1+\widehat{y}}{2}}\left(1 - \hat{h}_p(z)\right)^{\frac{1-\widehat{y}}{2}} e^{-\frac{z^2}{2\alpha\gamma^2}}\,\mathrm{d}z}{\int_{-\infty}^{\infty} e^{-\frac{\eta_t^2 z^2}{2} + \frac{yz}{\alpha}}\left(\hat{h}_p(z)\right)^{\frac{1+\widehat{y}}{2}}\left(1 - \hat{h}_p(z)\right)^{\frac{1-\widehat{y}}{2}} e^{-\frac{z^2}{2\alpha\gamma^2}}\,\mathrm{d}z} - \frac{y}{\alpha}, \tag{26}$$

*for $t \geq 1$, and $g_0^*(\cdot, \widehat{y}) = \widehat{y}$, where $(\eta_t)_{t \geq 1}$ is given by the following state evolution recursion:*

$$\eta_1 = \frac{2}{\alpha^{3/2}\gamma^2} \mathbb{E}[Z\hat{h}_p(Z)], \quad \eta_{t+1}^2 = \frac{1}{\alpha} \mathbb{E}\left\{\left(g_t^*(\alpha\eta_t^2 Z + \alpha\eta_t G, \widehat{Y})\right)^2\right\}, \tag{27}$$

*where $Z \sim \mathsf{N}(0, \alpha\gamma^2)$ and $G \sim \mathsf{N}(0,1)$ are independent of each other. In addition, $\widehat{Y} \in \{-1, +1\}$ with $\mathbb{P}(\widehat{Y} = 1|Z) = \hat{h}_p(Z)$.*

The proof of Theorem 4.3 is in Appendix I. Similar trends to those described after Theorem 3.2 also hold in the context of GLMs. In Appendix J, we present a detailed discussion of a special case of Theorem 4.3, specifically when the link function $h$ is the sign function, along with additional remarks.

We discuss extension of our theory to the multi-class case and non-linear models in Appendix K.

## 5 Experiments

We show that extensions of the optimal $g_t^*$ derived in Theorem 3.2 is very effective for improving the performance of standard linear probing (i.e., fitting a linear layer on top of a pretrained model) [1, 22] as well as full network training with the cross-entropy loss for binary classification in the presence of label noise. Here we consider two label noise models: *(a)* the uniform noise model in (2) with $p$ being the label flipping probability, and *(b)* a non-uniform noise model where $\mathbb{P}(\widehat{y}_i = -1|y_i = +1) = p$ and $\mathbb{P}(\widehat{y}_i = +1|y_i = -1) = q$ with $p \neq q$ (independently for all $i \in [n]$). We adapt the derivation in (34) (in the proof Theorem 3.2) by considering general means and variances (instead of symmetric means and equal variances as in (34)) obtained by fitting a bimodal GMM on the distribution of the unnormalized logits ($\in (-\infty, \infty)$) of the training set. Suppose the means and variances of the peaks corresponding to the positive and negative logits are $(\mu_+, \sigma_+^2)$ and $(\mu_-, \sigma_-^2)$, respectively. As shown in Appendix M, the aggregators we obtain here for logit $z$ and given label $\widehat{y}$ are the following for

*(a) uniform noise model:* $g(z, \widehat{y}) = \dfrac{2}{1 + \left(\frac{p}{1-p}\right)^{\widehat{y}} \exp\left(\frac{(z-\mu_+)^2}{2\sigma_+^2} - \frac{(z-\mu_-)^2}{2\sigma_-^2}\right)\frac{\pi_-}{\pi_+}} - 1,$ (28)

*(b) non-uniform noise model:* $g(z, \widehat{y}) = \dfrac{2}{1 + \left(\frac{q(1-q)}{p(1-p)}\right)^{1/2} \left(\frac{pq}{(1-p)(1-q)}\right)^{\widehat{y}/2} \exp\left(\frac{(z-\mu_+)^2}{2\sigma_+^2} - \frac{(z-\mu_-)^2}{2\sigma_-^2}\right)\frac{\pi_-}{\pi_+}} - 1,$

with $g(z, \widehat{y})$ being the soft prediction we use for training in the next round. We call our method using the aggregation functions defined in (28) BayesMix RT, where RT is an abbreviation for retraining as used in [9]. Note that these aggregation functions are specific to the noise model and also depend on the noise model's parameters (e.g., $p, q$), which may not be always available and easy to estimate. So we propose the following simpler aggregation function that can be used under more general noise models (e.g., sample-dependent noise) and in scenarios where the noise model is not known:

$$g_{\text{simple}}(z, \widehat{y}) = \frac{2}{1 + \gamma^{\widehat{y}} \exp\left(\frac{(z-\mu_+)^2}{2\sigma_+^2} - \frac{(z-\mu_-)^2}{2\sigma_-^2}\right)\frac{\pi_-}{\pi_+}} - 1, \tag{29}$$

where $\gamma \in (0, 1)$ is a constant which can be tuned (and $g_{\text{simple}}(z, \widehat{y})$ is the soft prediction we use for training in the next round). Essentially, (29) replaces the $\left(\frac{p}{1-p}\right)$ term in (a) of (28) by a (tunable) constant $\gamma$. In our experiments, we fix $\gamma$ to 0.7 throughout without any tuning, thereby ensuring no reliance on the knowledge of the underlying noise model. We call our method using $g_{\text{simple}}(z, \widehat{y})$ BayesMix-Simple RT and also propose to use it for full-network training.[4] We compare BayesMix RT/BayesMix-Simple RT with full RT and consensus-based RT proposed in [9].

For all our experiments, we use the body of a ResNet-50 model pretrained on ImageNet. In linear probing, we train only the linear head attached on top of the (frozen) pretrained body, while in full network training, we train the body as well as the linear head. We consider two datasets available on TensorFlow: **(i)** MedMNIST Pneumonia [42] which is a medical binary classification dataset, and **(ii)** Food 101 [6] which is a multi-class food-based classification dataset, but we use only two similar classes, for e.g., pho vs. ramen (because both are broth-based dishes). All the results are averaged over 3 runs. The experimental details are deferred to Appendix N due to lack of space.

**Linear probing.** We run each method for 10 iterations. In Tables 1 and 2, we list the average test accuracies of full RT, consensus-based RT, and BayesMix RT (28) after 1 and 10 iterations for

---

[4]It is worth clarifying that all our aggregation functions depend on the class prior probabilities $\pi_+$ and $\pi_-$.

MedMNIST Pneumonia corrupted by the uniform noise model with $p = 0.45$ (note that this is a high degree of label noise) and the non-uniform noise model with $p = 0.45$ and $q = 0.2$, respectively. Note that *BayesMix RT is better than full RT and consensus-based RT* after 10 iterations. Further, in Table 6 (Appendix L), we conduct an ablation study on the uniform noise model to compare the three RT methods at different values of $p$. Next, in Table 7 (Appendix L), we compare full RT, consensus-based RT, and BayesMix-Simple RT (29) on MedMNIST Pneumonia corrupted by an *adversarial* noise model described in Appendix L.

**Full network training.** As mentioned earlier, we propose to use BayesMix-Simple RT (29) for full network training. We run full RT, consensus-based RT, and BayesMix-Simple RT for 3 iterations.[5] In Tables 3 and 4, we list the average test accuracies of each method after 1 and 3 iterations for Food-101 pho vs. ramen corrupted by uniform noise with $p = 0.4$ and Food-101 spaghetti bolognese vs. spaghetti carbonara corrupted by non-uniform noise with $p = 0.45$ and $q = 0.2$, respectively. Note that *BayesMix-Simple RT is significantly better than full RT and consensus-based RT* here (in fact, the latter two methods do not yield any gains here).

Table 1: **Linear probing and uniform noise** ($p = 0.45$)**:** Average test accuracies ± standard deviation for MedMNIST Pneumonia. In the first iteration of retraining, consensus-based RT performs the best, *but at the tenth iteration, BayesMix RT performs the best*.

| Iteration # | Full RT | Consensus-based RT | BayesMix RT (ours) |
|---|---|---|---|
| 0 (initial model) | 64.58 ± 3.07 | 64.58 ± 3.07 | 64.58 ± 3.07 |
| 1 | 67.15 ± 4.28 (2.57 ↑) | **68.32 ± 1.88 (3.74 ↑)** | 65.06 ± 2.55 (0.48 ↑) |
| 10 | 68.06 ± 3.78 (3.48 ↑) | 70.03 ± 0.73 (5.45 ↑) | **71.42 ± 2.43 (6.84 ↑)** |

Table 2: **Linear probing and non-uniform noise** ($p = 0.45, q = 0.2$)**:** Average test accuracies ± standard deviation for MedMNIST Pneumonia. *BayesMix RT performs the best*.

| Iteration # | Full RT | Consensus-based RT | BayesMix RT (ours) |
|---|---|---|---|
| 0 (initial model) | 71.79 ± 1.71 | 71.79 ± 1.71 | 71.79 ± 1.71 |
| 1 | 77.62 ± 3.43 (5.83 ↑) | 77.85 ± 2.69 (6.06 ↑) | **82.63 ± 2.02 (10.84 ↑)** |
| 10 | 79.74 ± 3.40 (7.95 ↑) | 82.85 ± 1.44 (11.06 ↑) | **84.39 ± 0.79 (12.60 ↑)** |

Table 3: **Full network training and uniform noise** ($p = 0.4$)**:** Average test accuracies ± standard deviation for Food-101 pho vs. ramen (both are broth-based dishes). *BayesMix-Simple RT is the clear winner*; in fact, the other two RT methods do not lead to any improvements.

| Iteration # | Full RT | Consensus-based RT | BayesMix-Simple RT (ours) |
|---|---|---|---|
| 0 (initial model) | 61.13 ± 3.39 | 61.13 ± 3.39 | 61.13 ± 3.39 |
| 1 | 59.07 ± 5.44 (2.06 ↓) | 59.33 ± 4.83 (1.80 ↓) | **66.73 ± 3.72 (5.60 ↑)** |
| 3 | 57.67 ± 7.47 (3.46 ↓) | 58.47 ± 6.78 (2.66 ↓) | **70.67 ± 4.65 (9.54 ↑)** |

Table 4: **Full network training and non-uniform noise** ($p = 0.45, q = 0.2$)**:** Average test accuracies ± standard deviation for Food-101 spaghetti bolognese vs. spaghetti carbonara (both are spaghetti dishes). Again, *BayesMix-Simple RT is the clear winner*.

| Iteration # | Full RT | Consensus-based RT | BayesMix-Simple RT (ours) |
|---|---|---|---|
| 0 (initial model) | 65.45 ± 2.22 | 65.45 ± 2.22 | 65.45 ± 2.22 |
| 1 | 61.35 ± 7.31 (4.10 ↓) | 61.30 ± 4.48 (4.15 ↓) | **74.55 ± 3.75 (9.10 ↑)** |
| 3 | 56.50 ± 7.74 (8.95 ↓) | 59.70 ± 5.57 (5.75 ↓) | **80.30 ± 5.58 (14.85 ↑)** |

# 6 Conclusion

In this paper, we analyzed optimal model retraining by deriving the Bayes optimal aggregator function to combine the given labels and model predictions for binary classification. Our framework quantified the performance of this strategy over multiple retraining iterations. We also proposed a practical variant and showed that it outperforms existing baselines under different label noise models. In the future, we would like to extend our results to the multi-class classification setting and theoretically analyze more general label noise models.

---

[5]Note that training an entire deep network is much more computationally expensive than linear probing.

# Acknowledgments

AJ was partially supported by the Sloan fellowship in mathematics, the NSF CAREER Award DMS-1844481, the NSF Award DMS-2311024, an Amazon Faculty Research Award, an Adobe Faculty Research Award, and an iORB grant from USC Marshall School of Business.

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

# Appendix

## A Related Work on Approximate Message Passing

Approximate message passing (AMP) refers to a class of iterative algorithms derived through approximation of belief propagation on dense factor graphs [4, 28]. AMP algorithms were first proposed for estimation in linear models [13], and for GLMs [35, 18]. AMP has since been applied to a wide range of high-dimensional statistical estimation problems including Lasso and M-estimators [5], low rank matrix estimation [36, 19, 29], and group synchronization [33], among others. AMP algorithms come with many appealing properties. They can easily be tailored to take advantage of prior information on the structure of the signal, such as sparsity or other constraints. In addition, under suitable assumptions on the data matrix, AMP theory provides precise asymptotic characterization of the behavior of the estimator (despite the randomness of data) in the high dimensional regime where the ratio of the number of observations to dimensions converges to a constant. Even more, for a wide class of estimation problems, AMP is conjectured to be optimal among all polynomial-time algorithms (see e.g. [12, 3, 29]). We refer to [14] for a survey on AMP algorithms. In this work, we use the machinery of AMP algorithms to rigorously understand the effect of retraining a model on its predicted labels over multiple rounds.

## B Proof of Theorem 3.1

To build some intuition on the statement of the theorem and the choice of memory correction terms, we analyze the distribution of the first few estimate of the AMP updates in (4)-(5).

Since $g_0(\cdot, \widehat{y}) = \widehat{y}$, we have $C_0 = 0$ and $\boldsymbol{\theta}^1 = \frac{1}{\sqrt{n}} \boldsymbol{X}^\top \widehat{\boldsymbol{y}}$. Under the GMM, we have $\boldsymbol{X} = \boldsymbol{y}\boldsymbol{\mu}^\top + \boldsymbol{Z}$ with $Z_{ij} \sim \mathsf{N}(0,1)$, independently. Fixing $\boldsymbol{y}$, we have that $\widehat{\boldsymbol{y}}$ and $\boldsymbol{Z}$ are independent. We write

$$
\begin{aligned}
\boldsymbol{\theta}^1 &= \frac{1}{\sqrt{n}} \boldsymbol{X}^\top \widehat{\boldsymbol{y}} \\
&= \frac{(\boldsymbol{\mu}\boldsymbol{y}^\top + \boldsymbol{Z}^\top)\widehat{\boldsymbol{y}}}{\sqrt{n}} \\
&= \sqrt{\frac{n}{d}} \sqrt{d}\boldsymbol{\mu} \frac{\boldsymbol{y}^\top \widehat{\boldsymbol{y}}}{n} + \frac{\boldsymbol{Z}^\top \widehat{\boldsymbol{y}}}{\sqrt{n}} \,.
\end{aligned} \tag{30}
$$

By law of large number, $\frac{\boldsymbol{y}^\top \widehat{\boldsymbol{y}}}{n} \xrightarrow{p} (1 - 2p)$. Also under Assumption 1, $\frac{n}{d} \to \frac{1}{\alpha}$ and the empirical distribution of $\sqrt{d}\boldsymbol{\mu}$ converges weakly to distribution of $M \sim \nu_M$. Therefore the first term in the decomposition (30) converges to $(1 - 2p)\frac{M}{\sqrt{\alpha}}$. Given that $\widehat{\boldsymbol{y}}$ and $\boldsymbol{Z}$ are independent, thee second term is distributed as $\mathsf{N}(0,1)$. This implies that the empirical distribution of $\boldsymbol{\theta}^1$ converges weakly to $\frac{M}{\sqrt{\alpha}}(1 - 2p) + G$, with $G \sim \mathsf{N}(0,1)$. This implies the claim of the theorem for $t = 1$, with $m_1 = \gamma(1 - 2p)/\sqrt{\alpha}$ and $\sigma_1 = 1$.

We next characterize the distribution of $\widehat{\boldsymbol{y}}^1 = \frac{1}{\sqrt{n}} \boldsymbol{X}^\top \boldsymbol{\theta}^1 - \frac{d}{n} \widehat{\boldsymbol{y}}$, which also sheds light on the choice of correction term $-\frac{d}{n}\widehat{\boldsymbol{y}}$. We have

$$
\begin{aligned}
\boldsymbol{y}^1 &= \frac{1}{n} \boldsymbol{X}\boldsymbol{X}^\top \widehat{\boldsymbol{y}} - \frac{d}{n} \widehat{\boldsymbol{y}} \\
&= \frac{1}{n} (\boldsymbol{y}\boldsymbol{\mu}^\top + \boldsymbol{Z})(\boldsymbol{\mu}\boldsymbol{y}^\top + \boldsymbol{Z}^\top)\widehat{\boldsymbol{y}} - \frac{d}{n}\widehat{\boldsymbol{y}} \\
&= \frac{1}{n} (\boldsymbol{y}\boldsymbol{y}^\top \|\boldsymbol{\mu}\|_{\ell_2}^2 + \boldsymbol{Z}\boldsymbol{\mu}\boldsymbol{y}^\top + \boldsymbol{y}\boldsymbol{\mu}^\top \boldsymbol{Z}^\top + \boldsymbol{Z}\boldsymbol{Z}^\top)\widehat{\boldsymbol{y}} - \frac{d}{n}\widehat{\boldsymbol{y}} \\
&= \left( \|\boldsymbol{\mu}\|_{\ell_2}^2 \frac{\boldsymbol{y}^\top \widehat{\boldsymbol{y}}}{n} + \frac{1}{n}\boldsymbol{\mu}^\top \boldsymbol{Z}^\top \widehat{\boldsymbol{y}} \right) \boldsymbol{y} + \frac{\boldsymbol{Z}\boldsymbol{Z}^\top}{n} \widehat{\boldsymbol{y}} - \frac{d}{n}\widehat{\boldsymbol{y}} + \boldsymbol{Z}\boldsymbol{\mu} \frac{\boldsymbol{y}^\top \widehat{\boldsymbol{y}}}{n}
\end{aligned} \tag{31}
$$

By law of large number and recalling Assumption 1, we have $\|\boldsymbol{\mu}\|_{\ell_2}^2 \frac{\boldsymbol{y}^\top \widehat{\boldsymbol{y}}}{n} \xrightarrow{p} (1-2p)\gamma^2$. Also given that $\boldsymbol{Z}$ and $\widehat{\boldsymbol{y}}$ are independent, $\frac{\boldsymbol{Z}^\top \widehat{\boldsymbol{y}}}{\sqrt{n}} \sim \mathsf{N}(0, \boldsymbol{I}_n)$. Since $\|\boldsymbol{\mu}\|_{\ell_2}^2 \to \gamma$, which is of order one, we have $\frac{1}{n}\boldsymbol{\mu}^\top \boldsymbol{Z}^\top \widehat{\boldsymbol{y}}$ converges to zero.

The remaining terms in (31) can be written as $\boldsymbol{\Delta} := \left( \frac{\boldsymbol{Z}\boldsymbol{Z}^\top}{n} - \frac{d}{n}\boldsymbol{I}_n + \frac{\boldsymbol{Z}\boldsymbol{\mu}\boldsymbol{y}^\top}{n} \right)\widehat{\boldsymbol{y}}$. Note that the term $\mathbb{E}(\boldsymbol{Z}\boldsymbol{Z}^\top/n) = (d/n)\boldsymbol{I}_n$ and so $\boldsymbol{\Delta}$ is zero mean. (This also justifies the choice of correction term $-\frac{d}{n}\widehat{\boldsymbol{y}}$.) In addition, by virtue of the central limit theorem, each entry $\Delta_i$ is approximately normal with variance $\alpha^2 + (1-2p)^2\gamma^2$. This implies that the empirical distribution of entries $y^1$, converges to $\bar{m}_1 Y + \bar{\sigma}_1 G$, with

$$\bar{m}_1 = \gamma^2(1-2p), \quad \bar{\sigma}_1^2 = \alpha + \gamma^2(1-2p)^2 \,.$$

The proof of Theorem 3.1 follows by adapting techniques from standard AMP analysis (see e.g., [18, 29, 14]), and so omitted here. A major step in the proof to characterize the conditional distribution of $\boldsymbol{\theta}^t, \boldsymbol{y}^t$ given the previous iterates $(\boldsymbol{\theta}^\tau, \boldsymbol{y}^\tau)_{\tau<t}$, and then show that the 'non-Gaussian components' thereof are asymptotically canceled out by the memory correction terms. A technical challenge though is that in the AMP theory the initialization should be independent from the random matrix $\boldsymbol{X}$. Here the initial estimator is $\boldsymbol{\theta}^1 = \boldsymbol{X}^\top \widehat{\boldsymbol{y}}/\sqrt{n}$ which depends on $\boldsymbol{X}$. The trick is to consider the AMP updates, starting from $t = 0$ with $\boldsymbol{\theta}^0 = \boldsymbol{0}$ and $g_{-1}(\cdot, \cdot) = 0$. This way, the initialization is independent of $\boldsymbol{X}$ and by update rule (4)-(5) we have $\boldsymbol{y}^0 = \boldsymbol{0}$ and with $g_1(y, \widehat{y}) = \widehat{y}$ we get $\boldsymbol{\theta}^1 = \boldsymbol{X}^\top \widehat{\boldsymbol{y}}/\sqrt{n}$ so we will be on the trajectory of the updates.

To derive the limit of $P_e(\boldsymbol{\theta})$, we first note that the functions $\psi(x, y) = xy$ and $\psi(x, y) = x^2$ are pseudo-Lipschitz (of order 2). Therefore, using (9), the following limits hold almost surely,

$$\lim_{n\to\infty} \frac{1}{d}\|\boldsymbol{\theta}\|_{\ell_2}^2 = \lim_{n\to\infty} \frac{1}{d}\sum_{i=1}^d |(\theta_i^t)^2| = \mathbb{E}\left[ \left( \frac{m_t}{\gamma}M + \sigma_t G \right)^2 \right] = m_t^2 + \sigma_t^2 \,,$$

$$\lim_{n\to\infty} \frac{1}{\sqrt{d}}\boldsymbol{\mu}^\top \boldsymbol{\theta}^t = \lim_{n\to\infty} \frac{1}{d}\sum_{i=1}^d (\sqrt{d}\mu_i \theta_i^t) = \mathbb{E}\left[ \left( \frac{m_t}{\gamma}M + \sigma_t G \right)M \right] = \frac{m_t}{\gamma}\mathbb{E}[M^2] = m_t\gamma \,,$$

where we used the identity $\mathbb{E}[M^2] = \gamma^2$ (second bullet point in Assumption 1). Hence, by using (7), we obtain

$$\lim_{n\to\infty} P_e(\boldsymbol{\theta}) = \lim_{n\to\infty} \Phi\left( -\frac{\boldsymbol{\mu}^\top \boldsymbol{\theta}^t}{\|\boldsymbol{\theta}\|_{\ell_2}} \right) = \Phi\left( -\lim_{n\to\infty} \frac{\boldsymbol{\mu}^\top \boldsymbol{\theta}^t}{\|\boldsymbol{\theta}\|_{\ell_2}} \right) = \Phi\left( -\frac{m_t\gamma}{\sqrt{m_t^2 + \sigma_t^2}} \right) \,,$$

where the second equality is by continuity of $\Phi$.

## C  Proof of Theorem 3.2

Our derivation prior to the statement of Theorem 3.2 showed that the optimal aggregator $g_t^*$ is given by any scaling of $2q_t - 1$; we will set the scaling to 1 so that the output of $g_t^* \in [-1, 1]$. By Bayes' rule, we have

$$q_t(y, \widehat{y}) := \mathbb{P}(Y = 1 | \bar{m}_t Y + \bar{\sigma}_t G = y, \widehat{Y} = \widehat{y})$$

$$= \frac{\mathbb{P}\left( G = \frac{y - \bar{m}_t}{\bar{\sigma}_t}, \widehat{Y} = \widehat{y} | Y = 1 \right)\mathbb{P}(Y = 1)}{\mathbb{P}\left( G = \frac{y - \bar{m}_t Y}{\bar{\sigma}_t}, \widehat{Y} = \widehat{y} \right)}$$

$$= \frac{\mathbb{P}\left( G = \frac{y - \bar{m}_t}{\bar{\sigma}_t}, \widehat{Y} = \widehat{y} | Y = 1 \right)\mathbb{P}(Y = 1)}{\mathbb{P}\left( G = \frac{y - \bar{m}_t}{\bar{\sigma}_t}, \widehat{Y} = \widehat{y} \Big| Y = 1 \right)\mathbb{P}(Y = 1) + \mathbb{P}\left( G = \frac{y + \bar{m}_t}{\bar{\sigma}_t}, \widehat{Y} = \widehat{y} \Big| Y = -1 \right)\mathbb{P}(Y = -1)}$$

$$= \frac{\mathbb{P}\left( G = \frac{y - \bar{m}_t}{\bar{\sigma}_t} \right)\mathbb{P}\left( \widehat{Y} = \widehat{y} | Y = 1 \right)\mathbb{P}(Y = 1)}{\mathbb{P}\left( G = \frac{y - \bar{m}_t}{\bar{\sigma}_t} \right)\mathbb{P}\left( \widehat{Y} = \widehat{y} \Big| Y = 1 \right)\mathbb{P}(Y = 1) + \mathbb{P}\left( G = \frac{y + \bar{m}_t}{\bar{\sigma}_t} \right)\mathbb{P}\left( \widehat{Y} = \widehat{y} \Big| Y = -1 \right)\mathbb{P}(Y = -1)} \,.$$
(32)

Under our noise model, we have

$$\mathbb{P}\left( \widehat{Y} = \widehat{y} | Y = 1 \right) = (1-p)^{\frac{1+\widehat{y}}{2}} p^{\frac{1-\widehat{y}}{2}} \quad \text{and} \quad \mathbb{P}\left( \widehat{Y} = \widehat{y} | Y = -1 \right) = (1-p)^{\frac{1-\widehat{y}}{2}} p^{\frac{1+\widehat{y}}{2}} \,. \tag{33}$$

Also, $G \sim \mathsf{N}(0,1)$ and $\mathbb{P}(Y = 1) = \pi_+$ and $\mathbb{P}(Y = -1) = \pi_-$. By substituting in (32) we get

$$
\begin{aligned}
q_t(y, \widehat{y}) &= \frac{(1-p)^{\frac{1+\widehat{y}}{2}} p^{\frac{1-\widehat{y}}{2}} e^{-\frac{(y-\bar{m}_t)^2}{2\bar{\sigma}_t^2}} \pi_+}{(1-p)^{\frac{1+\widehat{y}}{2}} p^{\frac{1-\widehat{y}}{2}} e^{-\frac{(y-\bar{m}_t)^2}{2\bar{\sigma}_t^2}} \pi_+ + (1-p)^{\frac{1-\widehat{y}}{2}} p^{\frac{1+\widehat{y}}{2}} e^{-\frac{(y+\bar{m}_t)^2}{2\bar{\sigma}_t^2}} \pi_-} \\
&= \frac{1}{1 + (\frac{p}{1-p})^{\widehat{y}} e^{-2y\frac{\bar{m}_t}{\bar{\sigma}_t^2}} \frac{\pi_-}{\pi_+}} \, .
\end{aligned}
\tag{34}
$$

By invoking (13), we have

$$
\frac{\bar{m}_t}{\bar{\sigma}_t^2} = \frac{\alpha \eta_t^2}{\frac{\alpha^2}{\gamma^2} \eta_t^2 (\eta_t^2 + 1)} = \frac{\gamma^2}{\alpha(\eta_t^2 + 1)} \, ,
$$

which by plugging into (32) gives the desired result.

Next by Equation (10) we have

$$
P_e(\boldsymbol{\theta}^t) \xrightarrow{p} \Phi\left(-\frac{\gamma \eta_t}{\sqrt{\eta_t^2 + 1}}\right).
$$

For the sequence $(\eta_t)_{t \geq 1}$, note that by (12), we have

$$
\eta_{t+1}^2 := \frac{m_{t+1}^2}{\sigma_{t+1}^2} = \frac{\gamma^2}{\alpha} \mathbb{E}\{g_t^*(\bar{m}_t Y + \bar{\sigma}_t G, \widehat{Y})^2\} \, ,
$$

which gives the result after substituting for $\bar{m}_t$ and $\bar{\sigma}_t$ from (13). Also from Theorem 3.1, we have $\eta_1 = m_1/\sigma_1 = \gamma(1 - 2p)/\sqrt{\alpha}$. This concludes the proof of theorem.

## D    Proof of Proposition 3.3

We start by proving the monotonicity of $F$ in $(i)$. Since $\bar{\eta} = \gamma^2 \frac{u}{1+u}$ is increasing in $u$ it suffices to show that $\mathbb{E}\{\tilde{g}(\bar{\eta} Y + \sqrt{\bar{\eta}} \widehat{Y})^2\}$ is non-decreasing in $\bar{\eta}$. Recall that $\tilde{g}$ is the Bayes optimal estimator and can also be characterized as $\tilde{g}(\tilde{Y}, \widehat{Y}) = \mathbb{E}[Y|\tilde{Y} = \bar{\eta} Y + \sqrt{\bar{\eta}} G, \widehat{Y}]$ (see derivation 32). Hence, $\mathbb{E}[Y \tilde{g}(\tilde{Y}, \widehat{Y})] = \mathbb{E}[\tilde{g}(\tilde{Y}, \widehat{Y})^2]$ and we can write

$$
\mathcal{R}(\bar{\eta}) := \mathbb{E}[(Y - \tilde{g}(\tilde{Y}, \widehat{Y}))^2] = 1 - \mathbb{E}[\tilde{g}(\tilde{Y}, \widehat{Y}))^2].
$$

So we need to show that $\mathcal{R}(\bar{\eta})$ is decreasing in $\bar{\eta}$.

Next note that by Bayes-optimality of $\tilde{g}$ we have $\mathcal{R}(\bar{\eta}) = \inf_g \mathbb{E}[(Y - g(\tilde{Y}, \widehat{Y}))^2]$ where the infimum is with respect to all measurable functions $g$, and $\tilde{Y} = \bar{\eta} Y + \sqrt{\bar{\eta}} G$. Now take $\bar{\eta}_1 < \bar{\eta}_2$. The following holds in distribution:

$$
\bar{\eta}_1 Y + \sqrt{\bar{\eta}_1} G \overset{d}{=} \frac{\bar{\eta}_1}{\bar{\eta}_2}(\bar{\eta}_2 Y + \sqrt{\bar{\eta}_2} G) + \sqrt{\bar{\eta}_1 - \frac{\bar{\eta}_1^2}{\bar{\eta}_2}} Z \, ,
$$

where $G, Z \sim \mathsf{N}(0,1)$ independent of each other. We then write

$$
\begin{aligned}
\mathcal{R}(\bar{\eta}_1) &= \mathbb{E}[(Y - \tilde{g}(\bar{\eta}_1 Y + \sqrt{\bar{\eta}_1} G, \widehat{Y}))^2] \\
&= \mathbb{E}\left[\mathbb{E}\left[\left(Y - \tilde{g}\left(\frac{\bar{\eta}_1}{\bar{\eta}_2}(\bar{\eta}_2 Y + \sqrt{\bar{\eta}_2} G) + \sqrt{\bar{\eta}_1 - \frac{\bar{\eta}_1^2}{\bar{\eta}_2}} z, \widehat{Y}\right)\right)^2\right]\right],
\end{aligned}
\tag{35}
$$

where the inner expectation is conditional on $Z = z$ and the outer expectation is with respect to $Z$. For a fixed $z$, define the function

$$
h_z(y, \widehat{y}) = \tilde{g}\left(\frac{\bar{\eta}_1}{\bar{\eta}_2} y + \sqrt{\bar{\eta}_1 - \frac{\bar{\eta}_1^2}{\bar{\eta}_2}} z, \widehat{y}\right).
$$

Continuing from (35) we get

$$R(\bar{\eta}_1) = \mathbb{E}\left[\, \mathbb{E}\left[\left(Y - h_z\left(\bar{\eta}_2 Y + \sqrt{\bar{\eta}_2}G, \widehat{Y}\right)\right)^2\right]\right]$$

$$\geq \inf_g \mathbb{E}\left[\left(Y - g\left(\bar{\eta}_2 Y + \sqrt{\bar{\eta}_2}G, \widehat{Y}\right)\right)^2\right]$$

$$= \mathcal{R}(\bar{\eta}_2),$$

where the inequality holds since $h_z$ is measurable function. This completes the proof of the non-decreasing nature of $F$.

To prove the claim about fixed points in $(i)$, consider the function $H(u) := u - F(u)$. We have $F(0) = \frac{\gamma^2}{\alpha}\mathbb{E}[\tilde{g}(0, \widehat{Y})^2] > 0$ and so $H(0) < 0$. Also, $\lim_{u \to \infty} F(u) = \frac{\gamma^2}{\alpha}\mathbb{E}[\tilde{g}(\gamma^2 Y + \gamma, \widehat{Y})^2] < \frac{\gamma^2}{\alpha}$. Hence, $\lim_{u \to \infty} H(u) = \infty$ and by the intermediate value theorem, $H(u)$ has at least a zero which corresponds to a fixed point of $F$.

Let $\eta_*^2$ be the smallest fixed point of $F$ (and so $H(\eta_*^2) = 0$). Then, for any $\eta < \eta_*$ we have $H(\eta^2) < 0$ and so $\eta^2 < F(\eta^2)$. This implies that if $\eta_1 < \eta_*$, then $\eta_1^2 < F(\eta_1^2) = \eta_2^2$. Further, by monotonicity of $F$, per item (i), if $\eta_t^2 \leq \eta_{t+1}^2$, then $\eta_{t+1}^2 = F(\eta_t^2) \leq F(\eta_{t+1}^2) = \eta_{t+2}^2$, which proves that the sequence $(\eta_t)_{t \geq 1}$ is monotone non-decreasing. Hence, the first step strictly reduces the test error and the next rounds of retraining do not increase the test error.

We next proceed with $(ii)$. By the argument in $(i)$, if $\eta_1^2 < F(\eta_1^2)$ then the sequence $(\eta_t)_{t \geq 1}$ will be non-decreasing. To this end, we derive a lower bound on $F$, such that $\tilde{F}(u) < F(u)$, $\forall u \geq 0$, and establish condition on the label flipping probability $p$, so that $\eta_1^2 \leq \tilde{F}(\eta_1^2)$ with $\eta_1 = \frac{\gamma}{\sqrt{\alpha}}(1 - 2p)$.

To construct $\tilde{F}$, recall that $\tilde{g}$ is the Bayes-optimal aggregator given by $\tilde{g}(\tilde{Y}, \widehat{Y}) = \mathbb{E}[Y | \tilde{Y} = \bar{\eta}Y + \sqrt{\bar{\eta}}G, \widehat{Y}]$. Using the Cauchy–Schwarz inequality for all $g$,

$$\mathbb{E}[\tilde{g}(\tilde{Y}, \widehat{Y})^2] \geq \frac{\mathbb{E}[\mathbb{E}[Y|\tilde{Y}, \widehat{Y}]g(\tilde{Y}, \widehat{Y})]^2}{\mathbb{E}[g(\tilde{Y}, \widehat{Y})^2]} = \frac{\mathbb{E}[Yg(\tilde{Y}, \widehat{Y})]^2}{\mathbb{E}[g(\tilde{Y}, \widehat{Y})^2]}$$

with $\tilde{g}$ being the function for which the equality occurs. By choosing, $g(\tilde{Y}, \widehat{Y}) = \text{sign}(\tilde{Y})$, we obtain the following lower bound:

$$\mathbb{E}[\tilde{g}(\bar{\eta}Y + \sqrt{\bar{\eta}}G, \widehat{Y})^2] \geq \mathbb{E}[Y\text{sign}(\bar{\eta}Y + \sqrt{\bar{\eta}}G)]^2 = \mathbb{E}[\text{sign}(\bar{\eta} + \sqrt{\bar{\eta}}G)]^2,$$

since $Y \in \{-1, +1\}$ is independent of $G \sim \mathsf{N}(0, 1)$. This gives the following lower bound on $F(u)$,

$$F(u) := \frac{\gamma^2}{\alpha}\mathbb{E}\left\{\tilde{g}\left(\gamma^2\frac{u}{1+u}Y + \gamma\sqrt{\frac{u}{1+u}}G, \widehat{Y}\right)^2\right\}$$

$$\geq \frac{\gamma^2}{\alpha}\mathbb{E}\left\{\text{sign}\left(\gamma^2\frac{u}{1+u} + \gamma\sqrt{\frac{u}{1+u}}G\right)\right\}^2 := \tilde{F}(u).$$

We further simplify $\tilde{F}(u)$ as

$$\tilde{F}(u) = \frac{\gamma^2}{\alpha}\left(1 - 2\mathbb{P}\left(G < -\gamma\sqrt{\frac{u}{1+u}}\right)\right)^2 = \frac{\gamma^2}{\alpha}\left(1 - 2\Phi\left(-\gamma\sqrt{\frac{u}{1+u}}\right)\right)^2.$$

For $\eta_1 = \frac{\gamma}{\sqrt{\alpha}}(1 - 2p)$, the condition $\eta_1^2 \leq \tilde{F}(\eta_1^2)$ is equivalent to

$$p \geq \Phi\left(-\gamma\sqrt{\frac{\eta_1^2}{1+\eta^2}}\right) = \Phi\left(-\frac{\gamma^2(1-2p)}{\sqrt{\alpha + \gamma^2(1-2p)^2}}\right) \tag{36}$$

We next show that there is a unique $p_* \in (0, 1/2)$ for which the above inequality becomes equality. Define the function $h(p) = \Phi\left(-\frac{\gamma^2(1-2p)}{\sqrt{\alpha+\gamma^2(1-2p)^2}}\right) - p$. The first two derivatives are given by

$$h'(p) = \phi\left(-\frac{\gamma^2(1-2p)}{\sqrt{\alpha+\gamma^2(1-2p)^2}}\right)\frac{2\gamma^2\alpha}{(\alpha+\gamma^2(1-2p)^2)^{3/2}} - 1$$

$$h''(p) = \phi\left(-\frac{\gamma^2(1-2p)}{\sqrt{\alpha+\gamma^2(1-2p)^2}}\right)\frac{2\gamma^4\alpha(1-2p)}{(\alpha+\gamma^2(1-2p)^2)^2} + \phi\left(-\frac{\gamma^2(1-2p)}{\sqrt{\alpha+\gamma^2(1-2p)^2}}\right)\frac{12\gamma^4\alpha(1-2p)}{(\alpha+\gamma^2(1-2p)^2)^{5/2}}$$

$$= \phi\left(-\frac{\gamma^2(1-2p)}{\sqrt{\alpha+\gamma^2(1-2p)^2}}\right)\frac{2\gamma^4\alpha(1-2p)}{(\alpha+\gamma^2(1-2p)^2)^2}\left[1 + \frac{6}{\sqrt{\alpha+\gamma^2(1-2p)^2}}\right]$$

We have $h'(\frac{1}{2}) = \frac{2\gamma^2}{\sqrt{\alpha}}\phi(0) - 1 = \sqrt{\frac{2}{\pi}}\frac{\gamma^2}{\sqrt{\alpha}} - 1 > 0$. Also $h(\frac{1}{2}) = 0$, so $h$ should be negative in neighborhood before $1/2$. Since $h(0) > 0$, there should be a root $p_*$ for $h$ in $(0, \frac{1}{2})$. Therefore, $h$ has at least two zeros, $p_*$ and $\frac{1}{2}$. Also note that $h$ is a convex function because $h''(p) > 0$, and therefore, these are the only two zeros of $h$. These properties give a clear picture of the function $h$: it will be positive on $[0, p_*)$, negative on $(p_*, \frac{1}{2})$ and positive afterward. Hence, condition (36), i.e., $h(p) \le 0$ holds if $p \in [p_*, \frac{1}{2})$, which completes the proof.

# E    Simulations to Verify the Theory in Section 3

In Figure 2, we compare the performance of different retraining methods on synthetic data, generated from a GMM model. The mean vector $\boldsymbol{\mu}$ is generated by drawing its entries from $\mathsf{N}(0, 1)$ independently, and then normalize it to have $\|\boldsymbol{\mu}\|_{\ell_2} = \gamma$. The two plots in Figure 2 correspond to different values of label noise $p$ and $\gamma$. In both plots, we set the sample size to $n = 1000$, $d = n\alpha = 800$ and the class probabilities to $\pi_+ = 0.3$ and $\pi_- = 0.7$. The results are averages over 10 realizations of the setting.

The Opt-AMP is the algorithm based on AMP updates (4)-(5), with the optimal aggregator function (14). Vanilla is the linear classifier (3) without retraining, so its performance does not vary by iteration, and its represented by a flat line. SE is the theoretical curve based on the state evolution recursion, FT and CT respectively denote the full-retraining and the consensus-based retraining *without* the memory correction terms. (Note that the memory corrections are not well-defined in these cases since the aggregators are not Lipschitz and not differentiable everywhere.) From these plots, we see that there is a great match between SE predictions and the simulated data point (Opt-AMP). We also observe the superiority of Opt-AMP over other retraining methods as well as the vanilla estimator.

In Figures 3 and 4, we compare Opt-AMP with some 'approximate' versions of full-retraining and consensus-based retraining. Recall that the aggregator functions in these cases are given by $g_t(y, \widehat{y}) = \mathrm{sign}(y)$ and $g_t(y, \widehat{y}) = \widehat{y}\mathbb{1}(y\widehat{y} > 0)$. Since the aggregators are not Lipschitz, the result of Theorem 3.1 (validity of state evolution) does not apply. Even the correction terms are not well-defined. We instead approximate these aggregators as follows:

$$g_t^{\mathsf{FT}}(y, \widehat{y}) \approx \frac{2}{1+e^{-\beta y}} - 1, \quad g_t^{\mathsf{CT}}(y, \widehat{y}) \approx \frac{\widehat{y}}{1+e^{-\beta y\widehat{y}}}. \tag{37}$$

For fixed $\beta > 0$ these functions are Lipschitz and hence the state evolution will predict the limiting behavior. As $\beta$ grows these approximations become tighter. We refer to Appendix F for derivation of curves as $\beta \to \infty$ and further comparison between full retraining and consensus-based retraining.

As the state evolution curves indicate, Opt-AMP outperforms the other rules significantly. Also, as we observe in Figure 3a, for the chosen parameters, the error of full retraining increases across iteration, indicating that sometime the retraining may hurt the performance.

# F    Comparison of Full Retraining and Consensus-based Retraining

As discussed after Theorem 3.2, the AMP theory requires the aggregator function to be Lipschitz. For both, the full-retraining ($g_t(y, \widehat{y}) = \mathrm{sign}(y)$) and the consensus-based retraining ($g_t(y, \widehat{y}) =$

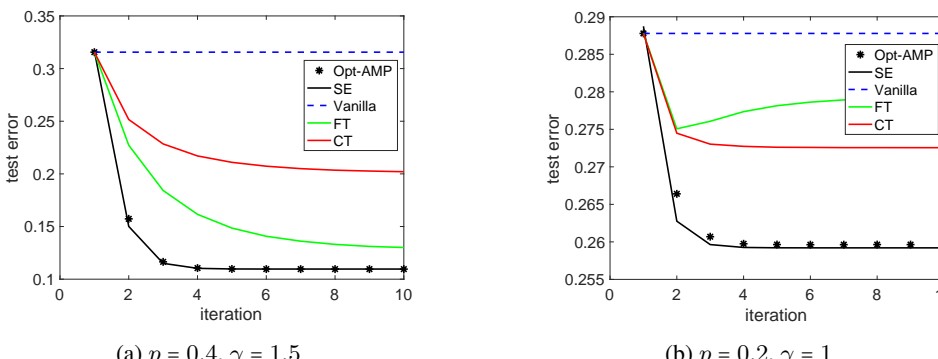

(a) $p = 0.4, \gamma = 1.5$
(b) $p = 0.2, \gamma = 1$

Figure 2: **Synthetic Experiments:** Comparison between different retraining methods. FT and CT respectively denote the full-retraining and the consensus-based retraining *without* the memory correction terms. Vanilla is the estimator without any retraining. Here $n = 1000$, $d = 800$, $\pi_+ = 0.3$, $\pi_- = 0.7$. Dots are the Opt-AMP algorithm and the solid black curve is the state evolution.

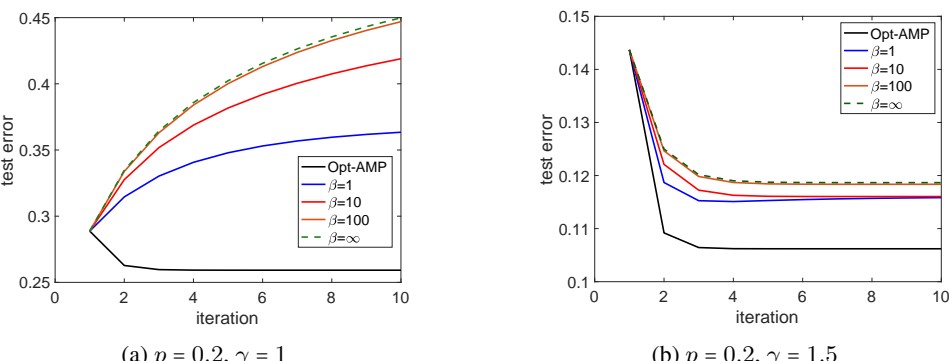

(a) $p = 0.2, \gamma = 1$
(b) $p = 0.2, \gamma = 1.5$

Figure 3: State evolution curves for Opt-AMP and the 'approximate' **full retraining** with the memory correction terms. As $\beta$ grows the approximation of full retraining becomes tighter. Here $\alpha = 0.8$, $\pi_+ = 0.3$, $\pi_- = 0.7$.

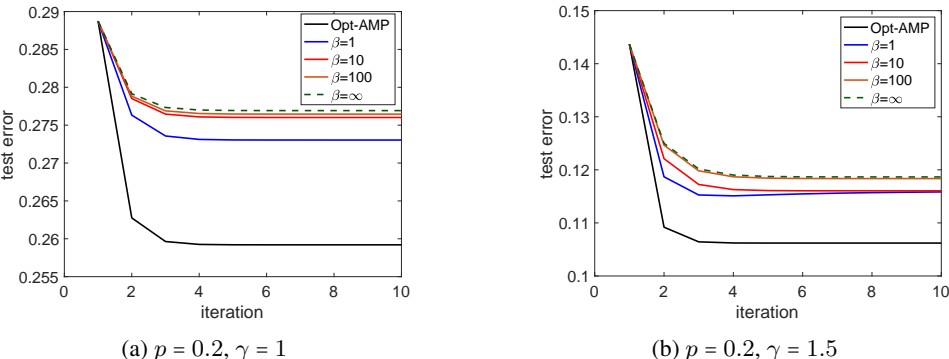

(a) $p = 0.2, \gamma = 1$
(b) $p = 0.2, \gamma = 1.5$

Figure 4: State evolution curves for Opt-AMP and the 'approximate' **consensus-based retraining** with the memory correction terms. As $\beta$ grows the approximation of full retraining becomes tighter. Here $\alpha = 0.8$, $\pi_+ = 0.3$, $\pi_- = 0.7$.

$\widehat{y}\mathbb{1}(y\widehat{y} > 0))$ the aggregator functions violate this assumption. As we discussed for Figures 3 and 4, we approximate the aggregator functions by Lipschitz functions given below:

$$g_t^{\mathsf{FT}}(y, \widehat{y}) \approx \frac{2}{1 + e^{-\beta y}} - 1, \quad g_t^{\mathsf{CT}}(y, \widehat{y}) \approx \frac{\widehat{y}}{1 + e^{-\beta y \widehat{y}}}. \tag{38}$$

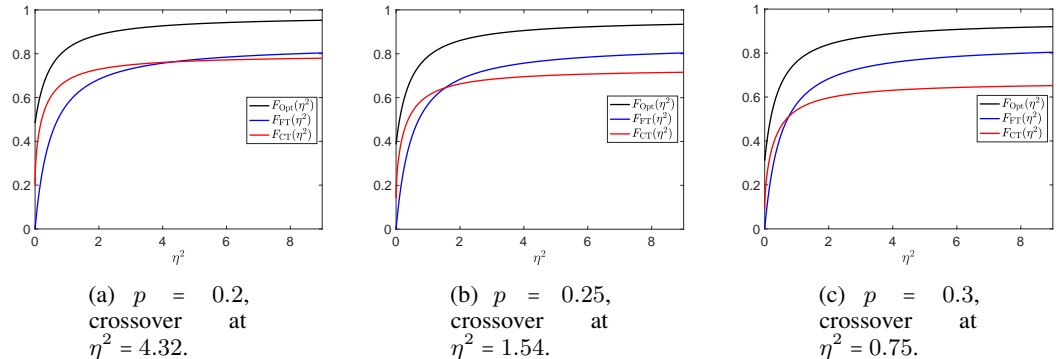

(a) $p = 0.2$, crossover at $\eta^2 = 4.32$.

(b) $p = 0.25$, crossover at $\eta^2 = 1.54$.

(c) $p = 0.3$, crossover at $\eta^2 = 0.75$.

Figure 5: The AMP update mappings for optimal aggregator, full-retraining and consensus-based retraining. Here, $\gamma = 1.5$, $\alpha = 2$, $\pi_+ = 0.3$, $\pi_- = 0.7$.

For these functions, we can run the AMP iterates (the memory correction terms are well defined) and also derive the state evolution recursion to predict the asymptotic behavior of estimates across iteration. We can then take the limit of $\beta \to \infty$ to make this approximation tight. It is worth noting that the order fo limits are important; we take the limit $\beta \to \infty$ *after* taking the limit $n, d \to \infty$. The state evolution curves for several values of $\beta$ are plotted in Figures 3-4. Here, we derive the state evolution curves for $\beta \to \infty$.

**Full retraining:** Consider the state evolution (8) with $g_t^{\mathsf{FT}}(y, \widehat{y})$. For fixed $t$, taking $\beta \to \infty$ results in the following update:

$$\bar{m}_t = \gamma\sqrt{\alpha}\, m_t, \quad \bar{\sigma}_t^2 = \alpha(m_t^2 + \sigma_t^2), \quad m_{t+1} = \frac{\gamma}{\sqrt{\alpha}}\, \mathbb{E}\{\mathrm{sign}(\bar{m}_t Y + \bar{\sigma}_t G) Y\}, \quad \sigma_{t+1}^2 = 1\,.$$

Defining $\bar{\eta}_t := \left(\frac{\bar{m}_t}{\bar{\sigma}_t}\right)^2$ and $\eta_t := \frac{m_t}{\sigma_t} = m_t$, the above recursion can be simplified to

$$\bar{\eta}_t = \gamma^2 \frac{\eta_t^2}{\eta_t^2 + 1}\,, \tag{39}$$

$$\eta_{t+1}^2 = \frac{\gamma^2}{\alpha}\, \mathbb{E}\{Y\,\mathrm{sign}(\sqrt{\bar{\eta}_t}Y + G)\}^2\,, \tag{40}$$

with initialization $\eta_1 = \gamma(1 - 2p)/\sqrt{\alpha}$.

Since $Y \in \{-1, +1\}$ independent of $G \sim \mathsf{N}(0, 1)$, we have

$$\mathbb{E}\{Y\,\mathrm{sign}(\sqrt{\bar{\eta}_t}Y + G)\} = \mathbb{E}\{\mathrm{sign}(\sqrt{\bar{\eta}_t} + G)\} = 2\mathbb{P}(-G < \sqrt{\bar{\eta}_t}) = 2\Phi(\sqrt{\bar{\eta}_t}) - 1\,.$$

So the state evolution can be simplified as:

$$\eta_{t+1}^2 = \frac{\gamma^2}{\alpha}\left(2\Phi\left(\frac{\gamma\eta_t}{\sqrt{1 + \eta_t^2}}\right) - 1\right)^2\,. \tag{41}$$

**Consensus-based retraining:** Consider the state evolution (8) with $g_t^{\mathsf{CT}}(y, \widehat{y})$. For fixed $t$, taking $\beta \to \infty$ results in the following update:

$$\bar{m}_t = \gamma\sqrt{\alpha}\, m_t, \quad \bar{\sigma}_t^2 = \alpha(m_t^2 + \sigma_t^2),$$

$$m_{t+1} = \frac{\gamma}{\sqrt{\alpha}}\, \mathbb{E}\{Y\widehat{Y}\mathbb{1}(\widehat{Y}(\bar{m}_t Y + \bar{\sigma}_t G) > 0)\}, \quad \sigma_{t+1}^2 = \mathbb{E}\{\mathbb{1}(\widehat{Y}(\bar{m}_t Y + \bar{\sigma}_t G) > 0)\}\,.$$

Defining $\bar{\eta}_t := \left(\frac{\bar{m}_t}{\bar{\sigma}_t}\right)^2$ and $\eta_t := \frac{m_t}{\sigma_t} = m_t$, the above recursion can be simplified to

$$\bar{\eta}_t = \gamma^2 \frac{\eta_t^2}{\eta_t^2 + 1}\,, \tag{42}$$

$$\eta_{t+1}^2 = \frac{\gamma^2}{\alpha}\, \frac{\mathbb{E}\{Y\widehat{Y}\mathbb{1}(\widehat{Y}(\sqrt{\bar{\eta}_t}Y + G) > 0)\}^2}{\mathbb{E}\{\mathbb{1}(\widehat{Y}(\sqrt{\bar{\eta}_t}Y + G)) > 0\}}\,, \tag{43}$$

with initialization $\eta_1 = \gamma(1 - 2p)/\sqrt{\alpha}$.

Let $\Delta := Y\widehat{Y}$. Then $\mathbb{P}(\Delta = 1) = 1 - p$ and $\mathbb{P}(\Delta = -1) = p$. Since $G \sim \mathsf{N}(0, 1)$ is independent of $Y, \widehat{Y}$, we have

$$
\begin{aligned}
\mathbb{E}\{Y\widehat{Y}\mathbb{1}(\widehat{Y}(\sqrt{\bar{\eta}_t}Y + G) > 0)\} &= \mathbb{E}\{\Delta\mathbb{1}(\sqrt{\bar{\eta}_t}\Delta + G > 0)\} \\
&= \mathbb{E}\{\Delta\Phi(\sqrt{\bar{\eta}_t}\Delta)\} \\
&= (1 - p)\Phi(\sqrt{\bar{\eta}_t}) - p\Phi(-\sqrt{\bar{\eta}_t}) \\
&= -p + \Phi(\sqrt{\bar{\eta}_t}) .
\end{aligned}
$$

We also have

$$
\begin{aligned}
\mathbb{E}\{\mathbb{1}(\widehat{Y}(\sqrt{\bar{\eta}_t}Y + G)) > 0\} &= \mathbb{E}\{\mathbb{P}(\Delta\sqrt{\bar{\eta}_t} + G > 0)\} \\
&= \mathbb{E}\{\Phi(\Delta\sqrt{\bar{\eta}_t})\} \\
&= (1 - p)\Phi(\sqrt{\bar{\eta}_t}) + p\Phi(-\sqrt{\bar{\eta}_t}) \\
&= p + (1 - 2p)\Phi(\sqrt{\bar{\eta}_t}) .
\end{aligned}
$$

Substituting these terms in (43), the state evolution reads:

$$
\bar{\eta}_t = \gamma^2 \frac{\eta_t^2}{\eta_t^2 + 1}
$$

$$
\eta_{t+1}^2 = \frac{\gamma^2}{\alpha} \frac{(\Phi(\sqrt{\bar{\eta}_t}) - p)^2}{p + (1 - 2p)\Phi(\sqrt{\bar{\eta}_t})} .
$$

These derivations also provide valuable insight into full-retraining and consensus-based retraining. We denote by $F_{\mathrm{FT}}$, $F_{\mathrm{CT}}$, and $F_{\mathrm{Opt}}$ the AMP update mappings for full-retraining, consensus-based retraining, and the Bayes-optimal aggregator, respectively. For example, the state evolution for full-retraining is given by $\eta_{t+1}^2 = F_{\mathrm{FT}}(\eta_t^2)$, with $F_{\mathrm{FT}}$ given by (41). In Figure 5, we plot these functions for different values of $p$. As we see, $F_{\mathrm{Opt}}$ is uniformly larger than the other two mappings, and so the corresponding state evolution sequence is uniformly larger than that of the other two aggregators. Recall that the test error of $\boldsymbol{\theta}^t$ decreases as $\eta_t$ increases, and hence the optimal aggregator performs better than the other two, at every iteration. For $F_{\mathrm{FT}}$ and $F_{\mathrm{CT}}$, there is a crossover point: before this, $F_{\mathrm{CT}}(\eta^2) > F_{\mathrm{FT}}(\eta^2)$, and after it, $F_{\mathrm{CT}}(\eta^2) < F_{\mathrm{FT}}(\eta^2)$. This is intuitive, because small $\eta$ means that the current model quality is poor, so the consensus-based aggregator, which takes into account the noisy labels, works better. However, larger $\eta$ means that the model quality is good enough that full-retraining, which ignores the noisy labels, works better. Furthermore, the crossover point depends on $p$: it increases as $p$ decreases. This is also intuitive, because smaller $p$ gives a larger range of $\eta$ where considering noisy labels in the retraining step is beneficial.

## G   Proof of Lemma 4.1

For a test data point $(\boldsymbol{x}, y)$, we have $\widehat{y} = \mathrm{sign}(\boldsymbol{x}^\mathsf{T}\boldsymbol{\theta})$ and $\mathbb{P}(y = +1|\boldsymbol{x}) = h(\boldsymbol{x}^\mathsf{T}\boldsymbol{\beta})$. We let $z_{\boldsymbol{\theta}} := \sqrt{n}\boldsymbol{x}^\mathsf{T}\boldsymbol{\theta}$ and $z_{\boldsymbol{\beta}} = \sqrt{n}\boldsymbol{x}^\mathsf{T}\boldsymbol{\beta}$. We have

$$
\begin{aligned}
P_e(\boldsymbol{\theta}) &= \mathbb{P}(y\widehat{y} < 0) \\
&= \mathbb{P}(z_{\boldsymbol{\theta}} > 0, y = -1) + \mathbb{P}(z_{\boldsymbol{\theta}} < 0, y = 1) \\
&= \mathbb{E}_{z_{\boldsymbol{\beta}}}[\mathbb{P}(z_{\boldsymbol{\theta}} > 0, y = -1|z_{\boldsymbol{\beta}}) + \mathbb{P}(z_{\boldsymbol{\theta}} < 0, y = 1|z_{\boldsymbol{\beta}})]
\end{aligned}
$$

We continue by calculating $\mathbb{P}(z_{\boldsymbol{\theta}} > 0, y = -1|z_{\boldsymbol{\beta}})$. Conditional on $z_{\boldsymbol{\beta}}$, $y$ and $z_{\boldsymbol{\theta}}$ are independent. Therefore,

$$
\mathbb{P}(z_{\boldsymbol{\theta}} > 0, y = -1|z_{\boldsymbol{\beta}}) = \mathbb{P}(z_{\boldsymbol{\theta}} > 0|z_{\boldsymbol{\beta}})\mathbb{P}(y = -1|z_{\boldsymbol{\beta}}) .
$$

We have $(z_{\boldsymbol{\theta}}, z_{\boldsymbol{\beta}}) \sim \mathsf{N}(0, \begin{bmatrix} \|\boldsymbol{\theta}\|_{\ell_2}^2 & \boldsymbol{\beta}^\mathsf{T}\boldsymbol{\theta} \\ \boldsymbol{\beta}^\mathsf{T}\boldsymbol{\theta} & \|\boldsymbol{\beta}\|_{\ell_2}^2 \end{bmatrix})$. Hence,

$$
z_{\boldsymbol{\theta}}|z_{\boldsymbol{\beta}} \sim \mathsf{N}(a, b), \quad a = \frac{\boldsymbol{\beta}^\mathsf{T}\boldsymbol{\theta}}{\|\boldsymbol{\beta}\|_{\ell_2}^2} z_{\boldsymbol{\beta}}, \quad b^2 = \|\boldsymbol{\theta}\|_{\ell_2}^2 - \frac{(\boldsymbol{\beta}^\mathsf{T}\boldsymbol{\theta})^2}{\|\boldsymbol{\beta}\|_{\ell_2}^2} .
$$

Using this characterization, we get

$$\mathbb{P}(z_{\boldsymbol{\theta}} > 0|z_{\boldsymbol{\beta}}) = \Phi(\frac{a}{b}) = \Phi\Big(\frac{\rho}{\sqrt{1-\rho^2}}\frac{z_{\boldsymbol{\beta}}}{\|\boldsymbol{\beta}\|_{\ell_2}}\Big)$$

Also under our data generative model, $\mathbb{P}(y = -1|z_{\boldsymbol{\beta}}) = 1 - h(z_{\boldsymbol{\beta}}/\sqrt{n})$. Note that $z_{\boldsymbol{\beta}} \sim \mathsf{N}(0, \|\boldsymbol{\beta}\|_{\ell_2}^2)$ and $\frac{\|\boldsymbol{\beta}\|_{\ell_2}}{\sqrt{n}} \to \sqrt{\alpha}\gamma$, in probability. Hence, by Slutsky's theorem,

$$\mathbb{P}(z_{\boldsymbol{\theta}} > 0, y = -1|z_{\boldsymbol{\beta}}) \to \Phi\left(\frac{\rho Z}{\sqrt{1-\rho^2}}\right)(1 - h(\sqrt{\alpha}\gamma Z)),$$

in distribution with $Z \sim \mathsf{N}(0,1)$. By a similar argument, we have

$$\mathbb{P}(z_{\boldsymbol{\theta}} < 0, y = 1|z_{\boldsymbol{\beta}}) \to \Phi\left(\frac{-\rho Z}{\sqrt{1-\rho^2}}\right)h(\sqrt{\alpha}\gamma Z).$$

Adding the previous two equations, we obtain the desired result.

We next show that $F$ is decreasing. Using the relation $\Phi(-x) = 1 - \Phi(x)$ we can write $F(\rho)$ as:

$$F(\rho) = \mathbb{E}\left[h(\sqrt{\alpha}\gamma Z) + \Phi\left(\frac{\rho Z}{\sqrt{1-\rho^2}}\right)(1 - 2h(\sqrt{\alpha}\gamma Z))\right].$$

Hence,

$$F'(\rho) = \frac{1}{(1-\rho^2)^{3/2}}\mathbb{E}\left[Z\phi\left(\frac{\rho Z}{\sqrt{1-\rho^2}}\right)(1 - 2h(\sqrt{\alpha}\gamma Z))\right],$$

with $\phi(u) = e^{-u^2/2}/\sqrt{2\pi}$. For any $z > 0$, we have

$$z\phi\left(\frac{\rho z}{\sqrt{1-\rho^2}}\right)(1 - 2h(\sqrt{\alpha}\gamma z)) - z\phi\left(\frac{-\rho z}{\sqrt{1-\rho^2}}\right)(1 - 2h(-\sqrt{\alpha}\gamma z))$$

$$= 2z\phi\left(\frac{\rho z}{\sqrt{1-\rho^2}}\right)(h(-\sqrt{\alpha}\gamma z) - h(\sqrt{\alpha}\gamma z)) < 0, \tag{44}$$

since $z > 0$ and by assumption $h(u) > h(-u)$ for all $u > 0$. Also, since the Gaussian density is symmetric, (44) implies that $F'(\rho) < 0$, which completes the proof.

## H   Proof of Theorem 4.2

Similar to the proof of Theorem 3.1, we build some intuition by analyzing the behavior of the first few estimate of AMP updates.

Since $g_0(\cdot, \widehat{y}) = \widehat{y}$, we have $C_0 = 0$ and $\boldsymbol{\beta}^1 = \frac{1}{\sqrt{n}}\boldsymbol{X}^\top\widehat{y}$. Recall that under GLM setting the entries of $\boldsymbol{X}$ are i.i. gaussian. Therefore, to analyze the distribution of $\boldsymbol{\beta}^1$, we use the rotation invariance of Gaussian distribution, and without loss of generality we take $\boldsymbol{\beta} = \|\boldsymbol{\beta}\|_{\ell_2} \boldsymbol{e}_1$. We then have $\mathbb{P}(\widehat{y} = 1|\boldsymbol{x}) = \hat{h}_p(\|\boldsymbol{\beta}\|_{\ell_2} x_1)$. Also, $\boldsymbol{\beta}^1 = \begin{bmatrix}\langle \boldsymbol{x}^{(1)}, \widehat{y}\rangle \\ \boldsymbol{X}_{-1}^\top\widehat{y}\end{bmatrix}$ where we used the decomposition $\boldsymbol{X} = [\boldsymbol{x}^{(1)}|\boldsymbol{X}_{-1}]$. Note that $\widehat{y}$ and $\boldsymbol{X}_{-1}$ are independent since $\boldsymbol{\beta} = \|\boldsymbol{\beta}\|_{\ell_2} \boldsymbol{e}_1$. Hence, $\boldsymbol{X}_{-1}^\top\widehat{y} \sim \mathsf{N}(0, \boldsymbol{I}_{d-1})$, given that the entries of $\boldsymbol{X}_{-1}$ are i.i.d drawn from $\mathsf{N}(0, 1/n)$.

For the first entry, we write $\boldsymbol{x}^{(1)} = \boldsymbol{z}_0/\sqrt{n}$ where $\boldsymbol{z}_0 \sim \mathsf{N}(0,1)$. Then $\langle \boldsymbol{x}^{(1)}, \widehat{y}\rangle = \frac{1}{\sqrt{n}}\langle \boldsymbol{z}_0, \widehat{y}\rangle$. Note that the variables $z_{0,i}\widehat{y}_i$ are i.i.d and

$$\mu := \mathbb{E}[z_{0,i}\widehat{y}_i] = \mathbb{E}\left[Z_0\big(2\hat{h}_p\big(\sqrt{\alpha}\gamma Z_0\big) - 1\big)\right] = 2\mathbb{E}\left[Z_0\hat{h}_p\big(\sqrt{\alpha}\gamma Z_0\big)\right],$$

with $Z_0 \sim \mathsf{N}(0,1)$. Also $\sigma^2 := \mathrm{Var}[z_{0,i}\widehat{y}_i] = 1 - 4\mathbb{E}\left[Z_0\hat{h}_p\big(\sqrt{\alpha}\gamma Z_0\big)\right]^2$. By the central limit theorem, $(\langle \boldsymbol{x}^{(1)}, \widehat{y}\rangle - \sqrt{n}\mu)$ converges in distribution to $\mathsf{N}(0, \sigma^2)$. Writing it differently, $\langle \boldsymbol{e}_1, \boldsymbol{\beta}^1 - \sqrt{n}\frac{\mu\boldsymbol{\beta}}{\|\boldsymbol{\beta}\|_{\ell_2}}\rangle$

converges in distribution to $\mathsf{N}(0, \sigma^2)$ with $\boldsymbol{e}_1 = (1, 0, \ldots, 0) \in \mathbb{R}^d$. In addition, by Assumption [2], $\frac{\sqrt{n}}{\|\boldsymbol{\beta}\|_{\ell_2}} \to \frac{1}{\sqrt{\alpha}\gamma}$, and the the empirical distribution of the entries of $\boldsymbol{\beta}$ converges weakly to $\bar{\beta} \sim \pi_{\bar{\beta}}$. This implies that the empirical distribution of entries of $\boldsymbol{\beta}^1$ converges in distribution to $\frac{\mu}{\sqrt{\alpha}\gamma}\bar{\beta} + G$ with $G \sim \mathsf{N}(0, 1)$ independent of $\bar{\beta}$, which verifies the claim of the theorem for $t = 1$ with

$$\mu_1 = \frac{\mu}{\sqrt{\alpha}\gamma} = \frac{2}{\sqrt{\alpha}\gamma}\mathbb{E}\left[Z_0 \hat{h}_p\left(\sqrt{\alpha}\gamma Z_0\right)\right] = \frac{2}{\alpha\gamma^2}\mathbb{E}\left[Z\hat{h}_p(Z)\right], \qquad \sigma_1 = \sqrt{\alpha}, \tag{45}$$

with $Z \sim \mathsf{N}(0, \alpha\gamma^2)$.

The formal proof largely follows from the analysis of generalized AMP (GAMP) algorithm [35], which given an initial estimate $\boldsymbol{\beta}^1$, iteratively produces estimates $\boldsymbol{\beta}$ and $\boldsymbol{y}^t$ of $\boldsymbol{\beta}$ and $\boldsymbol{X}\boldsymbol{\beta}$ as follows:

$$\boldsymbol{\beta}^{t+1} = \boldsymbol{X}^\mathsf{T} g_t(\boldsymbol{y}^t, \widehat{\boldsymbol{y}}) - C_t f_t(\boldsymbol{\beta}^t) \qquad C_t = \frac{1}{n}\sum_{i=1}^{n}\frac{\partial g_t}{\partial \boldsymbol{y}}(y_i^t, \widehat{y}_i)$$

$$\boldsymbol{y}^{t+1} = \boldsymbol{X}\boldsymbol{\beta}^{t+1} - B_t g_t(\boldsymbol{y}^t, \widehat{\boldsymbol{y}}) \qquad B_t = \frac{1}{n}\sum_{j=1}^{d}\frac{\partial f_t}{\partial \boldsymbol{y}}(\beta_i^t)$$

where $g_t : \mathbb{R}^2 \to \mathbb{R}$ and $f_t : \mathbb{R} \to \mathbb{R}$ are Lipschitz in their first argument. Suppose that

$$\frac{1}{n}\begin{bmatrix} \|\boldsymbol{\beta}\|_{\ell_2}^2 & \boldsymbol{\beta}^\mathsf{T}\boldsymbol{\beta}^1 \\ \boldsymbol{\beta}^\mathsf{T}\boldsymbol{\beta}^1 & \|\boldsymbol{\beta}^1\|_{\ell_2}^2 \end{bmatrix} \xrightarrow{p} \boldsymbol{\Sigma}^1 \,.$$

With $\boldsymbol{\Sigma}^1$, the state evolution parameters $\mu_t \in \mathbb{R}, \sigma_t \in \mathbb{R}_{\geq 0}, \boldsymbol{\Sigma}^t \in \mathbb{R}^{2 \times 2}$ are recursively defined by

$$\mu_{t+1} = \mathbb{E}(\partial_z g_t(Z_t, \widehat{Y}(Z))), \quad \sigma_{t+1}^2 = \mathbb{E}(g_t(Z_t, \widehat{Y}(Z))^2), \tag{46}$$

$$\boldsymbol{\Sigma}^{t+1} = \begin{bmatrix} \alpha\,\mathbb{E}(\bar{\beta}^2) & \alpha\,\mathbb{E}(\bar{\beta}f_{t+1}(\mu_{t+1}\bar{\beta} + \sigma_{t+1}G_{t+1})) \\ \alpha\,\mathbb{E}(\bar{\beta}f_{t+1}(\mu_{t+1}\bar{\beta} + \sigma_{t+1}G_{t+1})) & \alpha\,\mathbb{E}(f_{t+1}(\mu_{t+1}\bar{\beta} + \sigma_{t+1}G_{t+1})^2) \end{bmatrix}, \tag{47}$$

where $(Z, Z_t) \sim \mathsf{N}(0, \boldsymbol{\Sigma}_t)$, and $\widehat{Y} = \widehat{Y}(Z) \in \{-1, +1\}$ with $\mathbb{P}(\widehat{Y} = 1 | Z) = \hat{h}_p(Z)$. In addition, $G_{t+1} \sim \mathsf{N}(0, 1)$ independent of $\bar{\beta} \sim \pi_{\bar{\beta}}$.

Similar to [27, Proposition 3.1], Stein's lemma can be used to further simplify the state evolution. Define

$$\mu_{Z,t} = \frac{\boldsymbol{\Sigma}_{2,1}^t}{\boldsymbol{\Sigma}_{1,1}^t}, \quad \sigma_{Z,t}^2 = \boldsymbol{\Sigma}_{2,2}^t - \frac{(\boldsymbol{\Sigma}_{1,2}^t)^2}{\boldsymbol{\Sigma}_{1,1}^t}, \tag{48}$$

and let $Z_t \overset{d}{=} \mu_{Z,t}Z + \sigma_{Z,t}G$ with $G \sim \mathsf{N}(0, 1)$ independent of $Z$. Then, the state evolution can be characterized by the sequence $(\mu_t, \sigma_t)_{t \geq 0}$ with

$$\mu_{t+1} = \mathbb{E}\left(\frac{\mathbb{E}(Z|Z_t, \widehat{Y}) - \mathbb{E}(Z|Z_t)}{\mathrm{Var}(Z|Z_t)}g_t(Z_t, \widehat{Y})\right), \quad \sigma_{t+1}^2 = \mathbb{E}(g_t(Z_t, \widehat{Y})^2). \tag{49}$$

It is proved that the state evolution recursion precisely characterize the asymptotic behavior of the GAMP update as stated in the next theorem. We refer to [35] or [14] for a formal proof.

**Theorem H.1** *Let* $(\boldsymbol{\beta}^t, \boldsymbol{y}^t)_{t \geq 0}$ *be the AMP iterates given by* (17)-(18). *Also let* $(\mu_t, \sigma_t)_{t \geq 0}$ *be the state evolution recursions given by* (22). *Then, for any pseudo-Lipschitz function* $\psi : \mathbb{R}^2 \to \mathbb{R}$ *the following holds almost surely for* $t \geq 0$:

$$\lim_{n \to \infty}\left|\frac{1}{d}\sum_{i=1}^{d}\psi(\beta_i^t, \beta_i) - \mathbb{E}\left[\psi\left(\mu_t\bar{\beta} + \sigma_t G, \bar{\beta}\right)\right]\right| = 0 \,,$$

$$\lim_{n \to \infty}\left|\frac{1}{n}\sum_{j=1}^{n}\psi(y_i^t, \widehat{y}_i) - \mathbb{E}\left[\psi\left(\mu_{Z,t}Z + \sigma_{Z,t}\tilde{G}, \widehat{Y}(Z)\right)\right]\right| = 0 \,,$$

*where* $G \sim \mathsf{N}(0, 1)$ *and* $\bar{\beta} \sim \pi_{\bar{\beta}}$ *independently.*

A challenge in applying Theorem H.1 is that it requires the initialization to be independent of the random matrix $\boldsymbol{X}$, a property that does not hold for $\boldsymbol{\beta}^1 = \boldsymbol{X}^\mathsf{T}\widehat{\boldsymbol{y}}$. The trick is that we consider the AMP updates, starting from $t = 0$ with initialization $\boldsymbol{\beta}^0 = \boldsymbol{0}$ and $g_{-1}(\cdot,\cdot) = 0$, so the initialization is independent of $\boldsymbol{X}$. By the update rules, we get $\boldsymbol{y}^0 = \boldsymbol{0}$ and $\boldsymbol{\beta}^1 = \boldsymbol{X}^\mathsf{T}\widehat{\boldsymbol{y}}$, if we define $g_0(y,\widehat{y}) = \widehat{y}$ (and so $C_0 = 0$). In other words, in the next iteration we get our previous initialization. This way we can apply Theorem H.1. Note that

$$\frac{1}{n}\begin{bmatrix} \|\boldsymbol{\beta}\|_{\ell_2}^2 & \boldsymbol{\beta}^\mathsf{T}\boldsymbol{\beta}^0 \\ \boldsymbol{\beta}^\mathsf{T}\boldsymbol{\beta}^0 & \|\boldsymbol{\beta}^0\|_{\ell_2}^2 \end{bmatrix} \xrightarrow{p} \boldsymbol{\Sigma}^0 = \begin{bmatrix} \alpha\gamma^2 & 0 \\ 0 & 0 \end{bmatrix}.$$

From (48), we have $\mu_{Z,0} = \sigma_{Z,0} = 0$ and so by (49), we get

$$\mu_1 = \mathbb{E}\left(\frac{\mathbb{E}(Z|Z_0,\widehat{Y}) - \mathbb{E}(Z|Z_0)}{\mathrm{Var}(Z|Z_0)}g_0(Z_0,\widehat{Y})\right)$$

$$= \mathbb{E}\left(\frac{\mathbb{E}(Z|\widehat{Y})}{\alpha\gamma^2}\widehat{Y}\right) = \frac{1}{\alpha\gamma^2}\mathbb{E}(\widehat{Y}Z)$$

$$= \frac{1}{\alpha\gamma^2}\mathbb{E}(2(\hat{h}_p(Z) - 1)Z) = \frac{2}{\alpha\gamma^2}\mathbb{E}(Z\hat{h}_p(Z))$$

where we used the fact that $Z_0 = 0$ in this case (since $\mu_{Z,0} = \sigma_{Z,0} = 0$) and $g_0(y,\widehat{y}) = \widehat{y}$. In addition, $\sigma_1^2 = \mathbb{E}(g_0(Z_0,\widehat{Y})^2) = 1$. This is consistent with (45) where we applied the change of variable $\sqrt{\alpha}\sigma_t \to \sigma_t$.

Next to obtain the state evolution recursion for $t > 1$, we recall the construction of $\boldsymbol{\Sigma}^t$ from (47), and take $f_t$ to be the identity functions, by which we obtain

$$\boldsymbol{\Sigma}^t = \begin{bmatrix} \alpha\gamma^2 & \alpha\mu_t\gamma^2 \\ \alpha\mu_t\gamma^2 & \alpha(\mu_t^2\gamma^2 + \sigma_t^2) \end{bmatrix}. \tag{50}$$

By invoking (48), we have

$$\mu_{Z,t} = \frac{\alpha\mu_t\gamma^2}{\alpha\gamma^2} = \mu_t, \quad \sigma_{Z,t}^2 = \alpha(\mu_t^2\gamma^2 + \sigma_t^2) - \frac{(\alpha\mu_t\gamma^2)^2}{\alpha\gamma^2} = \alpha\sigma_t^2.$$

Hence, in the state evolution (49), we have $Z_t \overset{d}{=} \mu_t Z + \sqrt{\alpha}\sigma_t G$, with $Z \sim \mathsf{N}(0, \alpha\gamma^2)$. This results in (20) after the change of variable $\sqrt{\alpha}\sigma_t \to \sigma_t$, and so Theorem 4.2 follows from the result of Theorem H.1.

We next derive the limit of $P_e(\boldsymbol{\theta})$. As we showed in the proof of Theorem 3.1, functions $\psi(x,y) = xy$ and $\psi(x,y) = x^2$ are pseudo-Lipschitz (of order 2). Therefore, using (23), the following limits hold almost surely,

$$\lim_{n\to\infty}\frac{1}{d}\|\boldsymbol{\beta}^t\|_{\ell_2}^2 = \lim_{n\to\infty}\frac{1}{d}\sum_{i=1}^d |(\beta_i^t)^2| = \mathbb{E}\left[\left(\mu_t\bar{\beta} + \frac{\sigma_t}{\sqrt{\alpha}}G\right)^2\right] = \mu_t^2\,\mathbb{E}[\bar{\beta}^2] + \frac{\sigma_t^2}{\alpha} = \mu_t^2\gamma^2 + \frac{\sigma_t^2}{\alpha},$$

$$\lim_{n\to\infty}\frac{1}{d}\|\boldsymbol{\beta}\|_{\ell_2}^2 = \lim_{n\to\infty}\frac{1}{d}\sum_{i=1}^d |(\beta_i)^2| = \mathbb{E}[\bar{\beta}^2] = \gamma^2,$$

$$\lim_{n\to\infty}\frac{1}{d}\boldsymbol{\beta}^\mathsf{T}\boldsymbol{\beta}^t = \lim_{n\to\infty}\frac{1}{d}\sum_{i=1}^d (\beta_i\beta_i^t) = \mathbb{E}\left[\left(\mu_t\bar{\beta} + \frac{\sigma_t}{\sqrt{\alpha}}G\right)\bar{\beta}\right] = \mu_t\,\mathbb{E}[\bar{\beta}^2] = \mu_t\gamma^2,$$

where we used the identity $\mathbb{E}[\bar{\beta}^2] = \gamma^2$ (second bullet point in Assumption 2). Therefore, defining $\rho^t := \boldsymbol{\beta}^\mathsf{T}\boldsymbol{\beta}^t/(\|\boldsymbol{\beta}\|_{\ell_2}\|\boldsymbol{\beta}^t\|_{\ell_2})$, we have

$$\lim_{n\to\infty}\rho_t = \frac{\mu_t\gamma^2}{\gamma\sqrt{\mu_t^2\gamma^2 + \frac{\sigma_t^2}{\alpha}}} = \frac{\eta_t\gamma}{\sqrt{\eta_t^2\gamma^2 + \frac{1}{\alpha}}}.$$

Next by Lemma 4.1 and continuity of function $F$, we have

$$\lim_{n\to\infty}P_e(\boldsymbol{\beta}^t) = \lim_{n\to\infty}F(\rho_t) = F(\lim_{n\to\infty}\rho_t) = F\left(\frac{\eta_t\gamma}{\sqrt{\eta_t^2\gamma^2 + \frac{1}{\alpha}}}\right),$$

which concludes the proof.

# I    Proof of Theorem 4.3

The proof follows from (25) and an explicit derivation of $\mathbb{E}(Z|Z_t, \widehat{Y})$. By Bayes rule,

$$
\mathbb{P}(Z = z | Z_t = z_t, \widehat{Y} = \widehat{y}) = \frac{\mathbb{P}(Z_t = z_t, \widehat{Y} = \widehat{y} | Z = z)\mathbb{P}(Z = z)}{\int_{-\infty}^{\infty} \mathbb{P}(Z_t = z_t, \widehat{Y} = \widehat{y} | Z = z)\mathbb{P}(Z = z)\mathrm{d}z}
$$

$$
= \frac{\frac{1}{\sqrt{2\pi}\sigma_t} e^{-\frac{(z_t - \mu_t z)^2}{2\sigma_t^2}} \hat{h}_p(z)^{\frac{1+\widehat{y}}{2}} (1 - \hat{h}_p(z))^{\frac{1-\widehat{y}}{2}} \frac{1}{\sqrt{2\pi}\alpha\gamma} e^{-\frac{z^2}{2\alpha\gamma^2}}}{\int_{-\infty}^{\infty} \frac{1}{\sqrt{2\pi}\sigma_t} e^{-\frac{(z_t - \mu_t z)^2}{2\sigma_t^2}} \hat{h}_p(z)^{\frac{1+\widehat{y}}{2}} (1 - \hat{h}_p(z))^{\frac{1-\widehat{y}}{2}} \frac{1}{\sqrt{2\pi}\alpha\gamma} e^{-\frac{z^2}{2\alpha\gamma^2}} \mathrm{d}z}
$$

Therefore,

$$
\mathbb{E}(Z|Z_t, \widehat{Y}) = \frac{\int_{-\infty}^{\infty} z e^{-\frac{(u - \mu_t z)^2}{2\sigma_t^2}} \hat{h}_p(z)^{\frac{1+\widehat{y}}{2}} (1 - \hat{h}_p(z))^{\frac{1-\widehat{y}}{2}} e^{-\frac{z^2}{2\alpha\gamma^2}} \mathrm{d}z}{\int_{-\infty}^{\infty} e^{-\frac{(u - \mu_t z)^2}{2\sigma_t^2}} \hat{h}_p(z)^{\frac{1+\widehat{y}}{2}} (1 - \hat{h}_p(z))^{\frac{1-\widehat{y}}{2}} e^{-\frac{z^2}{2\alpha\gamma^2}} \mathrm{d}z}, \tag{51}
$$

which after substituting in (25), results in

$$
g_t^*(u, \widehat{y}) = \left(\frac{1}{\alpha\gamma^2} + \frac{\mu_t^2}{\sigma_t^2}\right) \frac{\int_{-\infty}^{\infty} z e^{-\frac{(u - \mu_t z)^2}{2\sigma_t^2}} \hat{h}_p(z)^{\frac{1+\widehat{y}}{2}} (1 - \hat{h}_p(z))^{\frac{1-\widehat{y}}{2}} e^{-\frac{z^2}{2\alpha\gamma^2}} \mathrm{d}z}{\int_{-\infty}^{\infty} e^{-\frac{(u - \mu_t z)^2}{2\sigma_t^2}} \hat{h}_p(z)^{\frac{1+\widehat{y}}{2}} (1 - \hat{h}_p(z))^{\frac{1-\widehat{y}}{2}} e^{-\frac{z^2}{2\alpha\gamma^2}} \mathrm{d}z} - \frac{\mu_t}{\sigma_t^2} u, \tag{52}
$$

In addition, using the characterization of $g_t^*$ in (25), the state evolution (22) can be written as

$$
\mu_1 = \frac{2}{\alpha\gamma^2} \mathbb{E}[Z\hat{h}_p(Z)], \quad \sigma_1 = \sqrt{\alpha},
$$

$$
\mu_{t+1} = \frac{1}{\alpha} \mathbb{E}\{g_t^*(\mu_t Z + \sigma_t G, \widehat{Y})^2\}, \quad \sigma_{t+1}^2 = \alpha\mu_{t+1}, \tag{53}
$$

with $Z \sim \mathsf{N}(0, \alpha\gamma^2)$ and $G \sim \mathsf{N}(0, 1)$ independent of each other. In addition, $\widehat{Y} \in \{-1, +1\}$ with $\mathbb{P}(\widehat{Y} = 1|Z) = \hat{h}_p(Z)$, and $g_0^*(\cdot, \widehat{y}) = \widehat{y}$.

We next note that by Equation 24, the test error depends on $\eta_t = \mu_t/\sigma_t$. We proceed by writing the state evolution (53) in terms of $\eta_t$. Note $\eta_{t+1} = \frac{\mu_{t+1}}{\sigma_{t+1}} = \sqrt{\frac{\mu_{t+1}}{\alpha}}$. By substituting for $\mu_{t+1} = \alpha\eta_{t+1}^2$ and $\sigma_t = \sqrt{\alpha\mu_t} = \alpha\eta_t$ we arrive at

$$
\eta_{t+1}^2 = \frac{1}{\alpha} \mathbb{E}\{g_t^*(\alpha\eta_t^2 Z + \alpha\eta_t G, \widehat{Y})^2\}, \quad \eta_1 = \frac{2}{\alpha^{3/2}\gamma^2} \mathbb{E}[Z\hat{h}_p(Z)].
$$

The optimal aggregator (52) can also be written in terms of $\eta_t$ as

$$
g_t^*(u, \widehat{y}) = \left(\frac{1}{\alpha\gamma^2} + \eta_t^2\right) \frac{\int_{-\infty}^{\infty} z e^{-\frac{1}{2}(\frac{u}{\alpha\eta_t} - \eta_t z)^2} \hat{h}_p(z)^{\frac{1+\widehat{y}}{2}} (1 - \hat{h}_p(z))^{\frac{1-\widehat{y}}{2}} e^{-\frac{z^2}{2\alpha\gamma^2}} \mathrm{d}z}{\int_{-\infty}^{\infty} e^{-\frac{1}{2}(\frac{u}{\alpha\eta_t} - \eta_t z)^2} \hat{h}_p(z)^{\frac{1+\widehat{y}}{2}} (1 - \hat{h}_p(z))^{\frac{1-\widehat{y}}{2}} e^{-\frac{z^2}{2\alpha\gamma^2}} \mathrm{d}z} - \frac{u}{\alpha}
$$

$$
= \left(\frac{1}{\alpha\gamma^2} + \eta_t^2\right) \frac{\int_{-\infty}^{\infty} z e^{-\frac{\eta_t^2 z^2}{2} + \frac{uz}{\alpha}} \hat{h}_p(z)^{\frac{1+\widehat{y}}{2}} (1 - \hat{h}_p(z))^{\frac{1-\widehat{y}}{2}} e^{-\frac{z^2}{2\alpha\gamma^2}} \mathrm{d}z}{\int_{-\infty}^{\infty} e^{-\frac{\eta_t^2 z^2}{2} + \frac{uz}{\alpha}} \hat{h}_p(z)^{\frac{1+\widehat{y}}{2}} (1 - \hat{h}_p(z))^{\frac{1-\widehat{y}}{2}} e^{-\frac{z^2}{2\alpha\gamma^2}} \mathrm{d}z} - \frac{u}{\alpha}
$$

Finally, replacing $u$ by $y$ gives us the desired result.

# J    Special Case: Sign Link Function

In this section, we specialize the result of Theorem 4.3 to sign link function namely $h(z) = \frac{1+\text{sign}(z)}{2}$. In this case $y = \text{sign}(x^\mathsf{T}\beta)$.

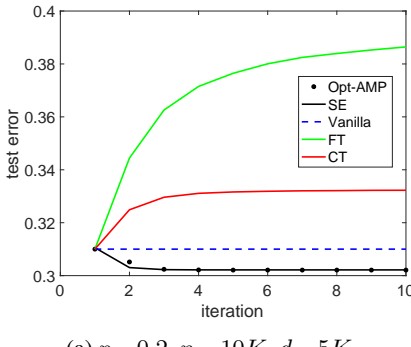
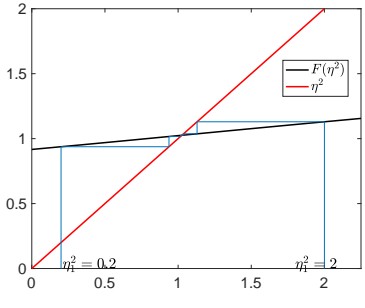

(a) $p = 0.2$, $n = 10K$, $d = 5K$

(b) Cobweb plot for the state evolution (56) in GLM setting

Figure 6: **Synthetic Experiments:** ($a$) Comparison between different retraining methods under GLM setting with link function $h(z) = (1 + \text{sign}(z))/2$. FT and CT respectively denote the full-retraining and the consensus-based retraining *without* the memory correction terms. Vanilla is the estimator without any retraining. ($b$) Cobweb plot for the state evolution mapping in (56).

**Proposition J.1** *Under the GLM setting with $h(z) = \frac{1+\text{sign}(z)}{2}$, the optimal aggregator functions in the AMP procedure are given by:*

$$g_t^*(u, \widehat{y}) = \frac{1}{s_t} \cdot \frac{(1 - 2p)\widehat{y}\sqrt{\frac{2}{\pi}} e^{-\frac{u^2 s_t^2}{2\alpha^2}}}{1 + (1 - 2p)\widehat{y}(2\Phi(\frac{u s_t}{\alpha}) - 1)}, \tag{54}$$

*for $t \geq 1$, and $\tilde{g}_0(\cdot, \widehat{y}) = \widehat{y}$, where $s_t^2 = (\frac{1}{\alpha} + \eta_t^2)^{-1}$ and the sequence $(\eta_t)_{t \geq 1}$ is given by the following state evolution recursion:*

$$\eta_1 = \frac{1 - 2p}{\alpha}\sqrt{\frac{2}{\pi}}, \tag{55}$$

$$\eta_{t+1}^2 = \frac{1}{\alpha} \mathbb{E}\{g_t^*(\alpha\eta_t^2 Z + \alpha\eta_t G, \widehat{Y})^2\}, \tag{56}$$

*with $Z \sim \mathsf{N}(0, \alpha)$ and $G \sim \mathsf{N}(0, 1)$ independent of each other. In addition, $\widehat{Y} \in \{-1, +1\}$ with $\mathbb{P}(\widehat{Y} = 1|Z) = \frac{1}{2} + \frac{1-2p}{2}\text{sign}(Z)$.*

*For the estimators $\boldsymbol{\beta}^t$ given by the AMP updates (17)-(18), with the optimal aggregator function (54), the following holds almost surely*

$$\lim_{n \to \infty} P_e(\boldsymbol{\beta}^t) = \frac{1}{\pi} \cos^{-1}\left(\frac{\eta_t}{\sqrt{\eta_t^2 + \frac{1}{\alpha}}}\right). \tag{57}$$

Figure 6a compares various retraining methods on synthetic GLM data, with $\boldsymbol{\beta}$ drawn from $\mathsf{N}(0, 1)$, $n = 1000$, $d = 500$, and label flip probability $p = 0.2$. Results are averaged over 10 runs.

Opt-AMP uses AMP updates with the optimal aggregator. Vanilla is a linear classifier without retraining, shown as a flat line. SE is the theoretical state evolution curve. FT and CT are full-retraining and consensus-based retraining methods without memory corrections, which are undefined here due to non-Lipschitz, non-differentiable aggregators. The plots show close agreement between SE predictions and Opt-AMP, and demonstrate the superior performance of Opt-AMP over other methods and the vanilla estimator.

In Figure 6b, we plot the cobweb plot for the state evolution (55)-(56). As we see the state evolution map is increasing and has at least one fixed point. Further, similar to the GMM setting, when $\eta_1$ is small (the initial model is poor), retraining helps improve its performance. However, if $\eta_1$ is large (above the fixed point) and so the initial estimator is already good enough, then retraining can actually hurt its performance.

## J.1 Proof of Proposition J.1

Consider the GLM setting with $h(z) = \frac{1+\text{sign}(z)}{2}$. In this case, $\hat{h}_p(z) = \frac{1}{2} + \frac{1-2p}{2}\text{sign}(z)$. Also, note that the data generative process does not depend on $\gamma$ (recall that $\|\boldsymbol{\beta}\|_{\ell_2}^2/d \to \gamma^2$), since the sign function is invariant to scaling. Hence, for simplicity we take $\gamma = 1$.

Before proceeding with this case, we derive an alternative expression for the optimal aggregator functions. We define the shorthand $s_t^2 = (\frac{1}{\alpha} + \eta_t^2)^{-1}$ and $m_t = \frac{u}{\alpha}s_t^2$. Also let $f(z) = \hat{h}_p(z)^{\frac{1+\hat{y}}{2}}(1 - \hat{h}_p(z))^{\frac{1-\hat{y}}{2}}$, where the explicit dependence on the parameters $p$ and $\hat{y}$ is omitted in the notation for simplicity. The function $g_t^*$ given by (54) can be written in terms of this notation as

$$g_t^*(u, \hat{y}) = \frac{1}{s_t^2}\frac{\int zf(z)e^{-\frac{(z-m_t)^2}{2s_t^2}}\,\mathrm{d}z}{\int f(z)e^{-\frac{(z-m_t)^2}{2s_t^2}}\,\mathrm{d}z} - \frac{u}{\alpha}$$

Let $Z_t = m_t + s_t Z$ with $Z \sim \mathsf{N}(0,1)$. Then,

$$\begin{aligned}
g_t^*(u, \hat{y}) &= \frac{1}{s_t^2}\frac{\mathbb{E}[Z_t f(Z_t)]}{\mathbb{E}[f(Z_t)]} - \frac{u}{\alpha}\\
&= \frac{1}{s_t^2}\frac{\mathbb{E}[(m_t + s_t Z)f(m_t + s_t Z)]}{\mathbb{E}[f(m_t + s_t Z)]} - \frac{u}{\alpha}\\
&= \frac{m_t}{s_t^2} - \frac{u}{\alpha} + \frac{1}{s_t^2}\frac{\mathbb{E}[s_t Z f(m_t + s_t Z)]}{\mathbb{E}[f(m_t + s_t Z)]}\\
&= \frac{1}{s_t}\frac{\mathbb{E}[Z f(m_t + s_t Z)]}{\mathbb{E}[f(m_t + s_t Z)]}.
\end{aligned}\tag{58}$$

For the case of sign link function, where $h(z) = \frac{1+\text{sign}(z)}{2}$, the function $f(z)$ can be written as

$$f(z) = \hat{h}_p(z)^{\frac{1+\hat{y}}{2}}(1 - \hat{h}_p(z))^{\frac{1-\hat{y}}{2}} = \frac{1}{2} + \frac{1-2p}{2}\text{sign}(\hat{y}z).$$

Using this expression in (58), we get

$$\begin{aligned}
g_t^*(u, \hat{y}) &= \frac{1}{s_t}\frac{\mathbb{E}\{Z f(m_t + s_t Z)\}}{\mathbb{E}\{f(m_t + s_t Z)\}}\\
&= \frac{1}{s_t}\cdot\frac{(1-2p)\,\mathbb{E}[Z\,\text{sign}(\hat{y}(m_t + s_t Z))]}{1 + (1-2p)\,\mathbb{E}[\text{sign}(\hat{y}(m_t + s_t Z))]}
\end{aligned}\tag{59}$$

To calculate the expectation in the numerator, we write

$$\begin{aligned}
\mathbb{E}\{Z\,\text{sign}(m_t + s_t Z)\} &= \int_{-m_t/s_t}^{\infty} z\frac{e^{-z^2/2}}{\sqrt{2\pi}} - \int_{-\infty}^{-m_t/s_t} z\frac{e^{-z^2/2}}{\sqrt{2\pi}}\\
&= \int_{-m_t/s_t}^{\infty} z\frac{e^{-z^2/2}}{\sqrt{2\pi}} + \int_{\infty}^{m_t/s_t} z\frac{e^{-z^2/2}}{\sqrt{2\pi}} = \frac{2}{\sqrt{2\pi}}e^{-\frac{m_t^2}{2s_t^2}}.
\end{aligned}$$

To compute the expectation in the denominator, we write

$$\mathbb{E}\{\text{sign}(m_t + s_t Z)\} = \int_{-m_t/s_t}^{\infty} \frac{e^{-z^2/2}}{\sqrt{2\pi}}\,\mathrm{d}z - \int_{-\infty}^{-m_t/s_t} \frac{e^{-z^2/2}}{\sqrt{2\pi}}\,\mathrm{d}z = 2\Phi(m_t/s_t) - 1.$$

Substituting the previous two identities in (59), we obtain

$$\begin{aligned}
g_t^*(u, \hat{y}) &= \frac{1}{s_t}\cdot\frac{(1-2p)\hat{y}\sqrt{2/\pi}e^{-\frac{m_t^2}{2s_t^2}}}{1 + (1-2p)\hat{y}(2\Phi(\frac{m_t}{s_t}) - 1)}\\
&= \frac{1}{s_t}\cdot\frac{(1-2p)\hat{y}\sqrt{\frac{2}{\pi}}e^{-\frac{u^2 s_t^2}{2\alpha^2}}}{1 + (1-2p)\hat{y}(2\Phi(\frac{u s_t}{\alpha}) - 1)}.
\end{aligned}\tag{60}$$

Next note that (55) and (56) follows from Theorem 4.3 where we can calculate $\eta_1$ explicitly as (note that $\gamma = 1$ in the current case)

$$
\begin{aligned}
\eta_1 &= \frac{2}{\alpha^{3/2}} \, \mathbb{E}[Z\hat{h}_p(Z)] \\
&= \frac{2}{\alpha^{3/2}} \, \mathbb{E}\Big[ Z\Big( \frac{1}{2} + \frac{1-2p}{2} \mathrm{sign}(Z) \Big) \Big] \\
&= \frac{1-2p}{\alpha^{3/2}} \, \mathbb{E}[|Z|] = \frac{1-2p}{\alpha\gamma} \, \mathbb{E}[|Z_0|] = \sqrt{\frac{2}{\pi}} \frac{1-2p}{\alpha} \,.
\end{aligned}
$$

To prove (57), note that under the GLM setting with sing link function, the true label of a features vector $\boldsymbol{x}$ is given by $y = \mathrm{sign}(\boldsymbol{x}^\mathsf{T}\boldsymbol{\beta})$, while the prediction by a model $\boldsymbol{\beta}^t$ are given by $\mathrm{sign}(\boldsymbol{x}^\mathsf{T}\boldsymbol{\beta}^t)$. Hence, $P_e(\boldsymbol{\beta}^t) = \mathbb{P}(\langle\boldsymbol{x},\boldsymbol{\beta}^t\rangle\langle\boldsymbol{x},\boldsymbol{\beta}\rangle < 0)$. Since $\boldsymbol{x} \sim \mathsf{N}(0, \boldsymbol{I}_d/n)$, letting $Z_1 = \sqrt{n}\langle\boldsymbol{x},\boldsymbol{\beta}\rangle/\|\boldsymbol{\beta}\|_{\ell_2}$ and $Z_2 = \sqrt{n}\langle\boldsymbol{x},\boldsymbol{\beta}^t\rangle/\|\boldsymbol{\beta}^t\|_{\ell_2}$, we have $(Z_1, Z_2) \sim \mathsf{N}(0, \begin{bmatrix} 1 & \rho \\ \rho & 1 \end{bmatrix})$ with $\rho = \frac{\langle\boldsymbol{\beta}^t,\boldsymbol{\beta}\rangle}{\|\boldsymbol{\beta}\|_{\ell_2}\|\boldsymbol{\beta}^t\|_{\ell_2}}$. Hence,

$$
P_e(\boldsymbol{\beta}^t) = \mathbb{P}(Z_1 Z_2 < 0) = \frac{1}{\pi} \cos^{-1}(\rho) \,.
$$

See e.g. [20, Section 15.10]. Using the result of Theorem 4.2, we have

$$
\rho \xrightarrow{p} \frac{\mu_t\gamma^2}{\gamma\sqrt{\mu_t^2\gamma^2 + \frac{\sigma_t^2}{\alpha}}} = \frac{\mu_t\gamma}{\sqrt{\mu_t^2\gamma^2 + \frac{\sigma_t^2}{\alpha}}} = \frac{\eta_t\gamma}{\sqrt{\eta_t^2\gamma^2 + \frac{1}{\alpha}}} = \frac{\eta_t}{\sqrt{\eta_t^2 + \frac{1}{\alpha}}} \,,
$$

since $\gamma = 1$. This completes the proof.

## K  Extension of Theory to the Multi-Class Case and Non-Linear Models

**Extension of theory to multi-class case.** This can be done by following similar ideas in the binary case, but will be heavier in notation and will involve more tedious analysis. Here we outline how this extension can be made for the GMM case in Section 3. We first describe the setting. We define a matrix $\boldsymbol{Y} \in \mathbb{R}^{n\times k}$, where $n$ is the number of samples and $k$ is the number of classes, using one-hot encoding, where column $i$ has 1 for the samples belonging to class $i$ and 0 otherwise. Next, we define $\boldsymbol{M} \in \mathbb{R}^{k\times d}$ where row $i$ is the mean feature vector of class $i$. Eq. (1) then transforms to $\boldsymbol{X} = \boldsymbol{YM} + \boldsymbol{Z}$, where $\boldsymbol{Z}$ is random noise; so this is a low rank matrix ($\boldsymbol{YM}$ is of rank $k$) plus noise. We can then generalize the AMP iterations for this low-rank model. The forms of eqs. (4) and (5) in this case will be the same with $g_t$ redefined appropriately for the matrix case (here it will be applied row-wise instead of coordinate-wise). Also the coefficient $C_t$ in (6) will now be a $k \times k$ matrix, defined similarly using the Jacobian of $g_t$ instead of derivative. Likewise, the state evolution recursion (8) will be of the same form, with the modification that $m_t, \sigma_t, \bar{m}_t, \bar{\sigma}_t$ are now vectors of length $k$. The insights from the analysis will be similar but the analysis will be on multi-dimensional objects. A similar extension can be made for GLMs in Section 4, where $\boldsymbol{\beta}$ should now be defined as a matrix of size $k \times d$. The extension here for the AMP iterates (17), (18) and the state evolution (22) are similar to the GMM case described above.

**Extension of theory to non-linear models.** We note that the AMP techniques can be used to analyze the statistical behavior of more general M-estimators (see for e.g., [14]). The corresponding update rule will involve applying the aggregator function (aggregating current predictions and the given noisy labels) as well as a gradient descent type of update (similar to eq. (4)). The exact details of these updates and the analysis will be quite tedious and are left for future work.

## L  Remaining Empirical Results

In Table 5, we list the average test accuracies of full RT, consensus-based RT, and BayesMix RT (28) after 1 and 10 iterations for Food-101 pho vs. ramen corrupted by the uniform noise model with $p = 0.45$. In Table 6, we perform an ablation study to compare these three RT methods at different values of $p$ for the same dataset. Note that BayesMix RT performs the best for larger values of $p$, i.e., in the high label noise regime, whereas consensus-based RT performs the best for smaller values of $p$.[6]

Table 5: **Linear probing and uniform noise ($p = 0.45$):** Average test accuracies ± standard deviation for Food-101 pho vs. ramen with $p = 0.45$. In the first iteration, consensus-based RT is the best, *but at the tenth iteration, BayesMix RT is the best by a big margin*.

| Iteration # | Full RT | Consensus-based RT | BayesMix RT (ours) |
|---|---|---|---|
| 0 (initial model) | 58.13 ± 2.54 | 58.13 ± 2.54 | 58.13 ± 2.54 |
| 1 | 61.33 ± 2.54 (3.20 ↑) | **63.60** ± 3.22 (**5.47** ↑) | 60.53 ± 3.27 (2.40 ↑) |
| 10 | 62.60 ± 2.12 (4.47 ↑) | 64.87 ± 4.07 (6.74 ↑) | **76.60** ± 6.00 (**18.47** ↑) |

Table 6: **Linear probing under uniform noise with different values of $p$:** Average test accuracies ± standard deviation for Food-101 pho vs. ramen after 10 iterations of full RT, consensus-based RT, and BayesMix RT. Observe that BayesMix RT performs the best for larger values of $p$, i.e., the more challenging high-noise regime, whereas consensus-based RT performs the best for smaller values of $p$.

| $p$ | Initial training | Full RT | Consensus-based RT | BayesMix RT (ours) |
|---|---|---|---|---|
| 0.45 | 58.13 ± 2.54 | 62.60 ± 2.12 | 64.87 ± 4.07 | **76.60** ± 6.00 |
| 0.40 | 67.53 ± 3.88 | 74.60 ± 6.57 | 79.87 ± 6.32 | **83.00** ± 1.30 |
| 0.35 | 73.20 ± 2.29 | 79.27 ± 4.34 | **86.00** ± 3.77 | 85.33 ± 0.50 |
| 0.30 | 79.47 ± 1.65 | 83.13 ± 3.55 | **87.33** ± 2.00 | 85.40 ± 0.57 |

**Adversarial noise model.** Now we consider a much harder noise model wherein an adversary gets to determine the "important" samples and flip their labels. The adversary determines important samples by training the same model which will be trained by us, *but with clean labels instead* and orders the samples in the decreasing order of the absolute value of the unnormalized logits. Note that a higher logit absolute value implies that the model is very confident on the sample, so flipping the label of such a sample would be more detrimental to the model's training as it would need to adjust the decision boundary by a large amount for this incorrectly labeled sample. The adversary then flips the labels of the top $\alpha$ fraction of the samples with largest absolute values of the unnormalized logits. Under this adversarial noise model with $\alpha = 0.25$ for MedMNIST Pneumonia, we compare BayesMix-Simple RT (29) with full RT and consensus-based RT after 1 and 10 iterations in Table 7 (again, the table below lists the average test accuracies).

Table 7: **Linear probing and adversarial noise:** Average test accuracies ± standard deviation for MedMNIST Pneumonia. *BayesMix-Simple RT performs the best*.

| Iteration # | Full RT | Consensus-based RT | BayesMix-Simple RT (ours) |
|---|---|---|---|
| 0 (initial model) | 63.94 ± 1.51 | 63.94 ± 1.51 | 63.94 ± 1.51 |
| 1 | 66.83 ± 1.93 (2.89 ↑) | 66.77 ± 1.78 (2.83 ↑) | **67.15** ± 1.83 (**3.21** ↑) |
| 10 | 73.77 ± 0.79 (9.83 ↑) | 76.76 ± 2.51 (12.82 ↑) | **78.04** ± 1.98 (**14.10** ↑) |

## M  Derivation of the Aggregator used in Experiments

Here we show the derivation of (28) used in our experiments. This is a straightforward extension of (32), (34) (in Appendix C) with means and variances of the Gaussians corresponding to the positive and negative logits being $(\mu_+, \sigma_+^2)$ and $(\mu_-, \sigma_-^2)$, respectively. Specifically, letting $z$ denote the logit

---

[6]Please note that real data may not necessarily follow the GMM setting in Section 3, so BayesMix RT being the optimal strategy is not necessary.

and $Y$ and $\widehat{Y}$ denote the ground truth label and observed label random variables, respectively, we have (following the derivation of (32)):

$$g(z, \widehat{y}) =$$

$$\frac{2\mathbb{P}\left(G = \frac{z-\mu_+}{\sigma_+}\right)\mathbb{P}\left(\widehat{Y} = \widehat{y}|Y = 1\right)\mathbb{P}(Y = 1)}{\mathbb{P}\left(G = \frac{z-\mu_+}{\sigma_+}\right)\mathbb{P}\left(\widehat{Y} = \widehat{y}\middle|Y = 1\right)\mathbb{P}(Y = 1) + \mathbb{P}\left(G = \frac{z-\mu_-}{\sigma_-}\right)\mathbb{P}\left(\widehat{Y} = \widehat{y}\middle|Y = -1\right)\mathbb{P}(Y = -1)} - 1. \quad (61)$$

(Notice that unlike (32), we dropped the dependence on $t$ to lighten the notation.) Again, $G \sim \mathsf{N}(0, 1)$ and $\mathbb{P}(Y = 1) = \pi_+$ and $\mathbb{P}(Y = -1) = \pi_-$. Also, under the uniform noise model, $\mathbb{P}\left(\widehat{Y} = \widehat{y}|Y = 1\right)$ and $\mathbb{P}\left(\widehat{Y} = \widehat{y}|Y = -1\right)$ are the same as (33). This can be extended to the non-uniform noise model as well, where $\mathbb{P}(\widehat{Y} = -1|Y = +1) = p$ and $\mathbb{P}(\widehat{Y} = +1|Y = -1) = q$, to get:

$$\mathbb{P}(\widehat{Y} = \widehat{y}|Y = 1) = (1 - p)^{\frac{1+\widehat{y}}{2}} p^{\frac{1-\widehat{y}}{2}} \quad \text{and} \quad \mathbb{P}(\widehat{Y} = \widehat{y}|Y = -1) = (1 - q)^{\frac{1-\widehat{y}}{2}} q^{\frac{1+\widehat{y}}{2}}. \quad (62)$$

Using all of this in (61) followed by some algebraic simplification gives us (28).

# N  Experimental Details

Here we provide details about our experiments.

Our experiments were done using TensorFlow and Keras on one 128 GB CPU and one 40 GB A100 GPU (per run). For initial training as well as for each iteration of retraining, the optimizer is Adam (with default values of $\beta_1 = 0.9$ and $\beta_2 = 0.999$) with batch size = 32 & number of epochs = 10 for linear probing and batch size = 128 & number of epochs = 2 for full network training.[7] We also apply weight decay = 0.1 in the case of full network training to mitigate overfitting.[8] We tune the learning rate by monitoring the accuracy on a small *clean* validation set (i.e., the labels of the validation set are not corrupted). For settings where we already know that the given labels will be noisy, we can manually clean up a small part of the dataset and use that as the validation set to mitigate overfitting. In each case, we first tune the learning rate for initial training; denote this by $\eta_0$. Then for each retraining method, we tune the learning rate for the first iteration and use it for all subsequent iterations; denote this by $\eta_1$.[9] Note that we use the same value of $\eta_0$ for all the retraining methods. Also recall that in the adversarial noise model described in Appendix L, the adversary initially trains the model with clean labels to determine the samples whose labels it wishes to flip. The adversary's training details are the same as described above; let us denote its learning rate, which is also tuned, by $\eta_{\mathrm{adv}}$. We tune $\eta_{\mathrm{adv}}$, $\eta_0$ and $\eta_1$ from $\{5 \times 10^{-3}, 10^{-3}, 5 \times 10^{-4}, 10^{-4}, 5 \times 10^{-5}, 10^{-5}, 5 \times 10^{-6}, 10^{-6}\}$.

1. **MedMNIST Pneumonia** (https://www.tensorflow.org/datasets/catalog/pneumonia_mnist): This has 4708 training examples and comes with a validation set of size 200. The test set consists of 624 examples.

2. **Food-101** (https://www.tensorflow.org/datasets/catalog/food101): Each class in Food-101 has 750 training examples; so the total number of examples for two classes (pho vs. ramen and spaghetti bolognese vs. spaghetti carbonara) is 1500. Out of these 1500 examples, we randomly select 100 examples as our validation set. The test set consists of 500 examples in total.

---

[7]Note that training an entire deep network is much more computationally expensive than linear probing.

[8]Note full network training is much more prone to overfitting than linear probing.

[9]We observed that one learning rate for initial training and retraining does not work well.

