# OpenReview forum: "Self-Boost via Optimal Retraining: An Analysis via Approximate Message Passing"
_NeurIPS.cc/2025/Conference — NeurIPS 2025 poster_

### Official Review · Reviewer_tTT7 · 2025-06-15

**Clarity:** 2
**Significance:** 3
**Originality:** 2
**Rating:** 4
**Confidence:** 4

**Summary:**

This paper addresses the problem of learning under noisy labels via retraining. Specifically, it investigates the optimal way to combine a model’s own predictions with noisy labels for binary classification, using the theoretical machinery of Approximate Message Passing (AMP). The authors derive the Bayes-optimal aggregator function for both Gaussian Mixture Models (GMMs) and Generalized Linear Models (GLMs), and analyze the iterative retraining dynamics through state evolution. They also propose a practical version for linear probing with binary cross-entropy loss, showing improved performance under high label noise.

**Questions:**

The authors claimed that this paper is the first work that addresses optimally using the predicted and given labels. To my knowledge, the following paper addresses how to combine the prediction and given labels to do label correction.
"SELC: self-ensemble label correction improves learning with noisy labels"

**Ethical Concerns:**

["NO or VERY MINOR ethics concerns only"]

**Final Justification:**

Thank you for the detailed response. The additional results and explanations have improved my understanding of the paper.

**Limitations:**

The analysis and evaluation both restricted to binary classification with linear models. Broader empirical validation is necessary.

**Quality:**

3

**Strengths And Weaknesses:**

Strengths:
Theoretical originality: The work provides the first principled derivation of optimal label aggregation in retraining using AMP. This paper addresses a key question on how to optimally combine labels and predictions.

Weaknesses:
The analysis is restricted to binary classification with linear models. No results are presented for multi-class settings or nonlinear deep networks.
The evaluation is limited to two binary classification datasets with artificially injected noise. Broader empirical validation (e.g., on naturally noisy datasets) is needed.
The paper is heavy on derivations and could be difficult to follow for non-experts. Additional intuition, figures, or explanations would aid understanding.

---

> ### Author Rebuttal · Authors · 2025-07-31
>
> Thanks for your review and feedback! You have raised good points and we respond to your concerns and questions below.
>
> **Extension of theory to multi-class case.** This can be done by following similar ideas in the paper for the binary case, but will be heavier in notation and will involve more tedious analysis. Here we outline how this extension can be made for the GMM case in Section 3. We first describe the setting. We define a label matrix $Y \in \mathbb{R}^{n \times k}$, where $n$ is the number of samples and $k$ is the number of classes, using one-hot encoding, where column $i$ has $1$ for the samples belonging to class $i$ and $0$ otherwise. Next, we define $M \in \mathbb{R}^{k \times d}$ where row $i$ is the mean feature vector of class $i$. Equation (1) then transforms to $X = YM+Z$, where $Z$ is random noise; so this is a low rank matrix ($YM$ is of rank $k$) plus noise. We can then generalize the AMP iterations for this low-rank model. The forms of eqs. (4) and (5) in this case will be the same with $g_t$ redefined appropriately for the matrix case (here it will be applied row-wise instead of coordinate-wise). Also the coefficient $C_t$ in line 126 will now be a $k \times k$ matrix, defined similarly using the Jacobian of $g_t$ instead of derivative. Likewise, the state evolution recursion (7) will be of the same form, with the modification that $m_t, \sigma_t, \bar{m_t}, \bar{\sigma}_t$ are now vectors of length $k$. The insights from the analysis will be similar but the analysis will be on multi-dimensional objects. A similar extension can be made for GLMs in Section 4, where $\beta$ should now be defined as a matrix of size $k \times d$. The extension here for the AMP iterates (16-17) and the state evolution (21) are similar to the GMM case described above.
>
> **Extension of theory to non-linear models.** We note that the AMP techniques can be used to analyze the statistical behavior of more general M-estimators (see for e.g., reference [14] in the paper). The corresponding update rule will involve applying the aggregator function (aggregating current predictions and the given noisy labels) as well as a gradient descent type of update (similar to eq. (4)). The exact details of these updates and the analysis will be quite tedious and are left for future work.
>
> **Extension in practice to different types of label noise (beyond uniform) and general ML models.** Although our theoretical results focused on linear models, the algorithm is Section 5 proposed for use in practice, i.e. BayesMix RT, is based on insights that carry over to general ML models. Specifically, eq. (27) is obtained by fitting a bimodal GMM on the distributions of the unnormalized logits (following the derivation in eq. (30) for theory). Following the same approach (and even in multi-class setting), we can rewrite the Bayesian posterior of the class memberships, after observing the current model's unnormalized logits with the prior based on the label noise model (in our paper, we have considered the uniform prior). We can then derive the corresponding aggregating function $g$ similar to eq. (27) in the paper. In short, *BayesMix RT can be used in deep learning models and even with non-uniform label noise priors* (e.g., when the flipping probability from class i to class j is some $p_{i, j}$). As an example, in the binary case, consider the **non-uniform noise model**: $\mathbb{P}(\hat{y} = -1| y = +1) = p$ and $\mathbb{P}(\hat{y} = +1| y = -1) = q$ with $p \neq q$. Then, eq. (27) changes to:
> $$g(z, \hat{y}) = \frac{2}{1+ \Big(\frac{q(1-q)}{p(1-p)}\Big) \Big(\frac{p q}{(1-p)(1-q)}\Big)^{\hat{y}/2} \exp\Big(\frac{(z - \mu_{+})^2}{2 \sigma_{+}^2} - \frac{(z - \mu_{-})^2}{2 \sigma_{-}^2}\Big) \frac{\pi_-}{\pi_+}} - 1.$$
> We show a result under the above **non-uniform noise model** with $p=0.45$ and $q=0.2$ for the MedMNIST Pneumonia dataset in the same setting as Section 5 using the formula above for BayesMix RT; the table below lists the average *test* accuracies $\pm$ standard deviation (over 3 runs) in this case. As we see below, **BayesMix RT performs the best**.
>
> |Iteration #| Full RT| Consensus-based RT| BayesMix RT (ours)|
> |---|---|---|---|
> |0 (initial model)|$71.85 \pm 1.18$|$71.85 \pm 1.18$|$71.85 \pm 1.18$|
> |1| $78.21 \pm 2.04$ | $77.51 \pm 1.92$ | $\textbf{81.14} \pm 0.46$ |
> |10| $81.62 \pm 2.16$ | $82.26 \pm 2.36$ | $\textbf{84.62} \pm 0.73$ |
>
> For more **general types of label noise** (e.g., sample-dependent noise) or in scenarios **where the noise model is not known**, we propose the following **simpler aggregation function** (instead of eq. (27)):
> $$g(z, \hat{y}) = \frac{2}{1+ \gamma^{\hat{y}} \exp\Big(\frac{(z - \mu_{+})^2}{2 \sigma_{+}^2} - \frac{(z - \mu_{-})^2}{2 \sigma_{-}^2}\Big) \frac{\pi_-}{\pi_+}} - 1,$$
> for some constant $\gamma \in (0,1)$ which can be tuned. We call this simpler version **BayesMix-Simple RT** (everything else is the same as BayesMix RT in the paper); in our implementation, we set $\gamma = 0.7$ throughout without any tuning. Also note that **this can be used with general ML models**.
>
> **(1)** To show the **efficacy of BayesMix-Simple RT**, we first consider an **adversarial noise model**. Specifically, an adversary gets to determine the "important" samples and flip their labels. The adversary determines the important samples by training the same model which will be trained by us *with clean labels* and orders the samples in the decreasing order of the absolute value of the unnormalized logits. Note that a higher logit absolute value implies that the model is very confident on the sample, so flipping the label of such a sample would be more detrimental to the model's training as it would need to adjust the decision boundary by a large amount for this incorrectly labeled sample. The adversary flips the labels of the top $\alpha$ fraction of the samples with largest absolute values of the unnormalized logits. Under this **adversarial noise model** (we don't assume knowledge of this process in training to be clear) with $\alpha=0.25$ for MedMNIST Pneumonia in the same setting as Section 5, we show the efficacy of **BayesMix-Simple RT** below compared to the baselines (again, the table below lists the average *test* accuracies $\pm$ standard deviation over 3 runs).
>
> |Iteration #| Full RT| Consensus-based RT| BayesMix-Simple RT (ours)|
> |---|---|---|---|
> |0 (initial model)|$63.94 \pm 1.51$|$63.94 \pm 1.51$|$63.94 \pm 1.51$|
> |1| $66.83 \pm 1.93$ | $66.77 \pm 1.78$ | $\textbf{67.15} \pm 1.83$ |
> |10| $73.77 \pm 0.79$ | $76.76 \pm 2.51$ | $\textbf{78.04} \pm 1.98$ |
>
> While we did not have time to find and work with naturally noisy datasets, we believe the above result under adversarial noise clearly demonstrates the strength of our method. But in the future, we will test our algorithm on naturally noisy datasets as you suggested.
>
> **(2)** We now show the **efficacy of BayesMix-Simple RT** even for **deep network training**. Recall that our earlier experiments are for linear probing with a pre-trained ResNet-50. Specifically, here we consider *training an **entire** pre-trained ResNet-50*. Because of the heavy computational cost of training an entire deep network, we only do 3 iterations of retraining here. We show results for two cases of Food-101:
>
> **i.** *Pho vs. Ramen (just like in the paper) corrupted by uniform noise with $p=0.4$*
> |Iteration #| Full RT| Consensus-based RT| BayesMix-Simple RT (ours)|
> |---|---|---|---|
> |0 (initial model)|$61.13 \pm 3.39$|$61.13 \pm 3.39$|$61.13 \pm 3.39$|
> |1| $59.07 \pm 5.44$ | $59.33 \pm 4.83$ | $\textbf{66.73} \pm 3.72$ |
> |3| $57.67 \pm 7.47$ | $58.47 \pm 6.78$ | $\textbf{70.67} \pm 4.65$ |
>
> **ii.** *Spaghetti Bolognese vs. Spaghetti Carbonara (not in the paper) corrupted by non-uniform noise described above with $p=0.45$ and $q=0.2$*
> |Iteration #| Full RT| Consensus-based RT| BayesMix-Simple RT (ours)|
> |---|---|---|---|
> |0 (initial model)|$65.45 \pm 2.22$|$65.45 \pm 2.22$|$65.45 \pm 2.22$|
> |1| $61.35 \pm 7.31$ | $ 61.30 \pm 4.48$ | $\textbf{74.55} \pm 3.75$ |
> |3| $56.50 \pm 7.74$ | $59.70 \pm 5.57$ | $\textbf{80.30} \pm 5.58$ |
>
> Note that in the above two tables (for entire ResNet-50 training), **BayesMix-Simple RT leads to significant gains** over the initial model, while Full RT and Consensus-based RT are actually detrimental.
>
> *So in summary, our proposed method is effective with different label noise models as well as for general ML models.*
>
> While we did not have enough time to concretely work on the multi-class case, *we believe our current contributions for binary classification are worthy and fundamental enough in their own right* and pave the way for the multi-class case in the future.
>
> We agree our paper is pretty dense and math-heavy. We will provide some intuition about the theoretical results in the next version. We encourage you to take a look at our discussion for "Interpretation of the Aggregator Function" in our response to Reviewer NMag (sorry we couldn't include it here due to lack of space).
>
> Thank you for pointing out the SELC paper; it is indeed relevant and we will cite it in the next version. However, please note that SELC is an empirical paper and it does not theoretically prove that their proposed method is optimal in any sense. On the other hand, our paper is primarily theoretical and our main goal is to derive the Bayes' optimal way to combine the given labels and the model's predictions for linear models.
>
> We will add all this discussion in the next version of our paper. We hope to have resolved your concerns and questions, but we're happy to discuss further if needed. If you're satisfied with our responses, **we sincerely hope you will raise your score**!

---

> > ### Author Response · Authors · 2025-08-05
> >
> > Dear Reviewer,
> >
> > The discussion period deadline is approaching and we haven't heard back from you. We'd appreciate it if you could go through our rebuttal and let us know if you have any other questions. Thank you!

---

> > ### Comment · Reviewer_tTT7 · 2025-08-09
> >
> > Thank you for the detailed response. The additional results and explanations have improved my understanding of the paper.
> > I hope these explanations will be clearly incorporated into the revised version.

---

### Official Review · Reviewer_NMag · 2025-06-26

**Clarity:** 3
**Significance:** 3
**Originality:** 3
**Rating:** 5
**Confidence:** 3

**Summary:**

**Summary**
This paper investigates how to optimally combine model predictions with noisy labels during iterative retraining in binary classification tasks. The authors develop a theoretical framework based on Approximate Message Passing (AMP) under both Gaussian Mixture Model (GMM) and Generalized Linear Model (GLM) settings. They derive the Bayes optimal aggregator function for label fusion and characterize the asymptotic performance via state evolution. A practical variant is also proposed for linear probing with cross-entropy loss, demonstrating empirical gains in high label noise regimes.

**Questions:**

1. Extension to Non-Uniform Noise Models:
Could the authors comment on the feasibility of extending the analysis to more realistic, non-uniform or instance-dependent label noise models? A discussion or simulation could help assess robustness beyond uniform noise.
2. Interpretation of the Aggregator Function:
The Bayes optimal aggregator is theoretically elegant, but can the authors provide more insight into its behavior (e.g., monotonicity, dependence on noise level, bias-variance trade-off)? This would help practitioners better understand when and how to use it.
3. Scalability and Runtime Considerations:
How computationally efficient is the AMP-based retraining approach compared to existing heuristics? A brief runtime comparison or complexity discussion would help readers assess its practicality for large-scale use.
4. Path Toward Non-Linear Models:
Given the growing use of deep models, can the authors elaborate on what specific challenges they foresee in extending their approach to non-linear architectures, and whether AMP or related techniques may still apply?

**Ethical Concerns:**

["NO or VERY MINOR ethics concerns only"]

**Final Justification:**

The authors have comprehensively addressed most of my concerns and conducted extensive empirical analyses to demonstrate the performance of their method. I think it is a technically solid paper, and thus I raise my score.

**Limitations:**

yes

**Paper Formatting Concerns:**

No formatting issues.

**Quality:**

3

**Strengths And Weaknesses:**

**Strengths**
1. Clean Theoretical Results: The paper provides a rigorous analysis of retraining procedures using AMP. The state evolution analysis provides transparent insight into how retraining affects generalization over iterations.
2. Optimal Aggregator Derivation: The derivation of a Bayes optimal aggregator for combining predictions and noisy labels is a significant theoretical contribution.
3. Empirical Gains: The theoretical insights are translated into a practical aggregator that can be used for linear probing to fine-tune deep neural networks. Empirical results show that this strategy performs particularly well in high label noise settings.

**Weakness**
1. Limited to Linear Models and Binary Classification: The current results are restricted to binary tasks and linear predictors. Extension to multi-class or non-linear models remains open.
2. Assumption of Uniform Label Noise: The theoretical analysis assumes uniform random label flips, which may not hold in many practical settings.
3. Lack of Theoretical Guarantees for Practical Variant:
While the proposed aggregator function is shown to work well in practice, the prediction error under this retraining strategy is not theoretically characterized, especially when the optimal aggregator function is estimated from data (e.g., it seems very difficult to obtain a consistent estimator for the optimal aggregator function for GLMs). A formal analysis of its generalization behavior or error bounds would strengthen the practical significance and reliability of the method.
4. Limited Empirical Evaluation: Experiments focus on synthetic and semi-synthetic settings. More diverse benchmarks would better support claims of practical relevance.

---

> ### Author Rebuttal · Authors · 2025-07-31
>
> Thanks for your review and feedback! You have raised great points and we address your questions below.
>
> **Extension in practice to different types of label noise (beyond uniform) and general ML models** (*Questions 1 and 4 & Weaknesses 1, 2 and 4*). Although our theoretical results focused on linear models, the algorithm is Sec. 5 proposed for use in practice, i.e. BayesMix RT, is based on insights that carry over to general ML models. Specifically, eq. (27) is obtained by fitting a bimodal GMM on the distributions of the unnormalized logits (following the derivation in eq. (30) for theory). Following the same approach, we can rewrite the Bayesian posterior of the class memberships, after observing the current model's unnormalized logits with the prior based on the label noise model (in our paper, we have considered the uniform prior). We can then derive the corresponding aggregating function $g$ similar to eq. (27) in the paper. In short, *BayesMix RT can be used in deep learning models and even with non-uniform label noise priors* (e.g., when the flipping probability from class i to j is some $p_{i, j}$). As an example, in the binary case, consider the **non-uniform noise model**: $\mathbb{P}(\hat{y} = -1| y = +1) = p$ and $\mathbb{P}(\hat{y} = +1| y = -1) = q$ with $p \neq q$. Then, eq. (27) changes to:
> $$g(z, \hat{y}) = \frac{2}{1+ \Big(\frac{q(1-q)}{p(1-p)}\Big) \Big(\frac{p q}{(1-p)(1-q)}\Big)^{\hat{y}/2} \exp\Big(\frac{(z - \mu_{+})^2}{2 \sigma_{+}^2} - \frac{(z - \mu_{-})^2}{2 \sigma_{-}^2}\Big) \frac{\pi_-}{\pi_+}} - 1.$$
> We show a result under the above **non-uniform noise model** with $p=0.45$ and $q=0.2$ for the MedMNIST Pneumonia dataset in the same setting as Sec. 5 using the formula above for BayesMix RT; the table below lists the average *test* accuracies $\pm$ std. (over 3 runs) in this case. Note **BayesMix RT performs the best**.
>
> |Iteration #| Full RT| Consensus-based RT| BayesMix RT (ours)|
> |---|---|---|---|
> |0 (initial model)|$71.85 \pm 1.18$|$71.85 \pm 1.18$|$71.85 \pm 1.18$|
> |1| $78.21 \pm 2.04$ | $77.51 \pm 1.92$ | $\textbf{81.14} \pm 0.46$ |
> |10| $81.62 \pm 2.16$ | $82.26 \pm 2.36$ | $\textbf{84.62} \pm 0.73$ |
>
> For more **general types of label noise** (e.g., sample-dependent noise) or in scenarios **where the noise model is not known**, we propose the following **simpler aggregation function** (instead of (27)):
> $$g(z, \hat{y}) = \frac{2}{1+\gamma^{\hat{y}} \exp\Big(\frac{(z-\mu_{+})^2}{2\sigma_{+}^2} - \frac{(z-\mu_{-})^2}{2\sigma_{-}^2}\Big) \frac{\pi_-}{\pi_+}} - 1,$$
> for some constant $\gamma \in (0,1)$ which can be tuned. We call this simpler version **BayesMix-Simple RT** (everything else is the same as BayesMix RT in the paper); in our implementation, we set $\gamma=0.7$ throughout w/o any tuning. Also note that **this can be used with general ML models**.
>
> **(1)** To show the **efficacy of BayesMix-Simple RT**, we first consider an **adversarial noise model**. Specifically, an adversary gets to determine the "important" samples and flip their labels. The adversary determines important samples by training the same model which will be trained by us *with clean labels* and orders the samples in the decreasing order of the absolute value of the unnormalized logits. Note that a higher logit absolute value implies that the model is very confident on the sample, so flipping the label of such a sample would be more detrimental to the model's training as it would need to adjust the decision boundary by a large amount for this incorrectly labeled sample. The adversary flips the labels of the top $\alpha$ fraction of the samples with largest absolute values of the unnormalized logits. Under this **adversarial noise model** (we don't assume knowledge of this process in training to be clear) with $\alpha=0.25$ for MedMNIST Pneumonia in the same setting as Sec. 5, we show the efficacy of **BayesMix-Simple RT** below compared to the baselines (table below lists the average *test* accuracies $\pm$ std. over 3 runs).
>
> |Iteration #| Full RT| Consensus-based RT| BayesMix-Simple RT (ours)|
> |---|---|---|---|
> |0 (initial model)|$63.94\pm1.51$|$63.94\pm1.51$|$63.94\pm1.51$|
> |1| $66.83\pm1.93$ | $66.77\pm1.78$ | $\textbf{67.15}\pm1.83$ |
> |10| $73.77\pm0.79$ | $76.76\pm2.51$ | $\textbf{78.04}\pm1.98$ |
>
> **(2)** We now show the **efficacy of BayesMix-Simple RT** even for **deep network training**. Recall that our earlier experiments are for linear probing with a pre-trained ResNet-50. Specifically, here we consider *training an **entire** pre-trained ResNet-50*. Because of the heavy computational cost of training an entire ResNet-50, we only do 3 iterations of retraining here. We show results for two cases of Food-101:
>
> **i.** *Pho vs. Ramen corrupted by uniform noise with $p=0.4$*
> |Iteration #| Full RT| Consensus-based RT| BayesMix-Simple RT (ours)|
> |---|---|---|---|
> |0 (initial model)|$61.13\pm3.39$|$61.13\pm3.39$|$61.13\pm3.39$|
> |1| $59.07\pm5.44$ | $59.33\pm4.83$ | $\textbf{66.73}\pm3.72$ |
> |3| $57.67\pm7.47$ | $58.47\pm6.78$ | $\textbf{70.67}\pm4.65$ |
>
> **ii.** *Spaghetti Bolognese vs. Spaghetti Carbonara corrupted by non-uniform noise described above with $p=0.45$ and $q=0.2$*
> |Iteration #| Full RT| Consensus-based RT| BayesMix-Simple RT (ours)|
> |---|---|---|---|
> |0 (initial model)|$65.45\pm2.22$|$65.45\pm2.22$|$65.45\pm2.22$|
> |1| $61.35\pm7.31$ | $ 61.30\pm4.48$ | $\textbf{74.55}\pm3.75$ |
> |3| $56.50\pm7.74$ | $59.70\pm5.57$ | $\textbf{80.30}\pm5.58$ |
>
> Note that in the above two tables (for entire ResNet-50 training), **BayesMix-Simple RT leads to significant gains** over the initial model, while Full RT and Consensus-based RT are actually detrimental.
>
> *So in summary, our proposed method is effective with different label noise models as well as for general ML models.*
>
> **Extension of theory to non-linear models** (*Question 4 & Weakness 1*). We note that the AMP techniques can be used to analyze the statistical behavior of more general M-estimators (see for e.g., reference [14] in the paper). The corresponding update rule will involve applying the aggregator function (aggregating current predictions and the given noisy labels) as well as a gradient descent type of update (similar to eq. (4)). The exact details of these updates and the analysis will be quite tedious and are left for future work.
>
> **Extension of theory to multi-class case** (*Weakness 1*). This can be done by following similar ideas in the paper for the binary case, but will involve more tedious analysis. Here we outline how this extension can be made for the GMM case in Sec. 3. We first describe the setting. We define a matrix $Y \in \mathbb{R}^{n \times k}$, where $n$ is the # of samples and $k$ is the # of classes, using one-hot encoding, where column $i$ has $1$ for the samples belonging to class $i$ and $0$ otherwise. Next, we define $M \in \mathbb{R}^{k \times d}$ where row $i$ is the mean feature vector of class $i$. Eq. (1) then transforms to $X = YM+Z$, where $Z$ is random noise; so this is a low rank matrix ($YM$ is of rank $k$) plus noise. We can then generalize the AMP iterations for this low-rank model. The forms of eqs. (4) & (5) in this case will be the same with $g_t$ redefined appropriately for the matrix case (here it will be applied row-wise instead of coordinate-wise). Also the coefficient $C_t$ in line 126 will now be a $k \times k$ matrix, defined similarly using the Jacobian of $g_t$ instead of derivative. Likewise, the state evolution recursion (7) will be of the same form, with the modification that $m_t, \sigma_t, \bar{m_t}, \bar{\sigma}_t$ are now vectors of length $k$. The insights from the analysis will be similar but the analysis will be on multi-dimensional objects. A similar extension can be made for GLMs in Sec. 4, where $\beta$ should now be defined as a matrix of size $k \times d$. The extension here for the AMP iterates (16-17) and the state evolution (21) are similar to the GMM case described above.
>
> **Interpretation of the Aggregator Function** (*Question 2*). Consider the aggregator in eq. (15) for the GMM case (similar insights hold for GLM and BayesMix RT). Recall that $g(y,\hat{y})$ is the soft label obtained based on the soft prediction $y$ and the given noisy label $\hat{y} \in$ {$+1,-1$}. It is easy to see that $g(y,\hat{y})\in[-1,1]$, it is monotonic increasing in $y$ and $\hat{y}$ (since $p<1/2$). Also it is increasing in $\pi_+/\pi_-$, accounting for class probabilities (so if $\pi_+>\pi_-$, our aggregated labels are biased towards the positive class). Also note the denominator can be rewritten as $1+\exp\Big(-\log(\frac{1-p}{p})\hat{y} - 2y\Big) \frac{\pi_-}{\pi_+}$; so the term inside the exponential is a negative linear combination of $\hat{y}$ and $y$. Now if the label flipping probability $p$ is small, we put a larger weight on the given labels $\hat{y}$ relative to the predictions $y$, but as $p$ gets larger we reduce the weight on $\hat{y}$, thereby relying more on the predictions $y$. These behavior trends of the aggregator are consistent with what we expect intuitively.
>
> **Scalability and Runtime Considerations** (*Question 3*). We focus on computational overhead of BayesMix(-Simple) RT as that's used in practice. The major extra cost compared to Full RT & Consensus RT is in fitting the bimodal GMM to obtain the aggregation function $g$. But this is negligible when compared to the training (backward & forward propagation) cost of every epoch, especially for large models. So our method doesn't introduce too much extra overhead. Note that even the AMP update rules for theory in Sec. 3 & 4 don’t introduce much extra overhead; the cost is dominated by the 2 matrix-vector multiplications involving $X$ and $X^\top$.
>
> Theory for BayesMix RT is left for future work.
>
> We hope to have resolved your concerns and questions, but we're happy to discuss further if needed. If you're satisfied with our responses, **we hope you will raise your score**!

---

> > ### Comment · Reviewer_NMag · 2025-08-01
> > **Response to Authors**
> >
> > The authors have comprehensively addressed most of my concerns and conducted extensive empirical analyses to demonstrate the performance of their method. However, I remain unclear about the practical implications of this work, as my question—"Path Toward Non-Linear Models: Given the increasing adoption of deep models, can the authors elaborate on the specific challenges they anticipate in extending their approach to non-linear architectures, and whether AMP or related techniques remain applicable?"—was not fully answered. Consequently, I am inclined to maintain my current score. I do not expect the authors to provide a detailed theoretical explanation of how their existing results could be extended to non-linear cases or deep neural networks. Instead, they could cite relevant literature and discuss how existing AMP methods have informed the practical application of non-linear models.

---

> ### Author Response · Authors · 2025-08-01
> **Path Toward Non-Linear Models: Applicability of AMP and challanges**
>
> Thanks for engaging in discussion! In our response “Extension in practice to different types of noise and general ML models”, we discussed how BayesMix RT and its underlying intuition (namely fitting a gaussian mixture model to soft predictions at each step and using Bayes' optimal estimates) can be used for general ML models. To that end, **we also presented empirical results for deep learning in our rebuttal above** (see the last two tables for Pho vs. Ramen and Spaghetti Bolognese vs. Spaghetti Carbonara). Here we focus our response on AMP techniques, its applicability to nonlinear models and the challenges.
>
> AMP techniques can be used to analyze the statistical behavior of the so-called **“general first order methods (GFOM)”**.  Consider estimation problems where the data is given as a random matrix $X$ and the goal is to estimate an unknown model parameter. These algorithms keep state sequences of vectors (or in more general form matrices) $u^1,\dotsc, u^t$ and $v^1,\dotsc, v^t$ which are updated by two types of operations: row-wise application of a (possibly **non-linear**) function, or multiplication by $X$ or $X^\top$. This cover a broad range of important practical problems, for example classical first order optimization (for convex and non-convex objectives), such as gradient descent and its accelerated versions where the loss is a function of $X\theta$ (note that the gradient updates here become $X^\top \nabla L(X\theta, y)$ and follows the above GFOM form). Other examples include sparse PCA, low rank matrix estimation,  phase retrieval, nonlinear power methods, finding planted hidden clique, among others (**see references below**). Each of these problems is an important application which has attracted an immense interest and significant research effort. **So in short AMP provides a very powerful class of iterative algorithms, for non-linear problems, and comes with an asymptotic *"precise"* statical characterization.** That said, it cannot be applied to general deep NNet architecture, for which the updates do not follow the form of general first order methods, discussed above.
>
> To outline the challenges for using these techniques (in its general form) for our problem, we should note that in AMP algorithms (in supervised learning), the given labels are fixed across iterations and can be used in the nonlinear function updates. However, here we iteratively use the current model to make predictions and then aggregate them with the initially given noisy labels to retrain the model. This complication requires new derivations, which we did for two data models: Gaussian Mixture model and generalized linear model, with linear classifier. Given the success of AMP in other nonlinear setting we expect the derivations to be extendable to other nonlinear classifiers, but a rigorous study is out of the scope of the current work.
>
> We hope that we have addressed your concerns. Please let us know if you have any other question!
>
> [1] The estimation error of general first order methods, M Celentano, A Montanari, Y Wu, Conference on Learning Theory, 2020
>
> [2] Information-theoretically optimal sparse PCA, Y Deshpande, A Montanari, IEEE International Symposium on Information Theory, 2014
>
> [3] Finding Hidden Cliques of Size $\sqrt{n/e}$ in Nearly Linear Time, Y Deshpande, A Montanari, Foundations of Computational Mathematics 15 (4), 1069-1128
>
> [4] Statistically optimal first order algorithms: a proof via orthogonalization, A Montanari, Y Wu, Information and Inference: A Journal of the IMA, 2024
>
> [5] A unifying tutorial on 342 approximate message passing, O. Y. Feng, R. Venkataramanan, C. Rush, R. J. Samworth, Foundations and Trends in Machine Learning, 15(4):335–536, 343 2022.
>
> [6] Optimization-Based AMP for Phase Retrieval: The Impact of Initialization and $\ell_2$ Regularization, J Ma, J Xu, A Maleki
> IEEE Transactions on Information Theory, 2019

---

> > ### Author Response · Authors · 2025-08-05
> >
> > Dear Reviewer,
> >
> > Thanks a lot for engaging in discussion. We'd appreciate if you could go over our latest response and let us know if that answers your question. Thank you!

---

### Official Review · Reviewer_Pm7K · 2025-07-03

**Clarity:** 3
**Significance:** 4
**Originality:** 4
**Rating:** 5
**Confidence:** 5

**Summary:**

This paper tackles the open question of how to optimally combine a model’s own predictions with the original (potentially noisy) labels during retraining. Focusing on binary classification, the authors develop a theoretical framework based on Approximate Message Passing (AMP) to analyze iterative self-retraining for two settings: a Gaussian Mixture Model (GMM) and a Generalized Linear Model (GLM).
The main contributions are: Bayes-Optimal Aggregator: Derivation of the Bayes-optimal aggregator function gt that combines the model’s current predicted label and the given noisy label at each retraining round such that retraining on this combination minimizes the model’s prediction error. To our knowledge, this is the first work providing a theoretically optimal way to mix predicted and given labels in retraining, extending prior retraining methods (which used heuristics like “full” or “consensus” retraining) by rigorously identifying the optimal mixture. AMP-based Analysis of Multiple Retraining Rounds: Introduction of an iterative retraining scheme formulated as an AMP algorithm. The authors derive a state evolution recursion that exactly characterizes the asymptotic behavior (as data dimension and sample size grow proportionally) of the model’s accuracy over successive retraining rounds. Using this analysis, they quantify how the optimal retraining strategy performs over multiple iterations, and they prove theoretical results such as Theorem 3.1 (asymptotic Gaussianity of model estimates via AMP) and Theorem 3.2/4.3 (formulas for the optimal aggregator in GMM/GLM).
Key Theoretical Insights: They show that using the Bayes-optimal aggregator strictly improves the model’s signal-to-noise ratio each round (when the model is not already near-perfect) and dominates earlier retraining schemes. A notable insight is that retraining is not universally beneficial: if the initial model is very strong, further retraining can hurt performance, whereas if the initial model is weaker, retraining helps (they formalize a condition for this crossover).
Practical Method and Experiments: Building on the theory, the authors propose a practical version of the optimal aggregator for real data. In particular, for a pretrained model fine-tuned with a linear probe, they use a bimodal GMM fit on the model’s logit distribution to approximate the Bayes-optimal combination of labels. This method, termed “BayesMix RT”, is evaluated on image classification tasks with artificial label noise. In experiments on a medical imaging dataset and a Food-101 subset, BayesMix RT significantly outperforms two baseline retraining strategies from prior work (full retraining and consensus-based retraining) under high label noise (45% flipped labels). For instance, after 10 retraining iterations on one task, BayesMix RT reached ~76.6% accuracy vs ~64–65% with the baselines, demonstrating a substantial performance gain in noisy conditions.
The paper’s contributions are thoroughly evaluated through theoretical proofs (with detailed appendices) and simulation/empirical results, providing a holistic understanding of optimal self-retraining.

**Questions:**

The work is restricted to binary class labels. How challenging do the authors expect it to be to extend the Bayes-optimal aggregator derivation to a multi-class setting?  Could the authors elaborate on what the optimal aggregator might look like for $K>2$ classes?

The optimal aggregator formulas (e.g., Eq. (13) for GMM) explicitly depend on the label flipping probability $p$ (and class priors). In practice, $p$ might not be known exactly. Did the authors experiment with estimating $p$ from data or treating it as a tuned parameter? How sensitive is BayesMix RT to mis-specification of $p$? It would be useful to know if using a slightly wrong noise rate (or an adaptive estimate of it) significantly hurts performance or not. Similarly, the practical method requires fitting a GMM to logits and if this fit is imperfect (e.g. if the logit distribution is not a clean mixture of two Gaussians), does it degrade the results? Clarifying the robustness of the approach to these assumptions would help practitioners know how to apply it.

The theory (Proposition 3.3) indicates a scenario where retraining can degrade performance if the initial model’s accuracy is above a certain threshold relative to noise. This is a compelling insight. Could the authors provide more intuition or empirical confirmation for this phenomenon? In the experiments presented, BayesMix RT always improved accuracy over 10 iterations, even when starting from a reasonably good initial model (e.g., ~79% in Table 5 for p=0.30). Did the authors observe any instance where continuing to retrain actually made things worse (for either their method or the baselines)? It would be useful to know, for example, if one should stop retraining after a certain point.

The empirical demonstration is for retraining only the last layer of a neural network. Do the authors have thoughts on extending BayesMix retraining to fine-tuning an entire deep network? In principle, one could use the model’s predictions (or soft probabilities) in the loss for all layers. Would the aggregator still be applicable or would new complications arise (e.g., the model’s feature representation shifting while we alter labels)? This is related to the assumption in GLM analysis that features are fixed i.i.d. random vectors. How might one integrate the optimal label aggregation into an end-to-end training pipeline?

The paper assumes uniform random noise (each label flipped with the same probability p). If the noise were class-dependent (each class has its own flip probability) or feature-dependent (hard examples more likely mislabeled), the current aggregator might not directly apply. In such cases, do the authors expect the strategy to change?

**Ethical Concerns:**

["NO or VERY MINOR ethics concerns only"]

**Final Justification:**

The authors' rebuttal has thoroughly addressed my concerns on multi-class extension, handling unknown p (via estimation and a simple p-independent variant), robustness to GMM fitting assumptions, and empirical confirmation of when retraining degrades performance (e.g., in full ResNet-50 experiments where baselines hurt but BayesMix-Simple RT helps). The new results on non-uniform and adversarial noise models further demonstrate the method's versatility beyond uniform noise, and the adaptation ideas for end-to-end deep training and complex noise priors suggest promising future extensions.

These responses reinforce the paper's strengths in deriving a theoretically optimal aggregator, providing insightful analysis via AMP, and bridging to practice with BayesMix RT. The authors have done a good job with the rebuttal, resolving all points raised. This remains a technically solid paper with high impact in theoretical ML and label noise handling, excellent evaluation (now bolstered by rebuttal experiments), and no ethical issues. I stand by my original accept recommendation.

**Limitations:**

The authors have adequately addressed the limitations of their work and potential societal impacts. They explicitly discuss key limitations in Section 6 (second paragraph): notably, the restriction to linear models, binary classification, and uniform noise. They frame these as avenues for future work, indicating awareness that the method needs extension to handle multi-class data and more complex noise distributions. They also acknowledge that their current practical method is tailored to linear probing and that extending to nonlinear deep models is a future goal. This openness about scope is commendable and readers are clearly informed about where the results apply or not which is good practice.

One limitation not deeply discussed (but worth reflecting on) is the assumption of knowing the noise level $p$. In practice how would one estimate it.

**Paper Formatting Concerns:**

The paper follows the NeurIPS style well. I did not notice any major formatting issues. Figures and tables are legible and referenced in the text appropriately. The mathematical notation is standard and the equations are properly numbered and aligned. All fonts, margins, and spacing appear correct.

**Quality:**

4

**Strengths And Weaknesses:**

The paper answers a clear open question - what is the optimal way to mix predicted and given labels during retraining? They obtain this  by deriving the Bayes-optimal solution for that mixture. This is a notable advance since prior works on label-noise retraining either retrained on predicted labels alone or used heuristics (e.g. ignore or trust labels based on agreement) but did not determine an optimal combination rule. The authors’ use of AMP theory to derive a closed-form aggregator (e.g. Equation (13) for GMM) is elegant and appears novel.

The paper is technically solid. It provides precise asymptotic analysis of the retraining dynamics via state evolution equations (Theorem 3.1, 4.2) and proves the optimality of their proposed aggregator (Theorem 3.2, 4.3) with all assumptions clearly stated. The proofs are given in a detailed appendix, and the authors reference where each result is proven. The analytical work is thorough. For instance, authors even discuss how to handle non-Lipschitz aggregator functions via approximation to satisfy AMP conditions.  This level of rigor is required for  high quality paper in the theory development.

The theoretical results yield interesting insights into when retraining helps vs hurts. Proposition 3.3, for instance, shows that if the initial model’s accuracy is below a certain threshold (related to a fixed point of their state evolution), each retraining round will improve accuracy, but if it’s above that threshold, retraining could degrade accuracy. This explains and quantifies a phenomenon that practitioners might have observed anecdotally. The paper also neatly recovers prior strategies as special cases – e.g., “full retraining” corresponds to using only model predictions.

Despite the heavy theory, the authors connect to practice by devising the BayesMix RT method for training a linear probe with noisy labels. This method is directly inspired by their theory (essentially implementing a softened version of the theoretical aggregator using a fitted mixture model). The experiments, though limited in scope (two datasets), demonstrate significant empirical gains in extremely noisy scenarios. This suggests the theory has practical merit. Furthermore, the authors conducted simulations (Appendix D) to verify that their state evolution predictions match finite-sample behavior and that the optimal aggregator improves over simpler retraining approaches.

---

> ### Author Rebuttal · Authors · 2025-07-31
>
> Thanks for your detailed review and positive assessment of our work! You have raised great points and we address your questions and concerns below.
>
> **Extension of theory to multi-class case.** This can be done by following similar ideas in the paper for the binary case, but will involve more tedious analysis. Here we outline how this extension can be made for the GMM case in Section 3. We first describe the setting. We define a label matrix $Y \in \mathbb{R}^{n \times k}$, where $n$ is the # of samples and $k$ is the # of classes, using one-hot encoding, where column $i$ has $1$ for the samples belonging to class $i$ and $0$ otherwise. Next, we define $M \in \mathbb{R}^{k \times d}$ where row $i$ is the mean feature vector of class $i$. Eq. (1) then transforms to $X = YM+Z$, where $Z$ is random noise; so this is a low rank matrix ($YM$ is of rank $k$) plus noise. We can then generalize the AMP iterations for this low-rank model. The forms of eqs. (4) and (5) in this case will be the same with $g_t$ redefined appropriately for the matrix case (here it will be applied row-wise instead of coordinate-wise). Also the coefficient $C_t$ in line 126 will now be a $k \times k$ matrix, defined similarly using the Jacobian of $g_t$ instead of derivative. Likewise, the state evolution recursion (7) will be of the same form, with the modification that $m_t, \sigma_t, \bar{m_t}, \bar{\sigma}_t$ are now vectors of length $k$. The insights from the analysis will be similar but the analysis will be on multi-dimensional objects. A similar extension can be made for GLMs in Section 4, where $\beta$ should now be defined as a matrix of size $k \times d$. The extension here for the AMP iterates (16-17) and the state evolution (21) are similar to the GMM case described above.
>
> **Dealing with unknown label flip probability $p$.** Great question! We propose the following way to estimate $p$. We focus on the samples for which the model is the most confident after training with the given noisy labels. Specifically, we pick the set of samples $S_\text{conf}$ with the highest $\rho$ (this should be small) fraction of absolute value of unnormalized logits; a higher value indicates higher confidence. The philosophy of self-training based ideas is that the model is probably correct on the samples for which it is very confident; so we treat the model's predicted labels for the samples in $S_\text{conf}$ as the ground truth labels and calculate the % of samples in $S_\text{conf}$ for which the predicted label matches the given label. This gives us an estimate for $p$. We tested this approach for MedMNIST Pneumonia dataset in the same setting as Sec. 5; here are the results averaged over 3 runs.
>
> |Actual $p$| Estimated $p$ (mean $\pm$ std.)|
> |---|---|
> |0.45 |$0.41\pm0.01$|
> |0.30| $0.26\pm0.02$ |
> |0.15| $0.12\pm0.01$ |
>
> So this gives us a good estimate. We also saw that using the estimated $p$'s above basically didn't change the results at all. Noting this and also taking into account that the estimated $p$ may be off by a lot sometimes, we tested the following **simpler aggregator function** (as a substitute for eq. (27)) *independent of* $p$:
> $$g(z, \hat{y}) = \frac{2}{1+ \gamma^{\hat{y}} \exp\Big(\frac{(z - \mu_{+})^2}{2 \sigma_{+}^2} - \frac{(z - \mu_{-})^2}{2 \sigma_{-}^2}\Big) \frac{\pi_-}{\pi_+}} - 1,$$
> for some constant $\gamma \in (0,1)$ which can be tuned. Note that $\gamma$ essentially plays the role of $p/(1-p)$. We call this simpler version **BayesMix-Simple RT** (everything else is the same as BayesMix RT in the paper); in our implementation, we set $\gamma = 0.7$ throughout w/o any tuning. We compare test accuracies (mean $\pm$ std. averaged over 3 runs) for MedMNIST Pneumonia obtained after 10 iterations of Full RT, Consensus RT, BayesMix RT with the exact $p$ and BayesMix-Simple RT (with $\gamma = 0.7$).
>
> |$p$|Initial Model|Full RT| Consensus RT| BayesMix RT with exact $p$| BayesMix-Simple RT|
> |---|---|---|---|---|---|
> |0.45|$64.58\pm3.07$|$68.06\pm3.78$|$70.03\pm0.73$|$71.42\pm2.43$|$72.38\pm1.66$|
> |0.30|$78.42\pm2.15$|$78.85\pm3.14$|$82.16\pm1.63$|$84.82\pm0.89$|$84.67\pm1.58$|
> |0.15|$81.41\pm0.47$|$82.10\pm1.74$|$84.88\pm1.29$|$87.29\pm1.40$|$85.90\pm0.33$|
>
> Notice that *BayesMix-Simple RT is competitive with BayesMix RT* even though it does not use the value of $p$.
>
> Regarding your question about fitting a GMM to the logits, note that in reality the distribution is unlikely to be a GMM exactly. Yet our method consistently leads to good performance (we show results with more noise models below).
>
> **Extension in practice to different types of label noise and general ML models**. BayesMix RT is based on insights that carry over to general ML models. Specifically, eq. (27) is obtained by fitting a bimodal GMM on the distributions of the unnormalized logits (following the derivation in eq. (30) for theory). Following the same approach, we can rewrite the Bayesian posterior of the class memberships, after observing the current model's unnormalized logits with the prior based on the label noise model. We can then derive the corresponding aggregating function $g$ similar to eq. (27) in the paper. In short, *BayesMix RT can be used in deep learning models and even with non-uniform label noise priors* (e.g., when the flipping probability from class i to class j is some $p_{i, j}$). As an example, in the binary case, consider the **non-uniform noise model**: $\mathbb{P}(\hat{y} = -1| y = +1) = p$ and $\mathbb{P}(\hat{y} = +1| y = -1) = q$ with $p \neq q$. Then, eq. (27) changes to:
> $$g(z, \hat{y}) = \frac{2}{1+ \Big(\frac{q(1-q)}{p(1-p)}\Big) \Big(\frac{p q}{(1-p)(1-q)}\Big)^{\hat{y}/2} \exp\Big(\frac{(z - \mu_{+})^2}{2 \sigma_{+}^2} - \frac{(z - \mu_{-})^2}{2 \sigma_{-}^2}\Big) \frac{\pi_-}{\pi_+}} - 1.$$
> We show a result under the above **non-uniform noise model** with $p=0.45$ and $q=0.2$ for the MedMNIST Pneumonia dataset in the same setting as Sec. 5 using the formula above for BayesMix RT; the table below lists the average *test* accuracies $\pm$ std. (over 3 runs) in this case. As we see below, **BayesMix RT performs the best**.
>
> |Iteration #| Full RT| Consensus RT| BayesMix RT (ours)|
> |---|---|---|---|
> |0 (initial model)|$71.85 \pm 1.18$|$71.85 \pm 1.18$|$71.85 \pm 1.18$|
> |1| $78.21 \pm 2.04$ | $77.51 \pm 1.92$ | $\textbf{81.14} \pm 0.46$ |
> |10| $81.62 \pm 2.16$ | $82.26 \pm 2.36$ | $\textbf{84.62} \pm 0.73$ |
>
> For **more general types of label noise** or **in scenarios where the noise model is not known**, we propose to use the **BayesMix-Simple RT** algorithm described above as it is independent of any noise distribution parameter and is competitive with BayesMix RT in the uniform noise case. Also note that **this can be used with general ML models**.
>
> **(1)** To show the **efficacy of BayesMix-Simple RT**, we first consider an **adversarial noise model**. Specifically, an adversary gets to determine the "important" samples and flip their labels. The adversary determines important samples by training the same model which will be trained by us *with clean labels* and orders the samples in the decreasing order of the absolute value of the unnormalized logits. Note that a higher logit absolute value implies that the model is very confident (and probably correct) on the sample, so flipping the label of such a sample would be more detrimental to the model's training as it would need to adjust the decision boundary by a large amount for this incorrectly labeled sample. The adversary flips the labels of the top $\alpha$ fraction of the samples with largest absolute values of the unnormalized logits. Under this **adversarial noise model** (we don't assume knowledge of this process in training to be clear) with $\alpha=0.25$ for MedMNIST Pneumonia in the same setting as Sec. 5, we show the efficacy of **BayesMix-Simple RT** below compared to the baselines (table below lists average *test* accuracies $\pm$ std. over 3 runs).
>
> |Iteration #| Full RT| Consensus RT| BayesMix-Simple RT (ours)|
> |---|---|---|---|
> |0 (initial model)|$63.94\pm1.51$|$63.94\pm1.51$|$63.94\pm1.51$|
> |1| $66.83\pm1.93$ | $66.77\pm1.78$ | $\textbf{67.15}\pm1.83$ |
> |10| $73.77\pm0.79$ | $76.76\pm2.51$ | $\textbf{78.04}\pm1.98$ |
>
> **(2)** We now show the **efficacy of BayesMix-Simple RT** for **deep network training**. Recall that our earlier experiments are for linear probing with a pre-trained ResNet-50. Specifically, here we consider *training an **entire** pre-trained ResNet-50*. Because of the heavy computational cost of training an entire ResNet-50, we only do 3 iterations of retraining here. We show results for 2 cases of Food-101:
>
> **i.** *Pho vs. Ramen corrupted by uniform noise with $p=0.4$*
> |Iteration #| Full RT| Consensus RT| BayesMix-Simple RT (ours)|
> |---|---|---|---|
> |0 (initial model)|$61.13\pm3.39$|$61.13\pm3.39$|$61.13\pm3.39$|
> |1| $59.07\pm5.44$ | $59.33\pm4.83$ | $\textbf{66.73}\pm3.72$ |
> |3| $57.67\pm7.47$ | $58.47\pm6.78$ | $\textbf{70.67}\pm4.65$ |
>
> **ii.** *Spaghetti Bolognese vs. Spaghetti Carbonara corrupted by non-uniform noise described above with $p=0.45$ and $q=0.2$*
> |Iteration #| Full RT| Consensus RT| BayesMix-Simple RT (ours)|
> |---|---|---|---|
> |0 (initial model)|$65.45\pm2.22$|$65.45\pm2.22$|$65.45\pm2.22$|
> |1| $61.35\pm7.31$ | $ 61.30\pm4.48$ | $\textbf{74.55}\pm3.75$ |
> |3| $56.50\pm7.74$ | $59.70\pm5.57$ | $\textbf{80.30}\pm5.58$ |
>
> Note that in the above two tables (for entire ResNet-50 training), **BayesMix-Simple RT leads to significant gains** over the initial model, while Full RT and Consensus RT are actually detrimental; **the latter also answers your question about whether retraining can hurt performance in some cases**. It is possible that BayesMix-Simple RT hurts performance in some cases.
>
> *But in summary, our proposed method is effective with different label noise models as well as for general ML models.*
>
> We hope to have resolved your concerns and questions, but we're happy to discuss further if needed.

---

> > ### Author Response · Authors · 2025-08-05
> >
> > Dear Reviewer,
> >
> > The discussion period deadline is approaching and we haven't heard back from you. We'd appreciate it if you could go through our rebuttal and let us know if you have any other questions. Thank you!

---

### Official Review · Reviewer_ZTCf · 2025-07-03

**Clarity:** 2
**Significance:** 2
**Originality:** 3
**Rating:** 5
**Confidence:** 3

**Summary:**

Previous work has proved the benefits of retraining, but few work have answered what is the optimal combination of the model's predictions and the provided labels. This paper addresses the fundamental question for binary classification tasks. They develop a framework based on approximate message passing to make an analysis for GMM and GLM. The contribution is the derivation of the bayes optimal aggregator function. Based on this, the authors propose a practically usable version of their theoretically optimal aggregator function. This practical version is tailored for linear probing with the cross-entropy loss. Through analysis and experiments, they demonstrate its superiority over baseline methods (like full retraining and consensus-based retraining) in high label noise regimes. The paper distinguishes its method from existing "full retraining" and "consensus-based retraining" schemes, which use hard predicted labels or only samples where predictions match noisy labels, respectively.

**Questions:**

1. How can you make a connection of the cases you study in the paper to other non-linear deep learning models? I would like to see the discussion on the practical computational overhead of the optimal method, particularly concerning the complexity of the optimal aggregator and its derivatives within the AMP updates, and how this might impact its usage on very large models or datasets.

**Ethical Concerns:**

["NO or VERY MINOR ethics concerns only"]

**Final Justification:**

The reviewers provide detailed reponses and additional experiments to answer my question. The concern regarding the generalization of the method to multi-class cases and other types of label noise have been resolved, while the concern regarding the generalization of the framework to non-linear model is still not clear, which the author left the detailed work for the future.

**Limitations:**

Yes.

**Quality:**

3

**Strengths And Weaknesses:**

Strength:
1. The paper presents a good theoretical framework based on AMP to analyze iterative retraining. And based on the theoretical analysis, the paper develops a strategy to combine given labels and predicted labels for linear probing with the binary cross-entropy loss.
2. The paper is the first work to analyze the optimal way to combine predicted and given labels for any retraining-like idea.
3. The paper has a good writing structure and clearly articulates the problem statement and its motivations

Weakness:
1. The current theoretical results are primarily for linear models and binary classification. Although not necessary for this paper, it should be better to try the framework in a more general setting, like self-training for other deep learning models. At least one case should make the method more scalable and convincing research communities that it can be incorporated into recent AI system.
2. Although the theoretical results are solid, the experimental results are using uniform noise. It is unknown how it can generalize to more complex scenarios like data dependent noise.

---

> ### Author Rebuttal · Authors · 2025-07-31
>
> Thanks for your review and feedback! You have raised good points and we respond to your concerns and questions below.
>
> **Extension of theory to multi-class case.** This can be done by following similar ideas in the paper for the binary case, but will be heavier in notation and will involve more tedious analysis. Here we outline how this extension can be made for the GMM case in Section 3. We first describe the setting. We define a label matrix $Y \in \mathbb{R}^{n \times k}$, where $n$ is the number of samples and $k$ is the number of classes, using one-hot encoding, where column $i$ has $1$ for the samples belonging to class $i$ and $0$ otherwise. Next, we define $M \in \mathbb{R}^{k \times d}$ where row $i$ is the mean feature vector of class $i$. Equation (1) then transforms to $X = YM+Z$, where $Z$ is random noise; so this is a low rank matrix ($YM$ is of rank $k$) plus noise. We can then generalize the AMP iterations for this low-rank model. The forms of eqs. (4) and (5) in this case will be the same with $g_t$ redefined appropriately for the matrix case (here it will be applied row-wise instead of coordinate-wise). Also the coefficient $C_t$ in line 126 will now be a $k \times k$ matrix, defined similarly using the Jacobian of $g_t$ instead of derivative. Likewise, the state evolution recursion (7) will be of the same form, with the modification that $m_t, \sigma_t, \bar{m_t}, \bar{\sigma}_t$ are now vectors of length $k$. The insights from the analysis will be similar but the analysis will be on multi-dimensional objects. A similar extension can be made for GLMs in Section 4, where $\beta$ should now be defined as a matrix of size $k \times d$. The extension here for the AMP iterates (16-17) and the state evolution (21) are similar to the GMM case described above.
>
> **Extension of theory to non-linear models.** We note that the AMP techniques can be used to analyze the statistical behavior of more general M-estimators (see for e.g., reference [14, section 4.4] in the paper). The corresponding update rule will involve applying the aggregator function (aggregating current predictions and the given noisy labels) as well as a gradient descent type of update (similar to eq. (4)). The exact details of these updates and the analysis will be quite tedious and are left for future work.
>
> **Extension in practice to different types of label noise (beyond uniform) and deep learning models.** Although our theoretical results focused on linear models, the algorithm is Section 5 proposed for use in practice, i.e. BayesMix RT, is based on insights that carry over to general ML models. Specifically, eq. (27) is obtained by fitting a GMM on the distribution of the unnormalized logits (following the derivation in eq. (30) for theory). Following the same approach (and even in multi-class setting), we can rewrite the Bayesian posterior of the class memberships, after observing the current model's unnormalized logits with the prior based on the label noise model (in our paper, we have considered the uniform prior). We can then derive the corresponding aggregating function $g$ similar to eq. (27) in the paper. In short, *BayesMix RT can be used in deep learning models and even with non-uniform label noise priors* (e.g., when the flipping probability from class i to class j is some $p_{i, j}$). As an example, in the binary case, consider the **non-uniform noise model**: $\mathbb{P}(\hat{y} = -1| y = +1) = p$ and $\mathbb{P}(\hat{y} = +1| y = -1) = q$ with $p \neq q$. Then, eq. (27) changes to:
> $$g(z, \hat{y}) = \frac{2}{1+ \Big(\frac{q(1-q)}{p(1-p)}\Big) \Big(\frac{p q}{(1-p)(1-q)}\Big)^{\hat{y}/2} \exp\Big(\frac{(z - \mu_{+})^2}{2 \sigma_{+}^2} - \frac{(z - \mu_{-})^2}{2 \sigma_{-}^2}\Big) \frac{\pi_-}{\pi_+}} - 1.$$
> We show an experimental result under the above **non-uniform noise model** with $p=0.45$ and $q=0.2$ for the MedMNIST Pneumonia dataset in the same setting as Section 5 using the formula above for BayesMix RT; the table below lists the average *test* accuracies $\pm$ standard deviation (over 3 runs) in this case. As we see below, **BayesMix RT performs the best**.
>
> |Iteration #| Full RT| Consensus-based RT| BayesMix RT (ours)|
> |---|---|---|---|
> |0 (initial model)|$71.85 \pm 1.18$|$71.85 \pm 1.18$|$71.85 \pm 1.18$|
> |1| $78.21 \pm 2.04$ | $77.51 \pm 1.92$ | $\textbf{81.14} \pm 0.46$ |
> |10| $81.62 \pm 2.16$ | $82.26 \pm 2.36$ | $\textbf{84.62} \pm 0.73$ |
>
> For more **general types of label noise** (e.g., sample-dependent noise) or in scenarios **where the noise model is not known**, we propose the following **simpler aggregation function** (instead of eq. (27)):
> $$g(z, \hat{y}) = \frac{2}{1+ \gamma^{\hat{y}} \exp\Big(\frac{(z - \mu_{+})^2}{2 \sigma_{+}^2} - \frac{(z - \mu_{-})^2}{2 \sigma_{-}^2}\Big) \frac{\pi_-}{\pi_+}} - 1,$$
> for some constant $\gamma \in (0,1)$ which can be tuned. We call this simpler version **BayesMix-Simple RT** (everything else is the same as BayesMix RT in the paper); in our implementation, we set $\gamma = 0.7$ throughout without any tuning though. Please note that *this can be used with general ML models*.
>
> **(1)** To show the **efficacy of BayesMix-Simple RT**, we first consider an **adversarial noise model**. Specifically, an adversary gets to determine the "important" samples and flip their labels. The adversary determines important samples by training the same model which will be trained by us *with clean labels* and orders the samples in the decreasing order of the absolute value of the unnormalized logits. Note that a higher logit absolute value implies that the model is very confident on the sample, so flipping the label of such a sample would be more detrimental to the model's training as it would need to adjust the decision boundary by a large amount for this incorrectly labeled sample. The adversary then flips the labels of the top $\alpha$ fraction of the samples with largest absolute values of the unnormalized logits. Under this **adversarial noise model** (we don't assume knowledge of this process in training to be clear) with $\alpha=0.25$ for MedMNIST Pneumonia in the same setting as Section 5, we show the efficacy of **BayesMix-Simple RT** below compared to the baselines (again, the table below lists the average *test* accuracies $\pm$ standard deviation over 3 runs).
>
> |Iteration #| Full RT| Consensus-based RT| BayesMix-Simple RT (ours)|
> |---|---|---|---|
> |0 (initial model)|$63.94 \pm 1.51$|$63.94 \pm 1.51$|$63.94 \pm 1.51$|
> |1| $66.83 \pm 1.93$ | $66.77 \pm 1.78$ | $\textbf{67.15} \pm 1.83$ |
> |10| $73.77 \pm 0.79$ | $76.76 \pm 2.51$ | $\textbf{78.04} \pm 1.98$ |
>
> **(2)** We now show the **efficacy of BayesMix-Simple RT** even for **deep network training**; recall that our earlier experiments are for linear probing with a pre-trained ResNet-50. Specifically, here we consider *training an **entire** pre-trained ResNet-50*; this is a **typical deep learning** setting. Because of the heavy computational cost of training an entire deep network, we only do 3 iterations of retraining here. We show results for two cases of Food-101:
>
> **i.** *Pho vs. Ramen corrupted by uniform noise with $p=0.4$*
> |Iteration #| Full RT| Consensus-based RT| BayesMix-Simple RT (ours)|
> |---|---|---|---|
> |0 (initial model)|$61.13 \pm 3.39$|$61.13 \pm 3.39$|$61.13 \pm 3.39$|
> |1| $59.07 \pm 5.44$ | $59.33 \pm 4.83$ | $\textbf{66.73} \pm 3.72$ |
> |3| $57.67 \pm 7.47$ | $58.47 \pm 6.78$ | $\textbf{70.67} \pm 4.65$ |
>
> **ii.** *Spaghetti Bolognese vs. Spaghetti Carbonara corrupted by non-uniform noise described above with $p=0.45$ and $q=0.2$*
> |Iteration #| Full RT| Consensus-based RT| BayesMix-Simple RT (ours)|
> |---|---|---|---|
> |0 (initial model)|$65.45 \pm 2.22$|$65.45 \pm 2.22$|$65.45 \pm 2.22$|
> |1| $61.35 \pm 7.31$ | $ 61.30 \pm 4.48$ | $\textbf{74.55} \pm 3.75$ |
> |3| $56.50 \pm 7.74$ | $59.70 \pm 5.57$ | $\textbf{80.30} \pm 5.58$ |
>
> Note that in the above two tables (for full ResNet-50 training), **BayesMix-Simple RT leads to significant gains** over the initial model, while Full RT and Consensus-based RT are actually detrimental.
>
> *So in summary, our proposed method is effective with different label noise models as well as for general ML models.*
>
> Regarding your question about practical computational overhead of our methods BayesMix RT and BayesMix-Simple RT, note that the major extra cost compared to Full RT and Consensus-based RT is in fitting the bimodal GMM to obtain the aggregation function $g$. But this is negligible when compared to the training (backward and forward propagation) cost of every epoch, especially for large models. So our proposed methods do not introduce too much extra overhead. It is worth mentioning that for larger models, having multiple rounds of retraining will be expensive. Also, just to clarify the AMP updates in eqs. (4-5) and (16-17) are for theory only. While the insights obtained from the AMP analysis are used to propose BayesMix/BayesMix-Simple RT, they do not use the specific AMP updates. Having said that, even if we consider the AMP updates in eqs. (4-5) for GMM (Sec 3), our algorithm has the same effective per-iteration complexity as Full RT and Consensus-based RT of $O(nd)$ due to the two matrix-vector multiplications (those involving $X$ and $X^\top$). The only difference is in applying the optimal aggregator $g_t$ and also computing the correction (Onsager) term $C_t$; both of these are $O(n)$ which is negligible compared to $O(nd)$ for large $d$. A similar argument holds for the GLM case (Sec 4).
>
> We will add all this discussion in the next version of our paper. We hope to have resolved your concerns and questions, but we're happy to discuss further if needed. If you're satisfied with our responses, **we sincerely hope you will raise your score**!

---

> > ### Author Response · Authors · 2025-08-05
> >
> > Dear Reviewer,
> >
> > The discussion period deadline is approaching and we haven't heard back from you. We'd appreciate it if you could go through our rebuttal and let us know if you have any other questions. Thank you!

---

> > > ### Comment · Reviewer_ZTCf · 2025-08-05
> > >
> > > Thank you for your detailed response and experiments. I will raise my score.

---

> > > > ### Author Response · Authors · 2025-08-05
> > > >
> > > > Thank you for raising your score!

---

### Decision · Program_Chairs · 2025-09-17

**Decision:**

Accept (poster)

**Comment:**

This paper is focused on retraining—i.e. first training a predictor and then re-training that predictor on some combination of the data labels and the outputs from the predictions in the previous iteration. This paper analyzes the behavior of such rules in two cases: classification with binary isotropic GMMs, and GLMs more generally. In each case optimal retraining weights are derived under a model where the observed labels are the true ones with some probability of corruption. The reviewers appreciated this problem setting and contribution, so I recommend acceptance.

That said, I (the AC) have read this paper in depth and discussed it with the SAC. I suggest that the paper is in some respects currently quite sub-optimal as regards clarity. The mathematical writing is generally a bit sloppy. For example, here are some ways the paper might be improved:

- The justification for Equation 3 is quite sparse. This can be seen as exactly minimizing Σ_n ||x_n y_n - θ||^2, or "approximately" minimizing Σ_n ||y_n - θ^T x_n||^2. The given references also don't contain much justification.

- The scheme in Equations 4-5 seems to be off by a factor of \sqrt{n} when there is no label noise. This is explained as a result of being split between the two updates (for θ and y) but this doesn't quite seem to make sense—with zero noise, won't θ still be off by a factor of \sqrt{n}? This isn't considered a huge issue since classification error doesn't depend on the magnitude.

- Theorem 3.1 is a bit odd, on several grounds. First, equation 8 pertains to the average of some function \psi over the dimensions of θ. It's not entirely obvious why such an expectation is of interest or what features of the mean or (co)variance of θ or y could be recovered from this and no discussion is given. Second, the text above theorem 3.1 states that this describes the mean and variance of θ and y, but y does not appear in Equation 8 or 9.

- I (the AC) and the SAC were both initially confused by Eq. 13. It was not immediately obvious to us that this yielded a sensible update rule in the limit of zero noise. After consideration, we see that it does, but it's due to a relatively subtle relationship of the limit of p and \hat{y} that should be made explicit.

- It should be stressed more strongly that actually using the update rules requires knowledge of quantities like p, π, γ, which may not always be available in practice.